# Demonstrating Aeolus capability to observe wind-cloud interactions

Zacharie Titus[1], Marine Bonazzola[1], Hélène Chepfer[1], Artem G. Feofilov[1], Marie-Laure Roussel[1], Benjamin Witschas[2], Sophie Bastin[3]

[1]Laboratoire de Météorologie Dynamique/Institut Pierre Simon Laplace (IPSL), Sorbonne Université, CNRS, Institut Polytechnique de Paris, Paris, France
[2]Deutsches Zentrum für Luft- und Raumfahrt e.V. (DLR), Institut für Physik der Atmosphäre, 82234 Oberpfaffenhofen, Germany
[3]Laboratoire ATmosphere Milieux Observations Spatiales/Institut Pierre Simon Laplace (IPSL), UVSQ Université Paris-Saclay, Sorbonne Université, CNRS, CNES, Guyancourt, France

*Correspondence to*: Zacharie Titus ( zacharie.titus@lmd.ipsl.fr )

**Abstract.** Model based studies have shown interactions between wind vertical profiles and cloudiness, but few observational studies corroborate them. The unique observations of Aeolus spaceborne Doppler wind lidar can contribute to fill this gap. In this paper, we merged global Aeolus observations of cloud profiles at full horizontal resolution (3 km along orbit track) with co-located profiles of horizontal winds.

We first observed wind-cloud interactions at regional scale over the Indian Ocean. Aeolus captures the strengthening of the Tropical Easterly Jet in early June 2020, with wind speeds exceeding 40 ms$^{-1}$ in its core, and a simultaneous increase of high cloud fraction up to above 30 %, until the decay of the jet during fall.

Secondly, we observed wind-cloud interactions at cloud scale (between 3-100 km) in different regions. Over the Indian Ocean as well as over cumulus and stratocumulus dominated regions, we found that the wind shear inside clouds is smaller than the wind shear in the clear sky surrounding the clouds (statistically significant). In addition, we found that the wind speed difference between the cloud and its surrounding clear sky increases with the clear sky wind shear, especially in cumulus (R=-0.94) and stratocumulus (R=-0.87) dominated regions. This study demonstrated that despite its coarse resolution, Aeolus can capture wind perturbations induced by convective motion.

 **1 Introduction**

Clouds play a critical role in Earth's climate as a major component of the water vapor cycle and because they have a large impact on the radiative budget at the top of the atmosphere and at the Earth surface. The formation and development of clouds are controlled by the surface temperature and by the thermodynamic structure of the lower troposphere, but also by dynamic variables. It was shown that fast horizontal winds are responsible for an increased cirrus cloud cover through different mechanisms like advection of humidity from warmer to cooler regions, favoring the in-situ formation of cirrus clouds (Das et al., 2011). Deep convective cloud systems tend to form in regions of large-scale wind convergence. They organize into rain bands and squall lines by the wind shear (e.g., Thorpe et al., 1982 ; Rotunno et al., 1988 ; Parker, 1996 ; Hildebrand, 1998 ; Robe and Emanuel, 2001 ; Weisman and Rotunno, 2004 ; Abramian et al., 2022). The wind shear can also inhibit deep and shallow convection by "blowing off" cloud tops (e.g., Koren et al., 2010 ; Sathiyamoorthy et al., 2004), or increase the cloud cover by tilting cloud tops away from their base (Mieslinger et al., 2019), thus influencing cloud-top height and cloud cover (Helfer et al. 2020). Over marine boundary layers, the wind shear can even locally deplete stratocumulus cloud tops (Wang et al. 2008 ; Schulz and Mellado, 2018). Reversely, clouds can have an influence on winds through their radiative effect. Fujiwara et al. (2004) showed that the radiative cooling associated to anvils creates a temperature gradient at the top of high convective clouds, that can generate a thermal wind. At a large scale, it was shown that the cloud radiative effect impacts the intensity and location of the jet stream by altering temperature gradients and redistributing energy within the atmosphere (Voigt et al, 2021).

To better understand wind-cloud interactions, a large number of studies have been performed. These studies are based on models or meteorological analyses. Observations of winds within cloudy systems are usually performed by radiosondes, airborne or ground based Doppler Radars, and are therefore limited in space and time. In this study, we benefit from the unique capabilities of the Atmospheric LAser Doppler INstrument (ALADIN), a 355 nm spaceborne Doppler Wind Lidar with High Spectral Resolution (HSRL) capabilities onboard the Aeolus satellite (Stoffelen et al., 2005 ; Reitebuch et al., 2012). Aeolus is primarily designed to retrieve profiles of horizontal winds but can also retrieve profiles of clouds (Flamant et al., 2008 ; Dabas et al., 2022 ; Feofilov et al., 2022). During its 5 years of operation, Aeolus scanned over a billion kilometers of atmosphere around the globe (Aeolus DISC, 2024), encountering all kinds of cloudy systems at various latitudes. Aeolus thus offers for the first time the possibility to analyze, at global-scale, co-located instantaneous profiles of clouds and profiles of horizontal winds within clouds and their clear sky surroundings.

In its current state, studying wind-cloud interactions with Aeolus is challenging. First of all, clouds can be as little as a few tens of meters horizontally (Koren et al., 2008), cloud detection thus needs to be performed at the highest possible spatial

resolution in order to avoid mixing clear and cloudy scenes. Recent work showed that it is possible to perform cloud detection at full horizontal resolution of 3 km (Donovan et al., 2024b ; Wang et al., 2024). Moreover, the wind profiles are available in a different Aeolus product with a different along-track resolution, therefore an additional processing is necessary to merge clouds and winds.

Aeolus is primarily designed to retrieve vertical profiles of horizontal winds in the troposphere and the lower stratosphere at global-scale. The laser is pointed 35° off-nadir and perpendicular to the satellite track, away from the Sun. The obtained measurement is not the actual horizontal wind, but the horizontal projection of the wind retrieved along Aeolus Line-of-Sight (LOS, Fig. 1). In most of the Aeolus literature, this wind is noted $v_{HLOS}$. Hereafter, we use the wind profiles from Aeolus

Level 2B (L2B, Baseline 16) scientific wind product. These wind profiles come from two channels. A "Mie channel" retrieves wind within entire optically thin clouds, which cover typically 35 % of the globe on average (Guzman et al., 2017) but also within the upper layers of opaque clouds, which cover typically 31 % of the globe on average. The "Rayleigh channel" retrieves wind in clear sky, which covers the remaining 34 % of the globe on average. In order to fully benefit from Aeolus observations to better understand wind cloud interactions, it is necessary to resample the Aeolus wind profiles at the

same fixed resolution as the cloud profiles.

The purpose of this paper is to evaluate the feasibility of studying wind-cloud interactions from large scale to cloud scale (between 3-100 km), making use of our dataset of merged global Aeolus observations of cloud profiles at full horizontal resolution (3 km along orbit track) with co-located profiles of horizontal winds. At a large scale, we particularly focus on the

80 relationship between high cloud cover and the Tropical Easterly Jet (TEJ) over India. At a lower scale, we evaluate the benefit of our observations for the validation of the K-theory for the wind.

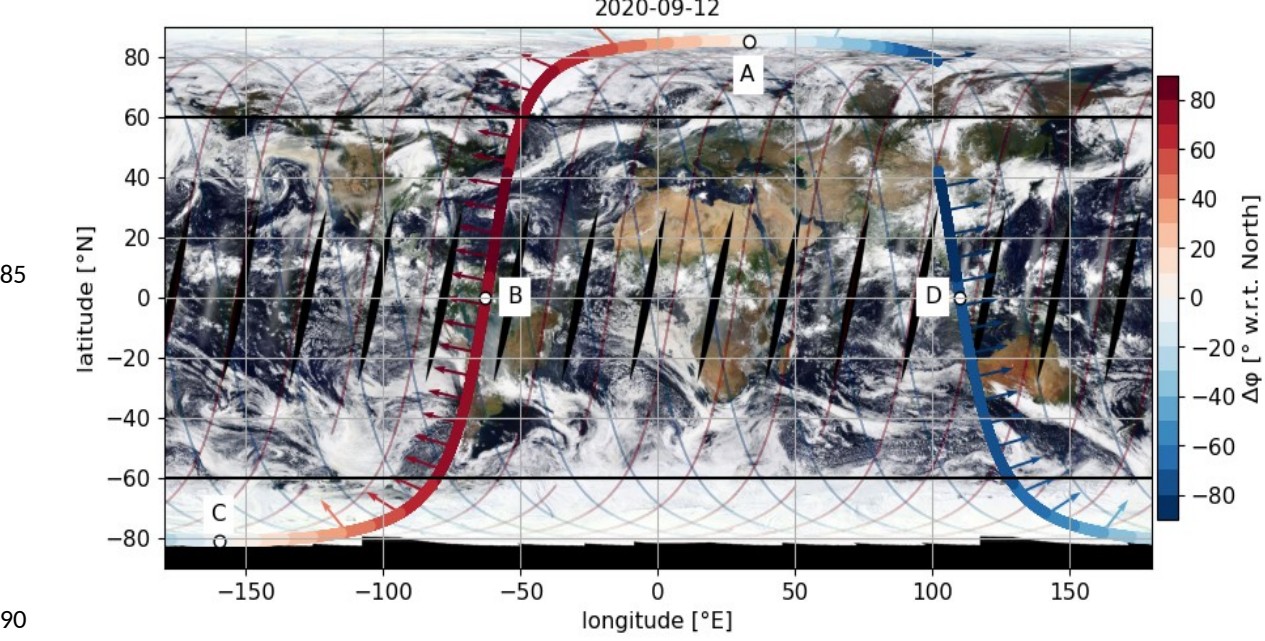

**Figure 1: Track of one Aeolus orbit (2020-09-12T09 – 2020-09-12T11). Aeolus retrieves the projection of wind aligned with the arrows. Δφ represents the angle between the South-North axis and the laser pointing direction, counter-clockwise. Thinner curves represent all obits for the day of 2020-09-12. A is the North-most point. B is the equatorial crossing point during descending phase (0600 LT). C is the South-most point. D is the equatorial crossing point during ascending phase (1800 LT)**

Section 2 of the paper details the method used to retrieve profiles of clouds. In Sect. 2.2 we assess the quality of this cloud detection by comparing it to another cloud climatology obtained with CALIPSO-GOCCP. In Sect. 3, we re-sample Aeolus L2B Mie and Rayleigh wind observations at 3 km of resolution along track and 480 m vertically and merge them using the cloud mask. We also quantify how much of the natural "sub-grid" variability is missed when re-sampling the wind from 87 km to a higher horizontal sub-grid resolution using high spatial resolution airborne Doppler Wind Lidar (DWL) data during AVATAR-T (Aeolus Validation Through Airborne Lidars in the Tropics) campaign, and using a high spatial resolution simulation performed with the Weather Research and Forecasting (WRF) model. Finally, in Sect. 4, we present the first descriptive results that we obtain with this dataset, focusing on different horizontal scales. We study the Tropical Easterly Jet and its correlation with high cloud fractions. We also assess the difference between cloudy and clear sky winds at cloud scales inferior to 100 km. We conclude this paper in Sect. 5.

## 2 Processing Aeolus clouds

### 2.1 Building cloud diagnostics from Aeolus particulate and molecular backscatter profiles

Hereafter, we build Aeolus cloud statistics based on a cloud mask defined at 480 m vertical resolution and best possible
horizontal resolution, to compare Aeolus data with CALIPSO-GOCCP (Chepfer et al., 2010) and to facilitate future use of
Aeolus data by the climate model community through the COSP Lidar Simulator (Bodas-Salcedo et al., 2011). To build this
cloud mask from Aeolus particulate and molecular backscatter profiles, we follow an approach similar to the one proposed
by Donovan et al., 2024b, with the following additions: a cross-polar correction from CALIPSO-GOCCP and a dedicated
processing of hot pixels. In this approach, we use Aeolus Level 1A (L1A) raw data, with a horizontal resolution of 3 km. We
only use the radiance retrieved by the detector of Aeolus Mie channel. The spectrum of the radiance illuminating the 16
pixels of the detector consists of a superposition of a narrow peak related to a particulate backscatter and a several times
broader peak associated with molecular backscatter (Fig. A1). The position of the centre of the joint envelope represents the
direction and the strength of the wind, whereas the integral of the signal is proportional to a total attenuated backscatter.

For a given profile, we process the spectrum measured by the detector of the Mie channel at each altitude level in six
successive steps.

1) Discard "hot pixels".
It has been known since the early days of the Aeolus mission that certain pixels of the detector are damaged by cosmic
particles and that the number of these pixels almost linearly increases over the mission's lifetime (Weiler et al., 2021). We
discard the hot pixels following the hot pixel map of the 31 December 2020 (Fig. A2), which corresponds to the end of the
period that we considered in our study. Once the hot pixel is discarded, we apply a "sliding fit" approach (Goldberg et al.,
2013; Feofilov and Stubenrauch, 2019) adapted to Aeolus (Feofilov, 2021), which considers the theoretical shape of the Mie
backscatter spectrum. In this approach, a predefined spectral shape function is systematically shifted across each row of the
Mie channel detector to find the optimal fit that minimizes the difference between the observed and theoretical spectral
profiles, thereby simultaneously determining the Mie peak center frequency and reconstructing the complete spectral
radiance values.

2) Intensities of the particulate backscatter and molecular backscatter in arbitrary unit.
For each profile and each altitude level, we subtracted the Detection Chain Offset (DCO, more details are given in Fig. A1),
the solar background, and compensated for the non-uniform intensity distribution on the Mie spectrometer following
Donovan et al., (2024b). Future work could include cross-talk correction. Then, based on the peak position, we selected eight

pixels either to its left or right. The signal retrieved in the pixels corresponding to the peak and the two following pixels are summed and correspond to the particulate backscatter, called $I_{part}(z_{L1A})$ here after, $z_{L1A}$ being the altitude of the centre of a

layer in a L1A profile. The signals retrieved in the six remaining pixels are summed and correspond to the molecular backscatter, called $I_{mol}(z_{L1A})$. Although the molecular or Rayleigh signal in this approach does not represent the actual molecular backscatter, it is proportional to it, enabling us to use the difference between $I_{part}(z_{L1A})$ and $I_{mol}(z_{L1A})$ to determine the cloud mask.

3) Constant vertical and horizontal resolutions.

To detect clouds consistently at all locations and all times, we need the intensities of the molecular and particulate signals at a fixed vertical and a fixed horizontal resolutions. Indeed, variations in the resolutions influence these quantities because a different volume of the atmosphere is probed. Aeolus L1A profiles have a fixed horizontal resolution (3 km) but a variable vertical resolution along the orbit. As the number of layers along the vertical is fixed (24 bins) but the altitudes of the top of

the vertical profiles vary between 15 and 25 km, the vertical resolution $\Delta z_{L1A}$ of Aeolus L1A layers varies along the orbit and ranges between 500 m in the boundary layer and up to 1 km in the free troposphere (Reitebuch et al., 2018). To detect clouds consistently at all locations and all times, we linearly interpolate the molecular and particulate signals at a fixed vertical resolution of $\Delta z$ = 480 m, similar to the one used in CALIPSO-GOCCP (Chepfer et al. 2010), from the sea-level up to 19.2 km of altitude. These new proxies are noted $I_{part-alt}(z)$ and $I_{mol-alt}(z)$ and are defined as :


$$I_{part-alt}(z) = I_{part}(z_{L1A}) \frac{480}{\Delta z_{L1A}} \tag{1}$$

$$I_{mol-alt}(z) = I_{mol}(z_{L1A}) \frac{480}{\Delta z_{L1A}} \tag{2}$$

where $z$ is the altitude of the centre of a 480 m layer in the new vertical scale. Note that the choice of a 480 m vertical resolution implies possibly losing portions of gradients from the original Aeolus dataset due to altitude mismatches in the

original and re-sampled datasets.

4) Depolarization correction.

ALADIN's emission is circularly polarized but the receiver is only able to measure the co-polarized component of the backscattered light. It misses the cross-polarized component. Backscattering by non-spherical particles modifies the state of

polarization of light. Therefore the intensity of the particulate backscatter measured by ALADIN is underestimated within mixed phase clouds and ice clouds that contain non-spherical particles. To compensate for this, we use a monthly

climatology of the depolarization ratio ($\delta P$) from CALIOP/CALIPSO observations (Feofilov et al., 2022) to correct $I_{part\text{-}alt}(z)$ as follows :

$$I_{part-alt-\delta P}(z) = \frac{I_{part-alt}(z)}{(1-\delta P)}$$

(3)

The output files at this stage are thus orbit files containing profiles of proxies of particulate $I_{part\text{-}alt\text{-}\delta P}(z)$ and molecular backscatter $I_{mol\text{-}alt}(z)$ at a fixed resolution of 3 km along orbit track and resampled at 480 m vertically from the surface up to 19.2 km of altitude.

5) Cloud detection.

For each profile, a layer is declared cloudy when $I_{part\text{-}alt\text{-}\delta P}(z) - I_{mol\text{-}alt}(z)$ exceeds a certain threshold. Aerosol layers are classified as clear sky. More details about the calculations of the threshold are given in appendix A and Fig. A3.

6) Fully attenuated bins

Below an opaque cloud, the laser is fully attenuated, making it impossible to retrieve valuable information, neither for the
cloud detection, nor for the wind. For each profile containing a cloud, we evaluate $I_{mol\text{-}alt}(z)$ at each layer between the surface and 1 km below the lowest cloudy layer. If $I_{mol\text{-}alt}(z)$ at each layer is inferior to the noise level, all the layers between the surface and the lowest cloudy layers are flagged as fully attenuated. Otherwise, they are flagged as clear sky. For each orbit the noise level is simply defined as three times the standard deviation of $I_{mol\text{-}alt}(z)$ between 60°S and 40°S and between 16 and 18 km of altitude.


At this stage, the output files are orbit files containing a cloud mask at a 3 km along-track resolution and a 480 m vertical resolution as shown later (Fig. 5b).

7) From cloud mask orbits, we compute daily gridded profiles of cloud fraction over 2° latitude x 2° longitude grid boxes.
For each 480 m thick layer, the cloud fraction is the ratio between the number of "cloudy" bins encountered within the grid box for the considered day at this vertical level, and the total number of non-attenuated bins observed within the same grid box at the same vertical level as described in Chepfer et al., (2010).


**2.2 Evaluation of Aeolus clouds against a CALIPSO-GOCCP climatology**

**2.2.1 CALIPSO-GOCCP dataset**

To assess the quality of our cloud detection, we compare it to independent cloud observations retrieved from another space
lidar. The GCM Oriented CALIPSO Cloud Product from the Cloud-Aerosol Lidar and Infrared Pathfinder Satellite
Observation, (CALIPSO-GOCCP, Chepfer et al., 2010) displays cloud profiles at a 333 m horizontal resolution and a 480 m
vertical resolution from 2006 to 2023, and thus, overlaps the Aeolus mission during over 4 years between 2018 and 2023.
We used CALIPSO-GOCCP version 3.1.4, in which low laser energy shots are discarded. An in-depth comparison between
CALIPSO-GOCCP and Aeolus clouds has already been done (Feofilov et al., 2022) but using scattering ratios derived from
Aeolus Level 2A calibrated optical properties data (instead of L1A data here) at a coarser resolution of 87 km along orbit
track. Even though CALIOP is also a space lidar, differences between CALIOP and ALADIN listed hereafter lead to
differences in cloud detection that need to be kept in mind in the comparison:

- CALIOP (Winker et al., 2004) operates at 1064 and 532 nm while ALADIN operates at 355 nm.

- CALIOP points at 3° off-nadir while ALADIN points at 35° off-nadir.

- CALIPSO follows a sun-synchronous orbit, with its ascending (resp. descending) equatorial crossing occurring at 1330 LT
(resp. 0130 LT), while Aeolus ascending and descending equatorial crossings respectively occur at 1800 LT and 0600 LT.
Therefore, close co-locations between the two instruments are rare, and the diurnal cycle of clouds (Noel et al., 2018;
Chepfer et al., 2019), is corrected using the Cloud–Aerosol Transport System (CATS) onboard the International Space
Station (McGill et al., 2015) data applied to CALIPSO-GOCCP between 60°S and 60°N as detailed in Feofilov et al., 2022.

- CALIOP is polarization-sensitive, ALADIN is not although we compensate the particulate backscatter by a climatology of
the depolarization ratio observed by CALIOP.

- In GOCCP, the bin encompassing the surface can contain information about its cloudiness while it is systematically
discarded with Aeolus.

- The horizontal along orbit track resolution is 333 m for CALIPSO-GOCCP and 3 km for Aeolus. For a consistent
comparison between the two instruments, we build the CALIPSO-GOCCP-COARSE dataset, whose spatial resolution is set
to the same as Aeolus (3 km) prior to the cloud detection.


**2.2.2 Comparison of Aeolus and CALIPSO-GOCCP-COARSE cloud climatology**

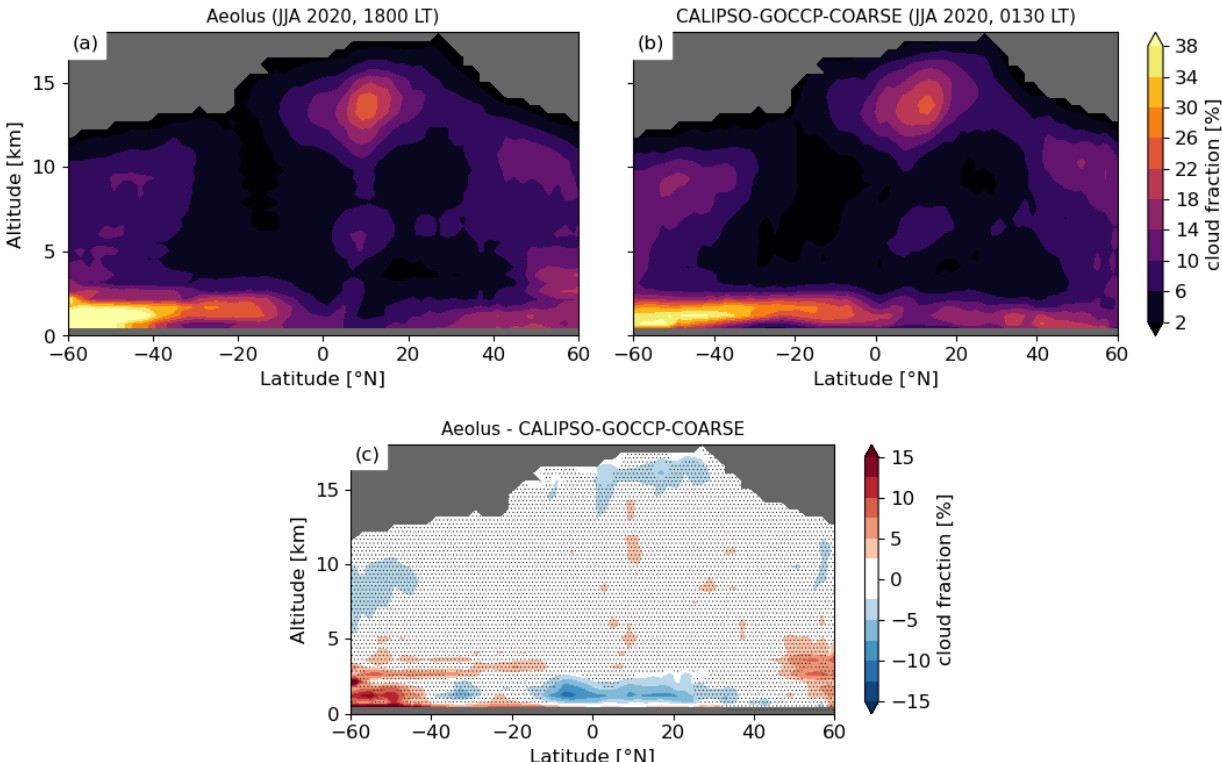

**Figure 2: Zonal average cloud fraction profiles for (a) Aeolus at 1800 LT and (b) CALIPSO-GOCCP-COARSE at 0130 LT corrected for the diurnal cycle (c) is the absolute difference of cloud fraction between Aeolus and CALIPSO-GOCCP-COARSE. Non-significant differences (two sided T-test with p-values > 0.05) are dotted. The lowest bin encompasses the surface and is discarded in this study (opaque gray bar). Cloud fractions < 1% are masked in gray.**

We compare the zonal average of cloud fraction profiles retrieved from ALADIN (Fig. 2a) to those retrieved from
CALIPSO-GOCCP-COARSE (Fig. 2b) between June and August 2020. Overall, CALIPSO-GOCCP-COARSE and Aeolus show similar cloud patterns. The cloud fractions are in good agreement with $R^2$=0.84 and Pearson correlation of 0.92. In both cases a local maximum of cloud fraction of about 25-30% are found around 10° N within the inter tropical convergence zone (ITCZ), between 12 and 15 km. Minima of cloud fractions with Aeolus and CALIPSO-GOCCP-COARSE appear within the tropical region on each side of the Equator in the middle troposphere, within the descending branch of Hadley circulation. In
Fig. 2c, we see that cloud fraction differences remain lower than 2.5% within most of the troposphere and are non-significant almost everywhere (two sided T-test with p-values > 0.05, dotted bins). Below 2 km of altitude and between 10°S-25°N, Aeolus retrieves cloud fraction about 5-10 % smaller (significant) than CALIPSO-GOCCP-COARSE between 30°S and 10°S, the laser of Aeolus being more often fully attenuated in the free troposphere by high clouds.

## 3 Processing Aeolus winds

The wind profiles from Aeolus Level 2B (L2B) scientific wind product have been continuously validated during the mission with airborne lidars (Lux et al., 2020; Witschas et al., 2020; Witschas et al., 2022), ground based lidars, radars and radiosondes (Ratynski et al., 2023; Iwai et al., 2021; Belova et al., 2021; Baars et al., 2020). So far, Aeolus wind data (L2B) provided to the community are orbit files that contain 2 types of wind profiles (the Mie wind and the Rayleigh wind) estimated from the molecular and particulate backscattered signals respectively. The latest validation report of Aeolus showed systematic error (bias) of below 0.5 ms$^{-1}$ for Mie winds and below 1 ms$^{-1}$ for Rayleigh winds, while the random error is about 3 to 4 ms$^{-1}$ for Mie winds and 3 to 6 ms$^{-1}$ for Rayleigh winds (Aeolus DISC, 2024). This study benefits from the latest reprocessing of L2B Baseline 16.

The Mie and Rayleigh wind profiles have a varying vertical resolution (500 m to 1 km) but also a varying horizontal resolution (ranging from 3 km to 15 km in the Mie channel and fixed at 87 km in the Rayleigh channel). Having a dataset with Aeolus wind profiles resampled at the same fixed resolution as Aeolus cloud profiles is crucial to ease the use of these data for wind cloud interaction studies. In this section we explain how we merge these two wind datasets making use of the cloud mask defined in Sect. 2. In a nutshell, our method consists in re-sampling Rayleigh and Mie winds by interpolating them at the same resolution as the cloud mask (3 km horizontally along orbit track and 480 m vertically), and then selecting the right wind (Rayleigh or Mie) based on the result of the cloud mask (clear or cloudy).

### 3.1 Re-sampling clear and cloudy sky winds and unifying them on a spatially regular curtain based on our cloud detection

We process the L2B Mie and Rayleigh wind profiles in three successive steps.

1) We first apply the prescribed quality controls for Aeolus L2B winds. We make sure that we only select the valid Mie winds (*validity_flag* = 1, *observation_type* = 1, "cloudy", *hlos_error_estimate* < 5 ms$^{-1}$) and the valid Rayleigh winds (*validity_flag* = 1, *observation_type* = 0, "clear", *hlos_error_estimate* < 9 ms$^{-1}$). The *validity_flag* ensures that the wind data have a sufficient quality. The *observation_type* flag filters out Mie wind observations when particulate backscatter is weak (no aerosols nor clouds) and filters out Rayleigh wind observations in the presence of strong particulate backscatter (clouds or aerosols). The *hlos_error_estimate* flag filters out gross outliers (Iwai et al., 2021; Lux et al., 2022).

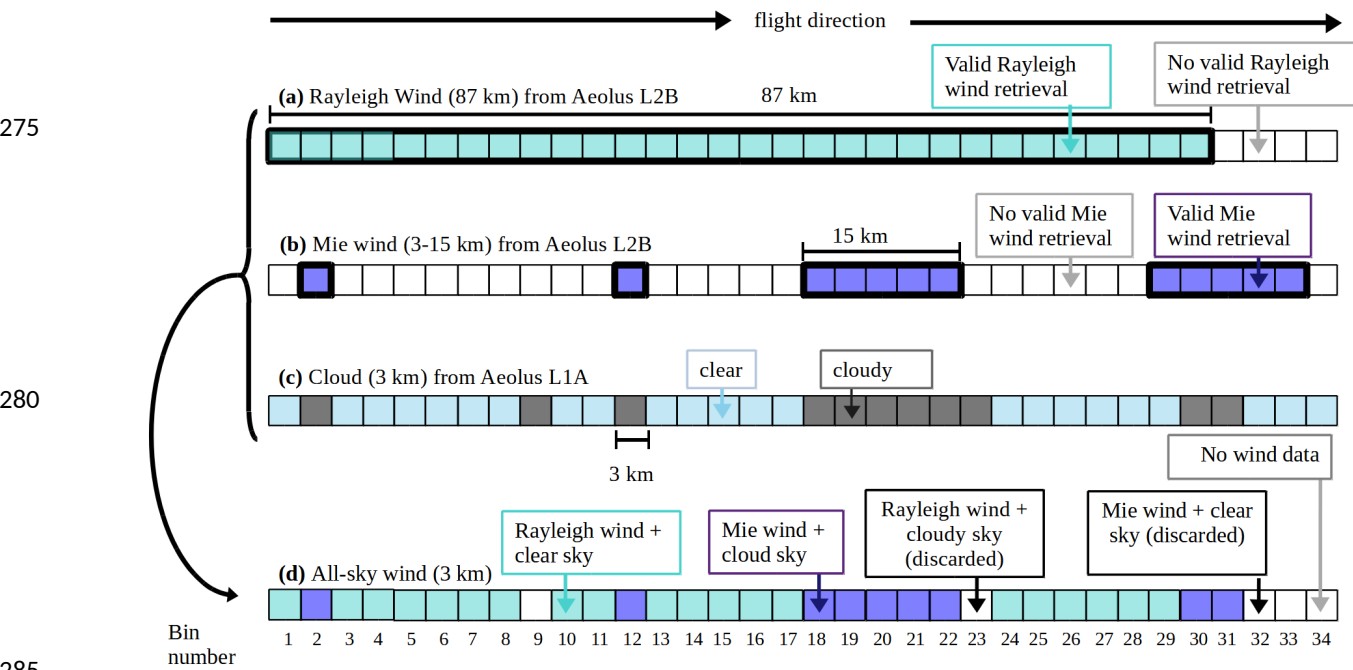

**Figure 3: Building the all-sky wind along an orbit segment at one altitude level by selecting the adequate wind from Aeolus L2B based on the cloud detection. Each square represents a bin of 3 km x 480 m. (a) Blue bins contain valid wind from Aeolus L2B Rayleigh channel (87 km resolution along orbit track). (b) Purple bins contain valid wind from Aeolus L2B Mie channel (3 to 15 km resolution along orbit track). (c) Cloud detection made from Aeolus L1A (3 km resolution along orbit track). (d) Aeolus all-sky wind at 3 km along orbit track resolution, built by compositing lines (a) to (c). Blue bins contain a valid clear sky wind, purple bins contain a valid cloudy sky wind and white bins contain no wind data.**

2) We then display each wind at a fixed resolution of 3 km x 480 m. For each 3 km x 480 m bin, we look for the spatially closest L2B Rayleigh wind, evaluated in latitude along the orbit track and in altitude relative to the bin center (Rennie et al., 2020) in a limit of 87 km, which corresponds to the horizontal resolution of Rayleigh wind observations. We duplicate its value on the 3 km x 480 m orbit file (Fig. 3a) before performing a 2D linear interpolation with a sliding average. The sliding window has the resolution of the original L2B Rayleigh wind observations of Aeolus (87 km horizontally and 500 m to 1 km vertically). Similarly, we look for the spatially closest L2B Mie wind in a limit of 15 km, which corresponds to the maximum horizontal resolution for the Mie wind. We duplicate its value (Fig. 3b) before performing a 2D linear interpolation with a sliding average. The sliding window has the resolution of original L2B Mie wind observations of Aeolus (3 to 15 km horizontally and 500 m to 1 km vertically). We thus obtain at this stage winds from both channels re-sampled at 3 km along track and 480 m vertically. A bin can contain either a Rayleigh wind (bin 26, Fig. 3a) or a Mie wind (bin 32, Fig. 3b) or both winds (bin 2, Fig. 3a and 3b) or no wind (bin 34, Fig. 3a and 3b).

3) Making use of the cloud mask (Fig. 3c), we select for each bin, either a Rayleigh (Fig. 3a) or a Mie (Fig. 3b) wind to build the all-sky wind. Consider bin 2, (Fig. 3) where both Rayleigh and Mie winds coexist. As a cloud was detected (bin 2, Fig. 3c), we select the Mie wind as an element of the all-sky wind (bin 2, Fig. 3d).  For bin 29, both Rayleigh and Mie winds also coexist, however, the sky is flagged as clear, so we select the Rayleigh wind as the all-sky wind (bin 29, Fig. 3d). For bin 23 where a cloud is detected, with a Rayleigh wind but no Mie wind, we decide to report "no data" in the all-sky wind dataset instead of a Rayleigh wind (bin 23, Fig. 3d). In a similar way, for bin 32 which shows no Rayleigh wind but a Mie wind in clear sky conditions, we report "no data" instead of a Mie wind (bin 32, Fig. 3d). If a bin is flagged as fully attenuated but a wind was retrieved, it is discarded. By doing so, we ensure that the Rayleigh winds indeed correspond to clear sky situations and the Mie winds to cloudy sky situations, consistently with our cloud mask.

The following statistics illustrate how often Rayleigh and Mie winds coexist within and outside the cloud mask. During the period extending from June to August 2020, 83% of the 3 km x 480 m bins flagged as cloudy with our cloud detection contained both a Rayleigh and a Mie wind, while 10% of bins flagged as cloudy contained only a Mie wind and 7% only a Rayleigh wind (on the edge of clouds or at cloud tops). On the other hand, 92% of bins flagged as clear contained only a Rayleigh wind, 7% contained both a Mie and a Rayleigh wind and 1% contained a Mie wind only.

Figure 4a shows the 2D-PDF of pairs of colocated Mie winds and Rayleigh winds which coexist within the cloud mask. The distribution is located around the 1:1 line for the entire range of wind speed, and particularly between -50 and 50 ms$^{-1}$ (98.8% of the values). For Mie wind speed between -40 and 10 ms$^{-1}$, we systematically observe Mie winds up to 1 ms$^{-1}$ larger than the co-located Rayleigh winds (Fig. 4b). For wind speeds between 10 and 50 ms$^{-1}$, the systematic differences switch signs and Rayleigh winds are up to 1 ms$^{-1}$ larger than the co-located Mie winds (Fig. 4b). For most of the wind speed values encountered in the troposphere, pairs of co-located Mie and Rayleigh winds within the cloud mask agree well, with systematic differences below 1 ms$^{-1}$ (similar to the maximum bias of Rayleigh winds, Aeolus DISC, 2024). The large spread is essentially caused by the random error of Mie and Rayleigh winds. Therefore, given the finer spatial resolution, lower random and systematic errors of Mie winds, it is preferable to substitute Rayleigh winds by the Mie winds within the cloud mask, especially for the study of wind-cloud interactions.

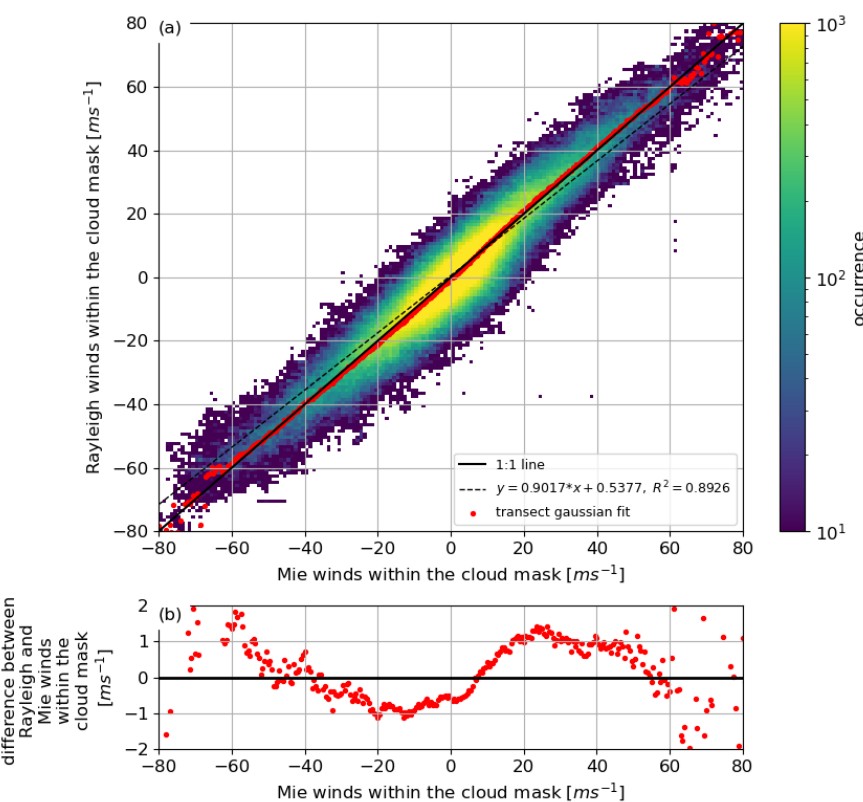

**Figure 4: (a) 2D-PDF of pairs of colocated Mie winds and the Rayleigh winds when they both coexist within the cloud mask. The black dotted line represents the best linear regression. The 1:1 line is represented as a solid black line. For each point along this 1:1 line, a Gaussian was fitted to all data points lying along a perpendicular transect. Where the data spread and statistics allow a satisfactory fit, the maximum of the Gaussian is plotted as a red filled circle each 0.5 ms$^{-1}$. (b) Maximum of the Gaussian of the differences between Rayleigh and Mie winds within the cloud mask as a function of the Mie winds within the cloud mask. A sample of 50 orbit files of the year 2020 are analysed with a total of $10^6$ bins of 3 km x 480 m where both Rayleigh and Mie winds coexist within the cloud mask.**

As this study is limited to the range 60°S – 60°N and as the laser pointing direction of Aeolus within this latitude range is quasi-eastward during ascending orbit and quasi-westward during descending orbit, (Fig. 1, Krisch et al., 2022), the HLOS wind observed by Aeolus, that is often noted $v_{HLOS}$ in the literature, is simply noted "$u$" all along the paper for simplicity. We adopt the convention that $u$ is negative (positive) when the wind is westward (eastward). We use $u_{cloud}$ ($u_{clear}$) to denote Mie (Rayleigh) winds within (outside of) the cloud mask while $u_{allsky}$ is the merging of both $u_{cloud}$ and $u_{clear}$. At this stage, the output files are individual orbit files with a cloud mask and Aeolus cloudy, clear and all-sky winds at 3 km along-track resolution and 480 m vertical resolution.

**3.2 About the sub-grid variability of wind at 3 km**

At a given altitude, each 87 km clear sky wind is horizontally resampled at 3 km (Fig. 3a, 3d). If clear sky winds were homogeneous over 87 km, this operation would lead to accurate winds at a resolution of 3 km, but it is a source of inaccuracies when the sub-grid variability (below 87 km) of clear-sky winds is large. To quantify this error, we use two independent datasets, from an aircraft and from a high resolution model.

First, we use profiles of wind acquired using an airborne 2µm Doppler Wind Lidar (DWL) operated by the German Aerospace Center (DLR) onboard a Falcon aircraft during the AVATAR-T validation campaign of Aeolus. This Lidar has a spatial resolution of 200 m horizontally and 100 meters vertically, extending between the surface and the aircraft which usually flies at about 10 km to 11 km of altitude. To retrieve the horizontal wind, an azimuth scan is applied and takes about 42 s, leading to a horizontal resolution of about 8 km (Witschas et al., 2017, 2022). The systematic error of horizontal wind measurements is estimated to be 0.1 ms$^{-1}$ and the random error about 1 ms$^{-1}$. A total of 8250 km was scanned by the aircraft near Cape Verde during the 5 flights we selected. A first selection is made to discard bins with an uncalibrated backscatter superior to 500 (that we estimated a good threshold to discriminate clear and cloudy sky). We first project the wind as if it was observed by Aeolus during its descending orbit and we average the wind vertically to a resolution similar to that of the Aeolus dataset, 480 m.

We then extract 11 adjacent 8 km x 480 m airborne DWL wind values within a curtain segment of 87 km horizontally x 480 m vertically to replicate Aeolus clear sky observations. A total of 94 independent segments were sampled with valid wind measurements in clear sky conditions. We calculate the standard deviation of the wind within each segment. A standard deviation equals to zero means that there is no horizontal "sub-grid" variability of the wind, and thus, the coarse resolution of Aeolus does not miss any sub-grid atmospheric circulation. The higher the standard deviation, the more sub-grid circulations are missed by Aeolus, making the re-sampling of the winds from 87 to 3 km questionable. We observed that in 97 % of the clear sky segments, the standard deviation of the wind within the segment is lower than 2 ms$^{-1}$ and in 80 % of the segments, it is lower than 1 ms$^{-1}$ (Fig. 5). This stresses out that when re-sampling the clear sky wind from 87 to 3 km, a sub-grid variability of about 1 ms$^{-1}$ is lost within the clear sky segment. Note that AVATAR-T wind observations are geographically limited around Cape Verde, but the wind encountered in this region are representative of a tropical marine trade winds regime (77 % of the Tropics, 40 % of the global surface) throughout most of the year (Bernardino et al., 2018).

To get even closer to the actual re-sampling resolution, we used a WRF simulation (more details about this simulation can be found in Ban et al., 2021) over Europe with a horizontal resolution of 3 km in clear sky. The domain is about 1200 km by 1500 km wide and tilted westward by about 8 degrees. This configuration means that the "latitude" axis of the domain is

aligned with typical ascending orbit tracks of Aeolus. This allows us to repeat the procedure described above for the 2µm DWL with the WRF simulation. A total of 375 independent segments of 87 km horizontally x 480 m vertically were sampled with clear sky only. We limited the domain vertically to 10 km to stay consistent with the airborne observations. The sub-grid variability of the horizontal wind is found to be similar to that observed by the DWL with 90 % of the segments having a sub-grid variability of less than 2 ms$^{-1}$ and 75 % of the segments less than 1 ms$^{-1}$. When extending the analysis of the WRF simulation up to 19 km (similar to the maximum altitude reached by Aeolus in our dataset, Fig. B3), the variability of the horizontal wind is even less, with 99 % of the segments having a sub-grid variability inferior to 2 ms$^{-1}$ and 93 % of the segments less than 1 ms$^{-1}$. The alpine region sets a high bound for horizontal wind variability as it is influenced by a large amount of gravity waves which induce perturbations of the horizontal wind of 1-2 ms$^{-1}$ on scales of 20-60 km (Hierro et al., 2018). With WRF the conclusions are the same that with the airborne DWL (which is noisy but coarser). Overall, a natural sub-grid variability of about 1 ms$^{-1}$ is missed by the coarse resolution of Aeolus, making atmospheric circulations of horizontal scale smaller than 87 km and with winds less than 1 ms$^{-1}$ in clear sky segments non observable. However, it is possible to study circulations at a 3-15 km horizontal scale in cloudy conditions and to compare them to the spatially closest clear sky observations, provided that we increase the uncertainty measurement in clear sky conditions by 1 ms$^{-1}$ to take account of the non-observed sub-grid variability. Section 4.3 is dedicated to such comparisons at cloud scale.

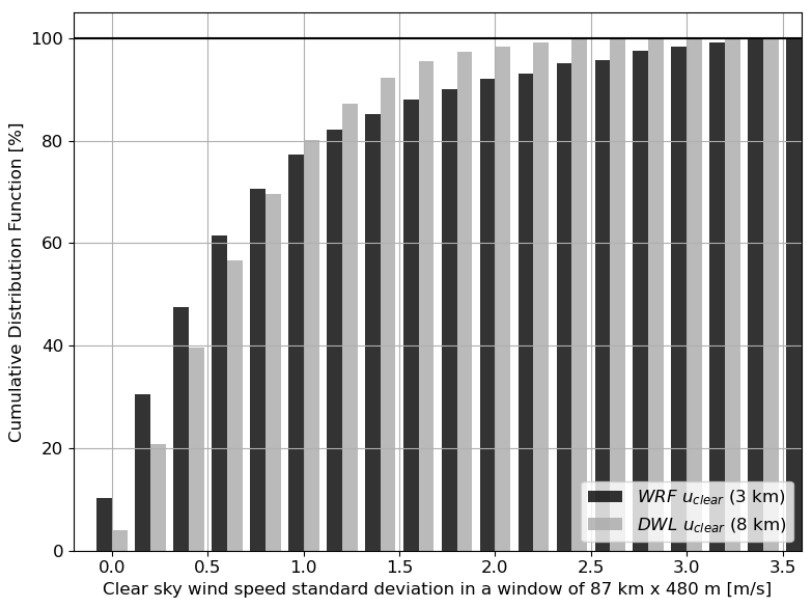

**Figure 5: Standard deviation of the wind speed within a segment of 87 km as observed by the airborne DWL with a horizontal resolution of 8 km and from a WRF model scene with a horizontal resolution of 3 km. DWL and WRF wind profiles are averaged vertically to a resolution of 480 m, similar to our Aeolus dataset. All altitudes between the surface and 10 km are included.**

### 3.3 Case study of the tropical cyclone Paulette observed with Aeolus

430

Figure 6 illustrates how Aeolus resampled cloud mask and winds allow us to observe from space different features ranging from cyclones to cumulus clouds. During its lifetime, Aeolus observed multiple cyclones, sometimes crossing them near their centre (Marinescu et al., 2022). Figure 6a shows an example of intersection between Aeolus and the tropical cyclone Paulette over the Atlantic Ocean during the hurricane season, on 12 September 2020. The wind and cloud curtains are displayed between 20°N and 40°N (Fig. 6b-g). Note that Aeolus covers this distance in about 4 minutes, so the curtains represent a snapshot of the scene. The cyclone is identified by the continuous high cloud cover between 26°N and 32°N at about 12 km of altitude (Fig. 6b). The laser typically only penetrates 1 to 2 km below the uppermost cloudy layer of the cyclone. This particular case study is also interesting as it encounters a diversity of clouds. We observe a cirrus cloud, northward of the cyclone, extending from 33°N to 34°N and between 12 and 15 km of altitude. Along half of its length, this cirrus does not fully attenuate the laser as some clear sky layers can be retrieved below its base. We also observe shallow cumulus clouds (Fig. 6a, 6b) between 20°N and 26°N, with their tops below 3 km of altitude and sometimes only occupying a single profile, surrounded by clear sky profiles. This stresses out the importance of performing cloud detection at full horizontal resolution of 3 km. Aeolus retrieves Rayleigh winds above and around the cyclone, up to 18 km of altitude (Fig. 6c). As the horizontal resolution of Rayleigh winds is fixed to 87 km, and molecular signal is still retrieved within clouds, some Rayleigh winds can be retrieved within clouds. For example, there are Rayleigh winds within the upper cloudy layers of the cyclone and in the entire boundary layer, even within shallow cumulus clouds (Fig. 6c). However, we only keep Rayleigh wind values outside of the cloud mask when building $u_{clear}$ (Fig. 6d). The cross section of $u_{clear}$ (Fig. 6d) reveals the wind shear found where counter-clockwise winds around the cyclone base meet the clockwise winds at the top of the cyclone. This happens at about 8 km of altitude at 25°N and at 35°N. The further we look from the cyclone, the higher in altitude the reversal of the wind occurs. Figure 6e shows the Mie winds retrieved by Aeolus. Most Mie winds are retrieved within the cloud mask. As the native resolution of Mie winds can be as coarse as 15 km, it is possible that Mie winds extend horizontally beyond the cloud mask as shown around shallow cumulus clouds, between 20°N and 26°N, below 3 km of altitude (Fig. 6e). $u_{cloud}$ (Fig. 6f) contains only Mie wind values within the cloud mask (as detailed in Sect 3.1). The merging of $u_{clear}$ and $u_{cloud}$ constitutes the all-sky wind, $u_{allsky}$ (Fig. 6f).

455

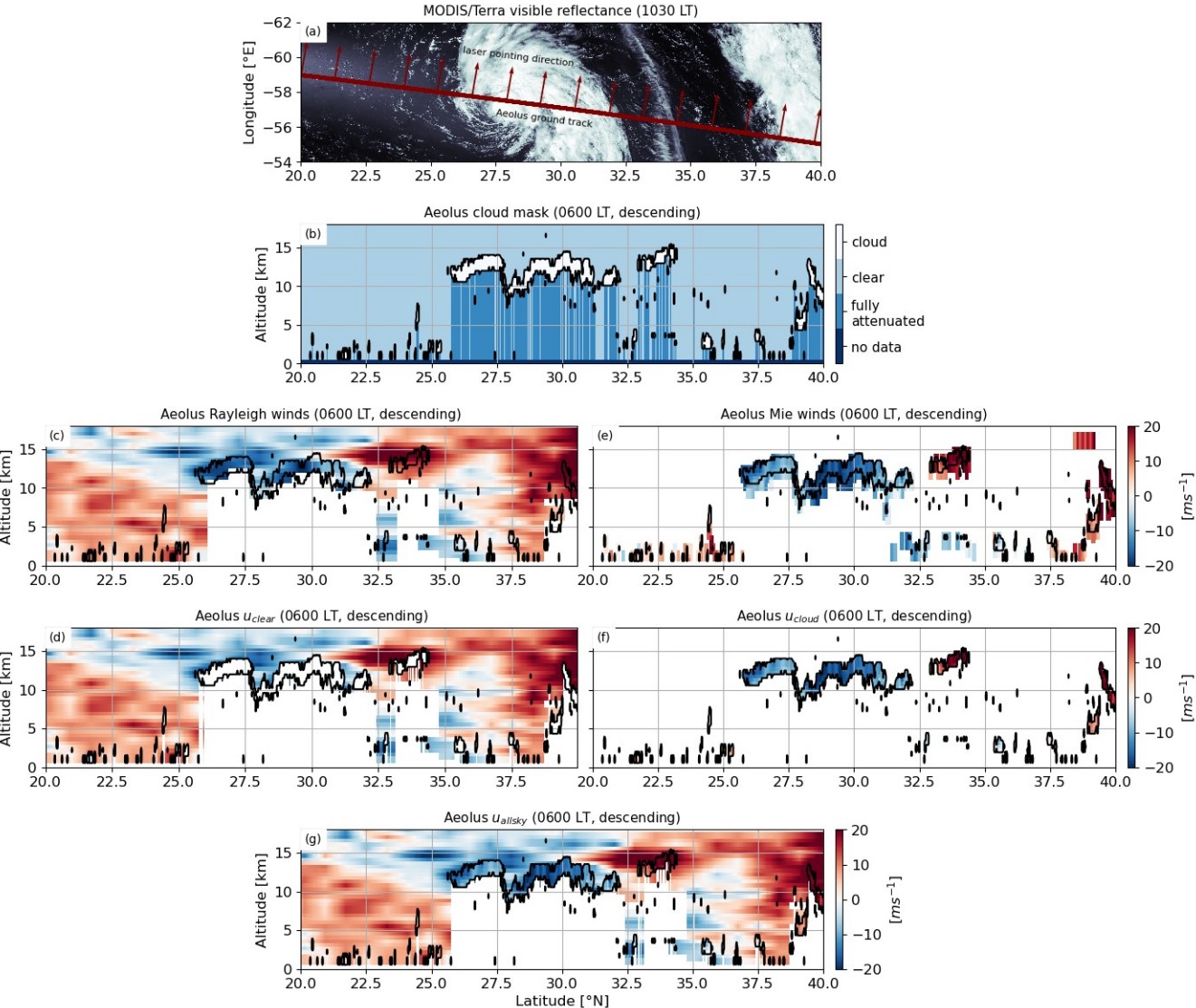

**Figure 6: (a)** Descending orbit segment crossing the tropical cyclone Paulette over the Atlantic ocean (2020-09-12T09–2020-09-12T11) plotted in red over a MODIS/Terra reflectance image. The red arrows represent the laser pointing direction. **(b)** Aeolus cloud mask. Aeolus **(c)** all Rayleigh winds and **(d)** Rayleigh winds only outside of the cloud mask, noted $u_{clear}$ along the paper. Aeolus **(e)** all Mie winds, **(f)** Mie winds only within the cloud mask, noted $u_{cloud}$ along the paper. **(g)** All-sky winds, noted $u_{allsky}$, result from the merging of $u_{clear}$ and $u_{cloud}$. The winds are negative when blowing westward and positive when blowing eastward. For panels (b-g), the resolution of the re-sampled data is 3 km horizontally and 480 m vertically and the black contour is the cloud mask.

## 4 Results at different scales

In this section, we illustrate analyses with our observations through three examples at different spatial scales: large, regional and cloud scale inferior to 100 km.

### 4.1 Global-scale circulations observed with Aeolus

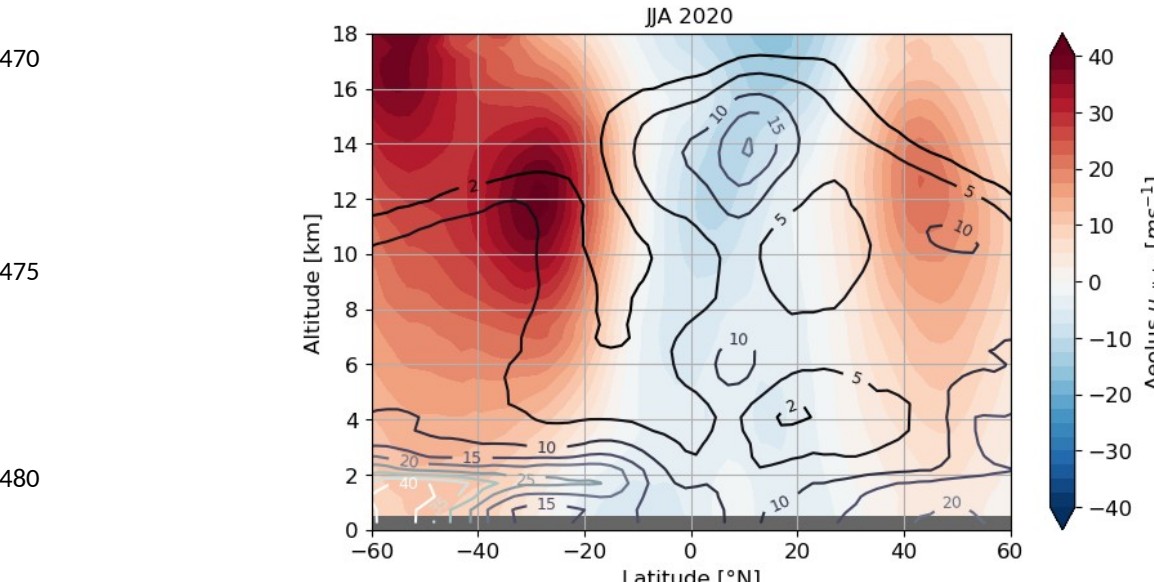

**Figure 7: (a) Zonal average of all-sky wind speed profiles from June to August 2020. Contours represent the zonal average cloud fraction.**

Aeolus observes the main features of the general circulation, like for example the trade winds below 2 km of altitude between 20°S and the equator and from 10°N to 20°N (Fig. 7) and the subtropical jet streams at 30°S and 40°N with their cores located at 12 km of altitude. As this zonal average of wind profiles is calculated from June to August 2020, the polar stratospheric jet is visible around 55°S and 17 km of altitude. Moreover, the speed of the subtropical jet stream in the southern hemisphere is larger than 35 ms$^{-1}$ due to a large meridional temperature gradient in the winter hemisphere through thermal wind balance, while the northern hemisphere subtropical jet stream only reaches 25 ms$^{-1}$.

495

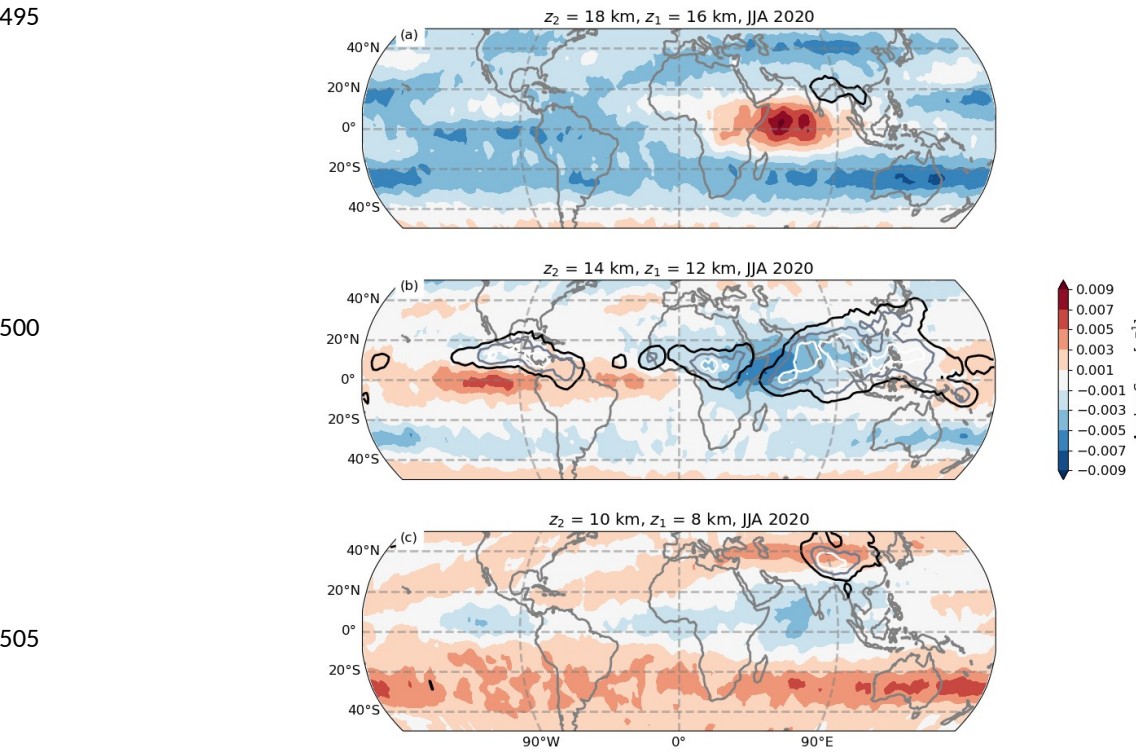

**Figure 8: Map of the median wind shear $S_{allsky}$ calculated a) between 8 km and 10 km, b) between 12 and 14 km and c) between 16 and 18 km from June to August 2020. Contours represent cloud covers of 5, 10, 15 and 20% for each altitude range.**

510 Figure 8 shows maps of the median wind shear $S_{allsky}$ computed between two layers separated by 2 km of altitude (larger than the native resolution of Aeolus) following Eq. 4:

$$S_{allsky} = \frac{u_{allsky}(z_2) - u_{allsky}(z_1)}{(z_2 - z_1)}$$  (4)

Note that a similar map, but with absolute wind shears is shown in Fig. B11.

Figure 8a focuses on the wind shear between 16 and 18 km of altitude, which corresponds to the tropical tropopause. $S_{allsky}$
515 mostly takes negative values within the tropics between $-3 \times 10^{-3}$ s$^{-1}$ and $-9 \times 10^{-3}$ s$^{-1}$ as the Quasi Biennial Oscillation is in a negative phase (fast westward winds in the stratosphere around $z_2$), but $S_{allsky}$ is positive and up to $10 \times 10^{-3}$ s$^{-1}$ over the Indian Ocean as the Tropical Easterly Jet (TEJ, Koteswaram, 1958) blows westward up to 40 ms$^{-1}$ at 16 km of altitude during the South-Asian Summer Monsoon (SASM). This region is also subject to deep convection, particularly over the Bay of Bengal (Zuidema, 2003 ; Hemanth Kumar et al., 2015), which then leads to cirrus clouds (Ali et al., 2022) distributed in the upper

half of the troposphere, up to 16 km of altitude. Figure 8a shows that indeed, a cloud cover of 5% is observed between 16 and 18 km of altitude, which corresponds to the highest clouds observed in this region (Ali et al., 2022). Jensen et al., (2025) demonstrated that wind shears of $10\times10^{-3}$ $s^{-1}$ were favourable for a faster sublimation of cirrus clouds particles, reducing the lifetimes of these clouds.

Between 12 and 14 km of altitude (Fig. 8b), the Northern hemisphere exhibits weak wind shear values between $-3\times10^{-3}$ and $3\times10^{-3}$ $s^{-1}$ except above the Indian Ocean where it reaches $-8\times10^{-3}$ $s^{-1}$ just under the core of the TEJ. This maximum of negative wind shear is located between central Africa and continental India, that both show a weaker negative wind shear, as these two regions prone to vigorous deep convection (up to 25% of cloud cover between 12 and 14 km) most likely experience a strong vertical mixing. At this altitude, another maximum of cloud cover is observed above central America, which also peaks at 20 % and results from continental deep convection.

In the southern hemisphere, a band of maximum negative wind shear of $-3\times10^{-3}$ $s^{-1}$ to $-9\times10^{-3}$ $s^{-1}$ is observed along the 30°S parallel. This maximum of negative wind shear is located just above the Sub-Tropical Jet (STJ) whose core's altitude is 12 km ($z_1$) and reaches up to 40 ms$^{-1}$ while the wind at 14 km does not exceed 30 ms$^{-1}$.

Between 8 and 10 km (Fig. 8c), the wind shear along the 30°S parallel is positive and reaches $5\times10^{-3}$ $s^{-1}$ as the core of the STJ is located just above. The strong wind shears induced by the STJ are not observed in the northern hemisphere as the meridional temperature gradients (which drive jet streams) are weaker than in the southern hemisphere from June to August. We observe nevertheless wind shears faster than $5\times10^{-3}$ $s^{-1}$ over continents in the northern hemisphere, extending from Turkey to coastal China. At the North bound of the map, at about 50°N, the tropopause layer is located between 9 and 10 km of altitude at the end of boreal summer (Schäfler et al., 2020). We observe a wind shear of about $1\times10^{-3}$ $s^{-1}$ around the globe at this latitude (Fig. 8c). Indeed, the wind profile is tilted eastward below the tropopause and westward above, which explains the weak positive wind shear around the tropopause. Schäfler et al., (2020) reported a weak, but negative wind shear of $-1\times10^{-3}$ $s^{-1}$ between 8 and 10 km for the month of October at about 60°N. The change of sign might be explained by a lower altitude tropopause at 60°N, 10 degrees northward of our observations, and thus by a larger contribution of the westward tilted profile above the tropopause.

## 4.2 Seasonal changes of clouds and winds over the Indian Ocean

During the months of June-August, we observe a maximum of high cloud fraction over the Indian Ocean (Fig. 9e), as well as the Tropical Easterly Jet (TEJ) that extends from the Tibetan plateau to the Western coast of Africa (Fig. 9f). The core of the TEJ, located at about 16 km of altitude, is visible on the map (Fig. 9c) with westward winds exceeding 30 ms$^{-1}$ over the Arabian Sea from June to August 2020 (consistent with Liu et al., 2024). We focused on a small domain located over the Indian Ocean on the Western coast of India and under the influence of the TEJ during summer (black rectangle, Fig. 9a). The domain extent of 20° of longitude ensures that Aeolus crosses it at least twice a day, once ascending and once descending. On the time series of wind profiles (Fig. 9f), westward (negative) winds appear in late May and reach values of above 30 ms$^{-1}$ within a few days only. The jet persists until the last week of October before decaying rapidly, giving way to eastward winds (positive values) again. It is worth noting that during the same period of the TEJ, Aeolus captures persisting eastward (positive) winds below 2 to 5 km of altitude associated to the monsoon circulation. Figure 9e displays the daily average profiles of the cloud fraction observed by Aeolus over the same domain. The low cloud fraction is persistent during the entire year and above 30 % during early winter over the cold Arabian sea and during late September. We notice two minima of low cloud fraction which correspond to the reversal of boundary layer winds, in April from westward to eastward dominant winds, and in October from eastward back to westward dominant winds (more visible in Fig. B4). Note that during the weeks preceding and following the reversal of the boundary layer winds in April, low clouds are confined between the surface and 1 km of altitude, while during periods of continuous westward winds (January to March 2020), low clouds typically extend up to 3 km. Part of this seasonal cycle of cloud top height is explained by a cooler Sea Surface Temperature (SST) from January to February, favorable for higher cloud tops (Höjgård-Olsen et al., 2022), and a warmer SST afterwards. Moreover, during the reversal of the winds, the evaporation flux at the surface the Indian Ocean is reduced, resulting in a shallower and dryer boundary layer, less favorable for the formation of low clouds (Mieslinger et al., 2019, Nuijens and Stevens, 2012). We also observe mid-level and high clouds above 5 km and up to 16 km, preferentially occurring between June and October, i.e. when the TEJ is the most active. As this period also corresponds to the South-Asian Summer Monsoon (SASM), the presence of high convective clouds is not surprising. The concurrence of a strong TEJ and monsoon clouds and rainfall was already mentioned by e.g. Koteswaram, 1958. However, deep convective cloud cover accounts for only 9% of the area of the Indian Ocean in July (Massie et al. 2002), and they are mainly located over the Bay of Bengal (Zuidema, 2003), thus the high cloud fraction seen in Fig. 9e between June and October must be essentially made of cirrus, as 90% of them are located outside of regions of deep convection (Massie et al. 2002). The large increase of the cirrus cloud fraction during the period where the TEJ is the most vigorous is thus thought to be favored by horizontal transport of moist air originating from convective towers over long distances (Das et al. 2011). However, below and above the core of the TEJ, we observe wind shears larger than 10$^{-2}$ s$^{-1}$, which were found to alter cirrus clouds structure and reduce their lifetime (Jensen et

al., 2025). Note that the time of ascending orbits of Aeolus (1800 LT) corresponds to a maximum of deep convection over the Bay of Bengal (Zuidema, 2003), while descending orbits (0600 LT) occur before the dissipation of cirrus clouds (Ali et al., 2022). Aeolus observations can thus be of a great help to better understand these interactions between horizontal winds and cirrus clouds.

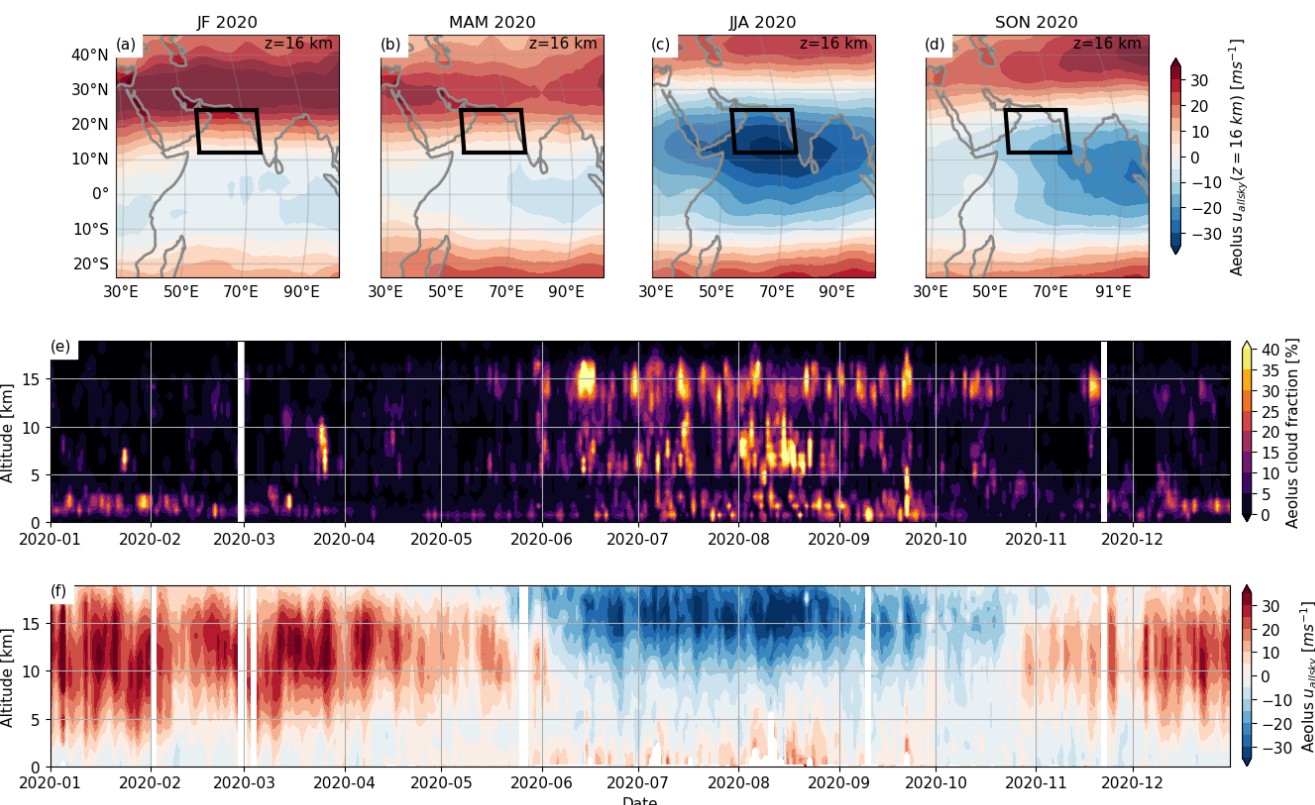

**Figure 9: (a) map of the averaged all-sky wind speed u$_{allsky}$(z=16 km) during the months of January and February 2020; (b, c and d) are the same for respectively March-May, June-August and September-November 2020. The rectangle (12°N-24°N, 55°E-75°E) represents a domain selected as it is under the influence of the Tropical Easterly Jet during summer. Time series at a daily resolution of (e) the average profiles of cloud fraction within the domain and (f) the average profiles of all-sky wind speed within the domain.**

## 4.3 Cloud scale circulations inferior to 100 km

When convection is triggered, the horizontal wind from the sub cloud layer is transported vertically within the cloudy layer, leading to different horizontal wind speeds within the cloudy layer and the surrounding environment at a given vertical level. This phenomenon referred as "Convective Momentum Transport" has been studied using Large Eddy Simulations (LES, Siebesma et al., 2003), and more recently using airborne measurements (Koning et al., 2022). In this subsection we investigate the ability of Aeolus to significantly retrieve different wind speeds within a cloud and in the clear sky surrounding the cloud.

### 4.3.1 Is it possible to capture the differences between wind speeds within clouds and their surrounding clear sky at the resolution of Aeolus?

To test the feasibility of significantly observing different wind speeds within clouds and their surroundings with Aeolus, we use DWL data measured with a Falcon flight during the AVATAR-T campaign. Figure 10 displays an example of cumulus clouds forming near Cape Verde and overflown by the DLR Falcon. We first project the zonal and meridional components of the wind as if Aeolus was observing the scene during its descending orbit. This results in profiles of $u_{DWL, allsky}$ at 100 m vertical resolution and 8 km horizontal resolution (Fig. 10a). We then coarsen the uncalibrated backscatter at the native horizontal and vertical resolutions of Aeolus before interpolating the signal at 3 km horizontal x 480 m vertical resolution, to create a cloud mask (Fig. 10b), consistent with our dataset. In the same way, we coarsen the profiles of $u_{DWL, allsky}$ at the native resolution of Aeolus, before interpolation on the 3km x 480 m grid. We further average the $u_{DWL, allsky}$ profiles encompassing the centre of the cloud between 1280 and 1320 km along flight (Fig. 10c, red curve). Similarly, we average the clear sky wind profiles on the left edge of the cloud, between 1210 and 1275 km along flight, to simulate a portion of clear sky wind observed by Aeolus (Fig. 10c, black curve).

We see that between the surface and the Cloud Base Height (CBH) of 960 m, the values of $u_{DWL\_allsky}$ are quite similar for a measurement performed just below the cloud base, or in the clear sky surrounding the cloud. In contrast, above 960 m, the clear sky wind is tilted westward and reaches -5 ms$^{-1}$ at 1.6 km of altitude (corresponding to the Cloud Top Height - CTH), while within the cloudy layer, as air masses from the surface are carried upward within the cloud, the wind only reaches -3 ms$^{-1}$ at the CTH. At this altitude, the difference between the clear sky wind and cloudy sky winds is the largest and reaches 2 ms$^{-1}$. Note that Koning et al., (2022) observed horizontal wind differences of up to 5 ms$^{-1}$ around convective updrafts using airborne wind observations. Between the CBH and the CTH, the wind shear within the cloud is about -5×10$^{-4}$ s$^{-1}$. The wind shear is larger (in absolute value) in the clear sky surrounding the cloud at the same altitude and is approximately equal to -3.7×10$^{-3}$ s$^{-1}$. Above the cloud top, cloudy and clear sky wind profiles join again at 2.1 km of altitude. Between the cloud top and 2.1 km, as the wind just above the cloud top experiences drag from the cloud top, the wind shear is negative and

maximum in absolute value just above the cloud top, reaching -4.6×10⁻³ s⁻¹, while the wind profile is not sheared in the clear sky surrounding above the CTH.

The case study presented in Figure 10 shows that when averaging the airborne DWL wind data to the coarser horizontal and vertical resolutions of Aeolus, it remains possible to capture significantly different wind speeds within clouds and within their surroundings in shallow convection. This finding encourages us to observe the impact of convective motions on the horizontal wind speed with the Aeolus dataset. However, because of Aeolus wind observations having a larger random error than the airborne DWL wind observations, an averaging of multiple Aeolus wind profiles is necessary to observe significant wind speed differences between the cloud and its surrounding clear sky.

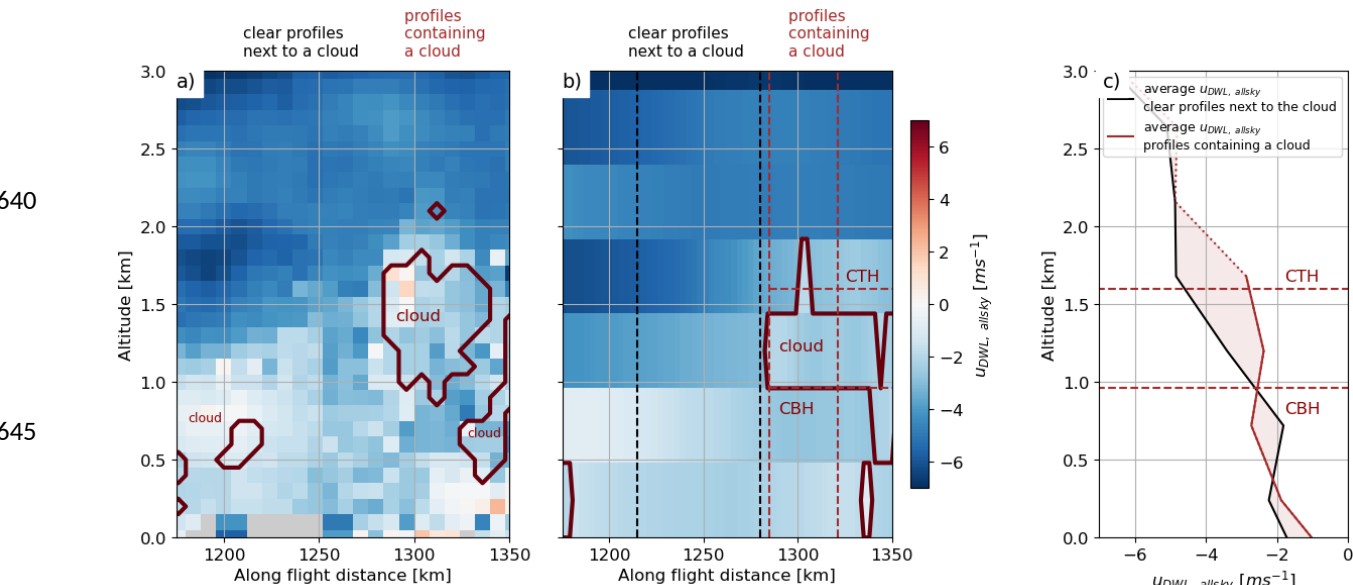

**Figure 10: (a) Curtain of $u_{DWL,\,allsky}$ acquired by the airborne DWL and projected along the laser pointing direction of Aeolus during the AVATAR-T campaign. (b) is the same $u_{DWL,\,allsky}$ wind curtain coarsened at the resolution of our Aeolus dataset (3 km along flight direction, 480 m vertically). The cloud mask is also coarsened and clouds are represented with solid red contours. Vertical red dashed lines represent the horizontal extent of the profiles encompassing a cloud while vertical black dashed lines represent the horizontal extent of clear sky profiles next to the clouds. Horizontal red dashed lines mark the average Cloud Base Height (CBH) and Cloud Top Height (CTH) for the profiles encompassing a cloud. (c) Profiles of $u_{DWL,\,allsky}$ encompassing the cloud (red curve) and in the clear sky surrounding the cloud (black curve).**

### 4.3.2 Differences between wind speed within clouds and their surrounding clear sky with Aeolus

We split the study within regions (Fig. 11a) exhibiting different types of clouds and different large-scale circulations and we focus on the entire year 2020. The first region is dominated by Stratocumulus decks sometimes transitioning to Cumulus clouds (TrSc). They are prevalent in the eastern subtropical oceans (Wood, 2012) and are capped by a strong inversion, usually created by the large scale subsidence associated with the descending branch of the Hadley-Walker circulation. The inversion is characterised by a sharp transition in most meteorological variables (Wang et al., 2008; Hourdin et al., 2019). The second region is dominated by Cumulus clouds (Cu) (McCoy et al., 2017 ; Qu et al., 2015). Cumulus clouds are usually found in the western subtropical oceans and are associated with a deeper boundary layer compared to the Stratocumulus region (Scott et al., 2010). These two regions are found above oceans and usually under the subtropical jet streams. The Indian Ocean is the third region, thermodynamically more unstable and prone to deep convection, it is also crossed by the Tropical Easterly Jet during boreal summer. The boundaries of this region are adapted from the INDian Ocean EXperiment (INDOEX, Mitra, 2004) in order not to overlap other boxes. Finally we choose a fourth region over the Pacific Ocean and between the latitudes of 10°S and the equator, referred below as the Pacific warm pool region (WP) and characterised by SST up to 32°C (Jauregui & Chen, 2023), favourable for deep convection.

Figures 11b and 11c show the average wind speed and wind shear profiles for each region over the year 2020. While Houchi et al. (2010) performed a climatology of atmospheric horizontal wind and wind shear, they did not examine the typical wind profiles for different cloud regimes. On the other hand, some observational studies (Tian et al. 2021; Savazzi et al. 2022) analysed wind shears in different convective regimes, but their observations were concentrated in particular regions. Here, our results show at a global scale the wind profiles associated to different cloud regimes. The TrSc and Cu wind profiles exhibit an eastward tilt from the surface to the core of the subtropical jet stream at about 12 km of altitude (Fig. 11b). The TrSc and Cu winds are both negative near the surface and change sign in the lower troposphere, at 3 km and 5 km for the TrSc and Cu regions, respectively (Fig. 11b). This is consistent with Helfer et al. (2020), who mention that the winds in trade-wind cumulus regions become increasingly eastward with height. The average wind profile observed over the Cu region is in good qualitative agreement with the averaged wind profile observed between the surface and 5 km of altitude during EUREC4A field campaign in January and February 2020, that targeted a shallow cumulus dominated area around Barbados (Savazzi et al., 2022). Within the altitude range of 1-12 km, the average wind shears in the TrSc and Cu regions are positive (Fig. 11b). Helfer et al. (2020) note that in the Cu region, the vertical shear in the zonal wind component is to first order set by large-scale meridional temperature gradients through the thermal wind relation, and therefore $\partial_z u > 0$ is typical for most of the year. The TrSc region exhibits a stronger wind shear than the Cu region in the lower troposphere, which is consistent with the sharp transition of meteorological variables noted there by Hourdin et al. (2019). Over the

Indian Ocean, we observe eastward (positive) winds between the surface and 14 km of altitude. Above 14 km of altitude, the wind speeds change direction and become westward. Below 10 km, the wind shear is positive but weak (lower than $10^{-3}$ $s^{-1}$). Above 10 km, the observed wind shears become negative, reaching $-1.8\times10^{-3}$ $s^{-1}$ at 13 km of altitude because of the presence of the Tropical Easterly Jet just above (as also depicted in Fig. 8b). Above the warm pool, the wind is westward in the whole troposphere and the wind shears are weak, between $-5\times10^{-4}$ and $5\times10^{-4}$ $s^{-1}$ throughout the entire profile. Weak wind shears are explained by the presence of a strong convection in this region of high SST (Hibbert et al. 2023). Our result is consistent with the findings of Tian et al. (2021), who note that the wind shear in the mid-troposphere in a deep convective regime is significantly weaker than in situations dominated by lower convective clouds (below 7 km).

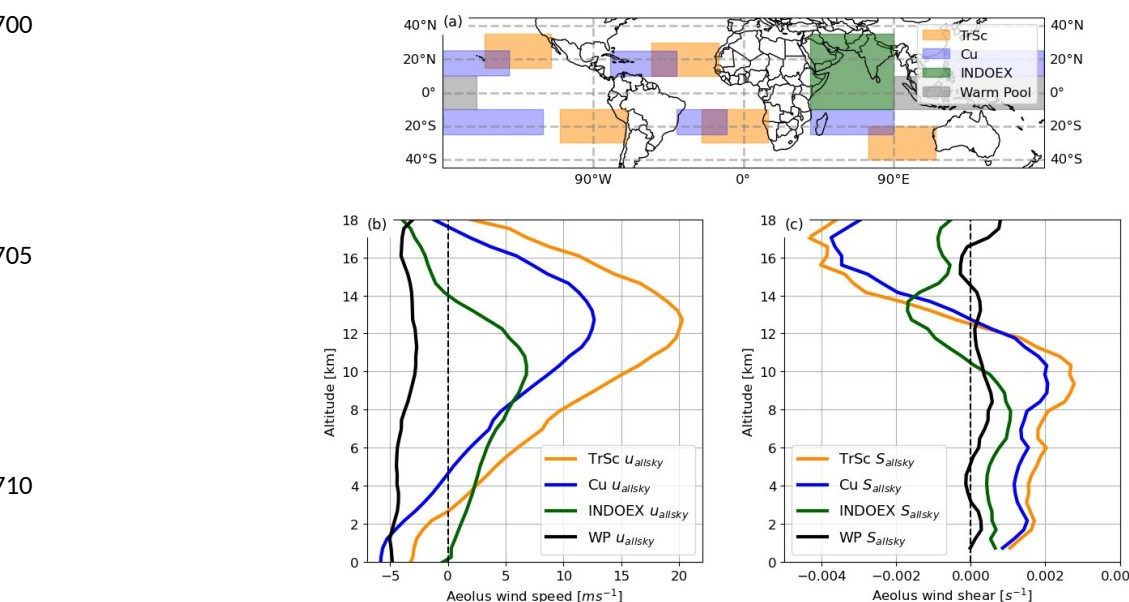

**Figure 11: (a) map of the different regions considered. Average (b) wind speed profiles and (c) wind shear profiles retrieved in each region during the year 2020. The wind shear $S_{allsky}(z)$ is computed at each altitude z using the $u_{allsky}(z_1)$ observed 1 km below z and at $u_{allsky}(z_2)$ observed 1 km above z, following Eq. 4.**

A sample of profiles is selected to study the differences between the wind speeds within clouds and their surrounding clear sky. We identify for each region profiles containing adjacent cloudy layers in the vertical direction and that are at least 2 km thick in total, typically associated to convective clouds. We record the cloudy wind speed observed in the uppermost cloudy layer (noted $u_{cloud\_up}$) as well as its altitude, and the cloudy wind speed 2 km below the uppermost cloudy layer (noted $u_{cloud\_down}$). We compute the wind shear between these two layers distant of 2 km (noted $S_{cloud}$). More details are given in Fig.

B9. In the same orbit, among the surrounding profiles, we look for the closest profile located at a distance shorter than 100 km that exhibits clear sky winds everywhere in this 2 km thick layer. We record the clear sky wind within this profile (noted $u_{clear\_surrounding\_cloud\_up}$) at the same altitude as the uppermost cloudy layer. We also record the clear sky wind speed (noted $u_{clear\_surrounding\_cloud\_down}$) 2 km under $u_{clear\_surrounding\_cloud\_up}$, coming from the same profile. We compute the clear sky wind shear in the surrounding of the cloud (noted $S_{clear\_surrounding\_cloud}$). Each cloud and its environment are therefore associated to a group of six variables including four wind speeds and two wind shears. A two-sided T-test is applied to test the significance of the differences in Fig. 12 and is passed at each altitude level if the p-value < 0.05.

730

Figure 12a shows the average profile of $u_{cloud\_up}(z)$ and the average profile of $u_{clear\_surrounding\_cloud\_up}(z)$ for the different regions. Because of the smaller vertical extent of low clouds, and particularly boundary layer clouds (Wood et al., 2012; Cesana et al., 2019) which typically do not exceed 1 km, and because of the stronger attenuation of low liquid clouds (Guzman et al., 2017), there are no occurrences of Aeolus wind shears calculated over 2 km vertically below 5 km of altitude (Fig. B6). We find that for both TrSc and Cu, between 5 and 10 km of altitude, $u_{cloud\_up}(z)$ is eastward and 1 to 5 ms$^{-1}$ slower (statistically significant) than its paired $u_{clear\_surrounding\_cloud\_up}(z)$ at the same altitude. We also note that at these altitudes the average wind shear within the cloud $S_{cloud}(z)$ is always lower than $10^{-3}$ s$^{-1}$, while the one observed in the clear sky surrounding the cloud is positive and ranges between $10^{-3}$ and $3.5 \times 10^{-3}$ s$^{-1}$ (Fig. 12b). In contrast, at altitudes higher than 14 km where the clear sky wind shear is negative (Fig. 12b), $u_{cloud\_up}(z)$ is significantly more eastward than its paired $u_{clear\_surrounding\_cloud\_up}(z)$, and the differences range from 1 to 2 ms$^{-1}$ (Fig. 12a). We thus emphasise that over the stratocumulus and cumulus dominated regions, the wind shear within clouds is systematically smaller than the wind shear in the clear sky surrounding the clouds, and this difference is statistically significant. In addition, over these two regions, the differences between $u_{cloud\_up}(z)$ and $u_{clear\_surrounding\_cloud\_up}(z)$ can reach above 3 ms$^{-1}$, particularly at altitudes where the wind shear $S_{clear\_surrounding\_cloud}(z)$ is the largest. Note that over the INDOEX region, there are no altitudes where $u_{cloud\_up}(z)$ and $u_{clear\_surrounding\_cloud\_up}(z)$ are significantly different and where $S_{cloud}(z)$ and $S_{clear\_surrounding\_cloud}(z)$ are significantly different, hence this region does not appear in Fig. 12a and Fig. 12b. Over the Pacific warm pool, $u_{clear\_surrounding\_cloud\_up}(z)$ and $u_{cloud\_up}(z)$ are both westward between 14 and 16 km of altitude, which correspond to deep convective cloud tops in this region (Sassen et al., 2009). Within this altitude range, $u_{clear\_surrounding\_cloud\_up}(z)$ is 1 to 2 ms$^{-1}$ faster than $u_{cloud\_up}(z)$ (Fig. 12a). $S_{clear\_surrounding\_cloud}(z)$ ranges between $-1.5 \times 10^{-3}$ and $-2.5 \times 10^{-3}$ s$^{-1}$ (Fig. 12b). Tian et al., (2021) reported similar wind shears in the upper troposphere around deep convective clouds over the Amazon. $S_{clear\_surrounding\_cloud}(z)$ is $10^{-3}$ s$^{-1}$ larger (in absolute value) than $S_{cloud}(z)$ (Fig. 12b).

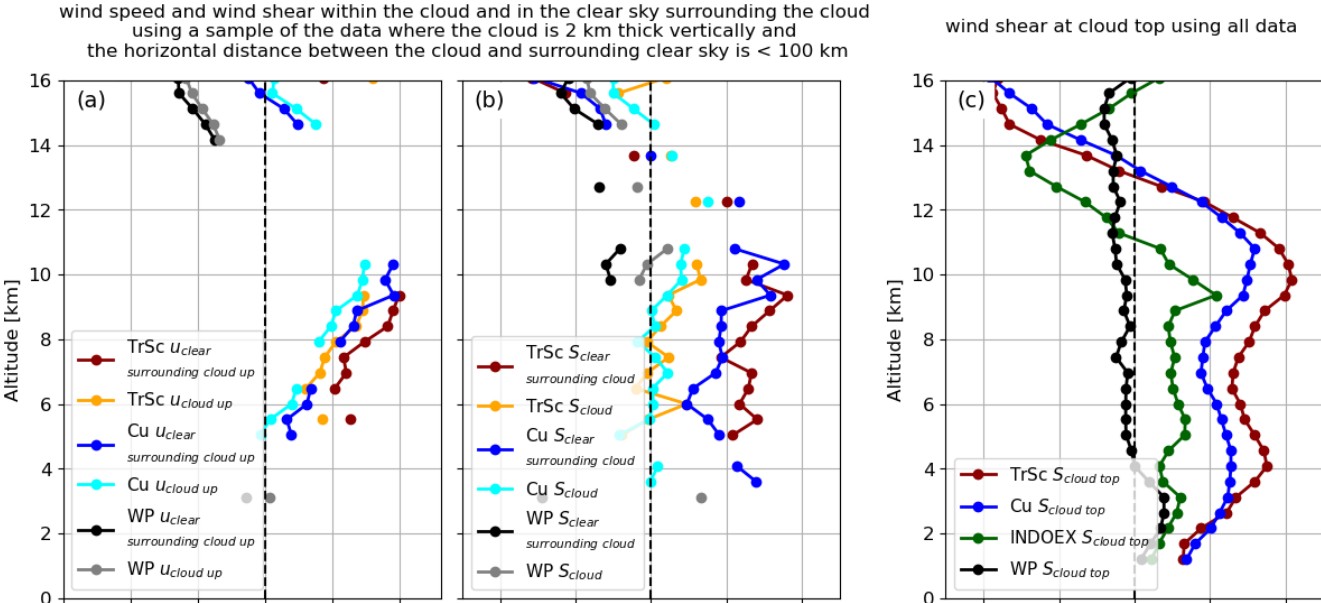

**Figure 12: (a)** average wind speed profiles retrieved within the uppermost cloudy layer $u_{cloud\ up}(z)$ and average of the closest clear sky wind speed $u_{clear\ surrounding\ cloud\ up}(z)$ observed over each region. Only values where $u_{cloud\ up}(z)$ and $u_{clear\ surrounding\ cloud\ up}(z)$ are significantly different (two sided T-test with p-value < 0.05) are plotted. **(b)** Average wind shear profiles within the cloud $S_{cloud}(z)$, and in the surrounding clear sky $S_{clear\ surrounding\ cloud}(z)$ are computed at each altitude z using the the wind speed observed at $z_1$ located 1 km below z and at the wind speed observed at $z_2$ located 1 km above z. Note that for (a) and (b), only a sample of the data is used as each cloud should be at least 2 km thick vertically, and the horizontal distance between $S_{cloud}(z)$ and $S_{clear\ surrounding\ cloud}(z)$ must be < 100 km and only values where $S_{cloud\ up}(z)$ and $S_{clear\ surrounding\ cloud\ up}(z)$ are significantly different (two sided T-test with p-value < 0.05) are plotted. **(c)** Average cloud top wind shear $S_{cloud\ top}(z)$ profiles retrieved in each region during the year 2020 using all data collected by Aeolus over each region, contrarily to (a) and (b) that only use a sample of the data.

By retrieving the wind both within clouds and above cloud tops, this Aeolus dataset gives access to the wind shear at the top of the clouds (Fig. 12c). Note that to study the wind shear at the top of clouds (Fig. 12c), we analyse all the data collected over each region, contrarily to Fig. 12a and 12b. Within each region, we record the wind speed observed in the uppermost cloudy layer (noted $u_{cloud\_up}$) and the clear sky wind speed 2 km above (noted $u_{above\_cloud}$). We then compute $S_{cloud\_top}$, the wind shear between these two layers (see Fig. B9). Figure 12c shows the average profile of $S_{cloud\_top}(z)$ for the different regions. Within the lower troposphere, the largest number of cloud top wind shear observations is found between 2 and 3 km of altitude over the TrSc and Cu regions (Fig. B10), consistent with Wood (2012) and Cesana et al., (2019). At these altitudes, the average wind shear at cloud top $S_{cloud\_top}(z)$ ($2 \times 10^{-3}$ to $3 \times 10^{-3}$ s$^{-1}$, Fig. 12c), is larger than in all sky conditions $S_{allsky}(z)$ (about $1.5 \times 10^{-3}$ s$^{-1}$, Fig. 11c). This result is consistent with previous work stating that a temperature inversion above cloud

tops isolates the cloudy layer from the clear sky above. A zone of larger wind shear can thus develop around the temperature inversion (Wang et al., 2008; Hourdin et al., 2019), which can in turn affect the morphology of these clouds through entrainment and drying of the boundary layer (Schulz & Mellado, 2018; Zamora Zapata et al., 2021).

### 4.3.3 First validation of K-theory for wind in the free troposphere with Aeolus

We observe for the first time from space a systematic anti-correlation between the sign of the $u_{cloud\_up}(z) - u_{clear\_surrounding\_cloud\_up}(z)$ and the sign of the wind shear in the surroundings of clouds, $S_{clear\_surrounding\_cloud}(z)$, at each altitude. In fact, the differences $u_{cloud\_up}(z) - u_{clear\_surrounding\_cloud\_up}(z)$ show a quasi linear relationship (Fig. 13) with the values of $S_{clear\_surrounding\_cloud}(z)$ for the TrSc, Cu and the INDOEX regions with correlation coefficients of respectively R=-0.87, R=-0.94 and R=-0.8. Our results are very consistent with those obtained with a Cloud Resolving Model by Grubisic and Moncrieff (2000), who also show a weaker horizontal wind in the updraft (analogous to our cloudy sky) than in the downdraft region (analogous to the clear sky surrounding the cloud) in positive wind shear conditions, and a larger difference between updraft and downdraft winds when the wind shear increases. This result suggests that for strongly sheared regions, Aeolus, at its coarse resolution can observe significant differences between the winds in the uppermost layer of convective clouds and their clear sky surrounding, and these differences anti-correlate well with the wind shear in the surrounding of clouds. This is in line with the K-theory, that stipulates that the averaged wind perturbations in a turbulent fluid are proportional to the averaged wind shears. Over the Pacific warm pool and the INDOEX region, as expected, the wind profiles are overall less sheared than over TrSc and Cu regions, with differences between $u_{cloud\_up}(z)$ and $u_{clear\_surrounding\_cloud\_up}(z)$ below 2 ms$^{-1}$.

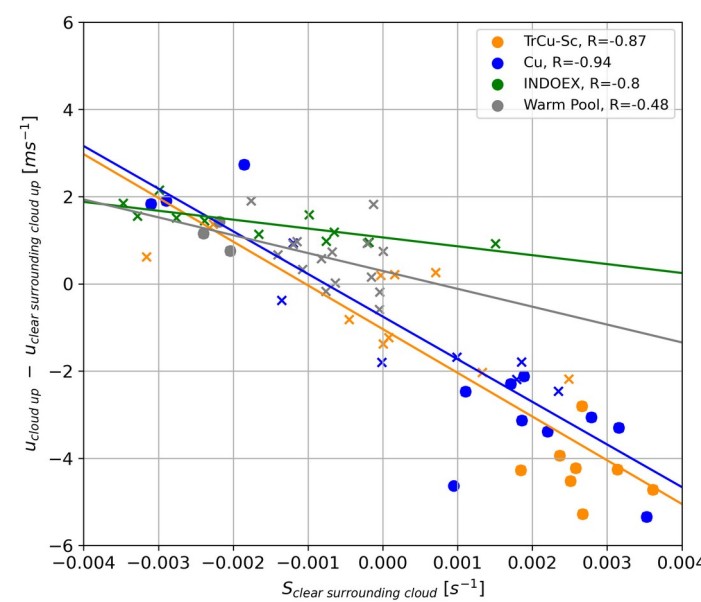

**Figure 13: Scatter plot of the wind shear in the clear sky surrounding the cloud $S_{clear\_surrounding\_cloud}(z)$ and the wind speed perturbation associated to the presence of a cloud, $u_{cloud}(z) - u_{clear\_surrounding\_cloud}(z)$ for each region. Each point represents an altitude level. We used circles to denote altitudes where $u_{cloud}(z)$ and $u_{clear\_surrounding\_cloud}(z)$ are significantly different (two sided T-test with p-values < 0.05) and crosses otherwise.**

## 5 Summary and future work

The observations presented in this paper display for the first time merged cloud vertical profiles and vertical profiles of horizontal wind at global-scale. We constructed cloud profiles at 3 km of horizontal resolution and re-sampled vertically at 480 m, using Aeolus L1A uncalibrated backscatter data coming from the detector of the Mie channel only. Corrections were applied to compensate for the varying vertical resolution and optical properties of the detector, the lack of the cross-polar backscattered signal as well as the increasing number of hot pixels during the mission. Globally, the obtained cloud fraction

profiles showed a good agreement with CALIPSO-GOCCP cloud profiles with an $R^2$ of 0.84, Pearson correlation of 0.92 and local cloud fraction differences below 2.5 % in most of the entire free troposphere. Using this cloud detection, we re-sampled the already calibrated and validated L2B Aeolus winds on a curtain of 3 km of horizontal resolution and 480 m of vertical resolution. We showed that perfectly colocated Rayleigh and Mie wind values agree well within the cloud mask with differences below 1 ms$^{-1}$. As Mie winds have a better spatial resolution, lower systematic and random errors than Rayleigh

winds, we substituted Rayleigh wind values by Mie wind values within the cloud mask. We also assessed that Aeolus re-sampled clear sky winds at 3 km are representative of the actual wind at 3 km of resolution, (with differences below 1 ms$^{-1}$ in 78% of the cases), based on airborne Doppler Wind Lidar data and a regional weather model simulation at high spatial resolution.

To highlight the potential of this dataset, we showed unique global, perfectly co-located, direct observations of cloud and wind profiles within the entire troposphere during boreal summer 2020. Unsurprisingly, the main zonal global-scale circulations are well captured by Aeolus. This includes the almost cloud-free subtropical and tropical jet streams as well as the tropical tropopause circulation. This opens perspectives of exploring deeper the shift in intensity and position of the subtropical jet stream induced by the cloud radiative effect, particularly in regions with a low number of in-situ observations.

Over the Indian Ocean, we observed low altitude cloud fractions of about 30 % in January that decrease until April 2020 and then increase again while the Monsoon onsets (June to September 2020). In the upper troposphere, when the Tropical Easterly Jet starts (early June), winds in its core quickly reach speeds of above 40 ms$^{-1}$ and high cloud fractions suddenly increase at the same time, exceeding 30 %.

Finally, regarding circulations at cloud scales inferior to 100 km, we analysed the averaged wind speed differences between

the uppermost layer of convective clouds and the surrounding clear sky. After confirming that these wind speed differences can be observed at the resolution of Aeolus observations (using an airborne case study averaged at the resolution of Aeolus), we split the study in regions having different large scale circulations. Over regions dominated by Stratocumulus and Cumulus clouds, convective motions induce large wind speed differences between the uppermost cloudy layer and their clear sky environment, exceeding 3 ms$^{-1}$ at some altitudes. We finally showed that these wind speed differences anti-correlate with

the wind shear in the clear sky surrounding the cloud. This anti-correlation is particularly strong with R=-0.94 over Cumulus and R=-0.87 over Stratocumulus dominated regions. This is a direct evidence that horizontal momentum transported by convective motions can be observed by Aeolus. We also found that the observed cloud top wind shear above Stratocumulus and Cumulus clouds ($2\times10^{-3}$ to $3\times10^{-3}$ s$^{-1}$) was larger than the observed all-sky wind shear at the same altitude ($1.5\times10^{-3}$ s$^{-1}$).

These few applications show the potential of this new observations for studying wind-cloud interactions at different horizontal scales, extending from 3 km to the global scale. In the near future we plan to focus on the correlation between cirrus covers and the strengthening of horizontal winds (Das et al., 2011), and on the interactions between the cloud radiative effect and jet stream shifting (Voigt et al., 2021). The case study on the tropical cyclone also opens perspectives to study how the wind shear contributes in the organization of shallow convection from random patterns to clusters (Mieslinger et al.,
2019 ; Bony et al., 2020) and sometimes mesoscale convective systems (Houze 2004 ; Schumacher and Rasmussen, 2020 ; Abramian et al., 2022).




## Appendix A: Complements relative to cloud detection

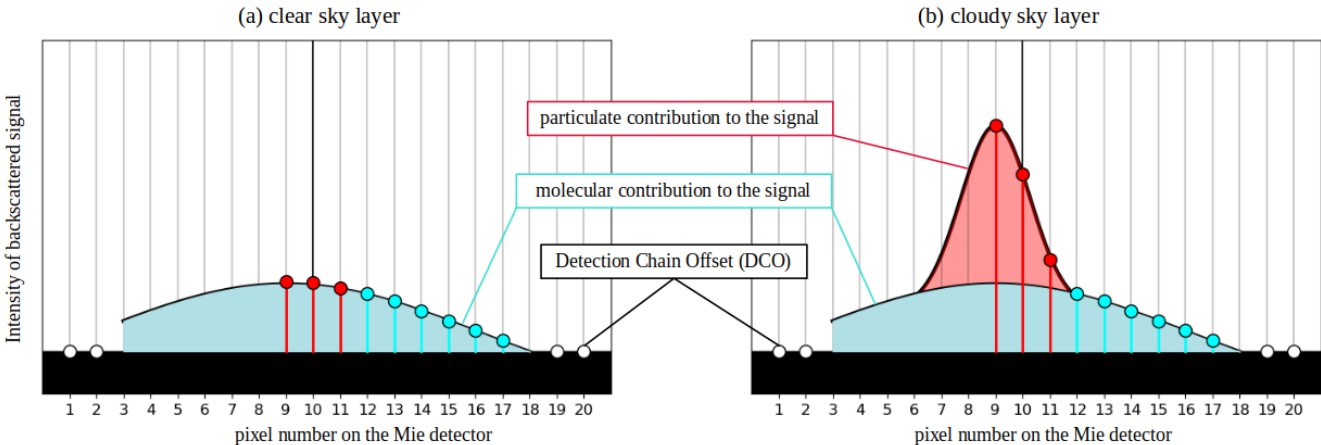

**Figure A1. Schematics of the signal received on the detector of the Mie channel (a) in the case of a clear sky layer and (b) in the case of a cloudy layer. Note that these are not at scale and in the case of a cloudy scene, the particulate backscattered signal peak is much larger compared to the molecular backscattered spectrum.**

On Fig. A1, we display the intensity of the backscattered signal retrieved on each of the 16 central pixels (pixels 3-16) of the

Mie channel detector in a single profile at one altitude level. Note that the pixels 1, 2, 19 and 20 only store information about the Detection Chain Offset (DCO), and the average value stored on these four pixels is averaged and subtracted to the backscattered signal. In the case of Aeolus, a fraction of the molecular backscattered signal is retrieved on the Mie detector and shown in blue. The intensity of this molecular signal essentially depends on the molecular density. The centre of the distribution, contains information about both the molecular and particulate backscattered signals. The red part is only due to

the presence of aerosols such as cloud droplets or ice particles which are much slower than individual molecules. Therefore, the intensity of the red peak at the center increases in the presence of a cloud as shown in Fig. A1b and is nonexistent or small in the absence of clouds as shown in Fig. A1a.

In the case of Fig. A1, the signal retrieved in the pixels corresponding to the peak of the backscattered signal is found on pixel 9 (left part of the Mie channel detector), therefore the value of the signal stored in pixel 9 and in the two neighboring

pixels on its right (pixels 10 and 11) are summed and noted $I_{part}(z_{L1A})$ in this paper, $z_{L1A}$ being the altitude of the center of a layer in a L1A profile. The signal retrieved in the six following pixels to the right of pixel 11 (pixels 12-17) are summed and correspond to the molecular backscatter, called $I_{mol}(z_{L1A})$. Although the molecular backscattered signal (in arbitrary unit) in this approach does not represent the actual molecular backscatter (in m$^{-1}$ sr$^{-1}$), it is proportional to it, enabling us to use the difference $I_{part}(z_{L1A})$ - $I_{mol}(z_{L1A})$ to determine the cloud mask.


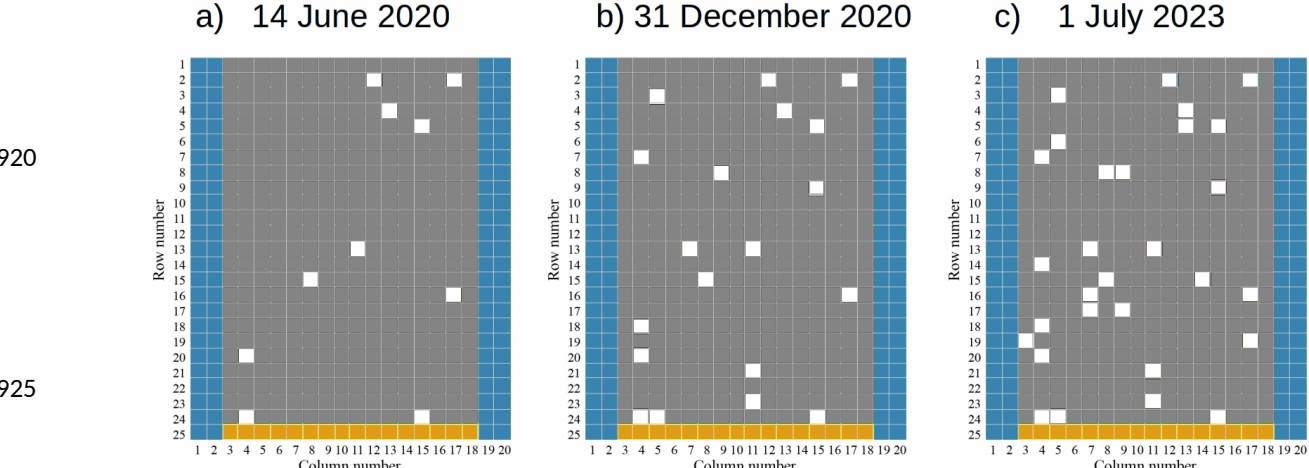

**Figure A2. Hot pixel maps corresponding to (a) 14 June 2020 (compare with Weiler et al., 2021), (b) 31 December 2020 (end of our study) and 1 July 2023 (end of Aeolus mission). Hot pixels are denoted in white, regular pixels in gray. Hot pixels are identified by comparing each detector pixel value against its immediate neighbors within the same Mie detector row: pixels exceeding all neighboring values are flagged and counted across all daily orbits. The resulting frequency maps are normalized by their maximum count and thresholded at 0.2, with pixels above this empirical threshold marked as "hot".**

For our study, we used the hot pixel map of 31 December 2020 (Fig. A2b) but follow-on study based on a longer period may use the most conservative hot pixel map of 1 July 2023 (Fig. A2c).

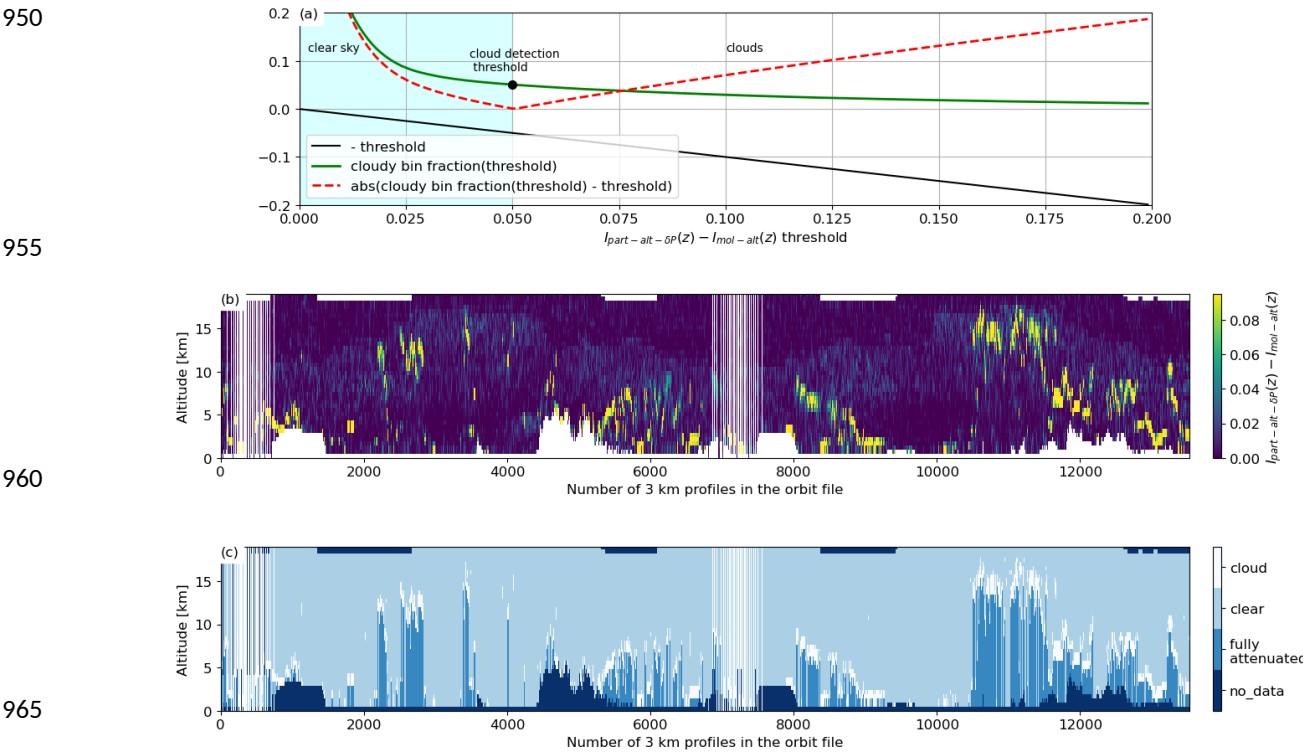

**Figure A3. (a) Fraction of cloudy bins retrieved depending on the applied cloud detection threshold (green curve) and cloud detection threshold (black dot) for the orbit 2020-09-12T09–2020-09-12T11 (same as Fig. 6). (b) Cross section of $I_{part\text{-}alt\text{-}\delta P}(z) - I_{mol\text{-}alt}(z)$ for the same orbit and (c) the resulting cloud mask.**

For each profile, a layer is declared cloudy when and $I_{part\text{-}alt\text{-}\delta P}(z) - I_{mol\text{-}alt}(z)$ exceeds a certain threshold. We found that from an orbit to the other, the distributions of particulate and molecular backscatters fluctuate. We suggest that a good way to find the cloud detection threshold on $I_{part\text{-}alt\text{-}\delta P}(z) - I_{mol\text{-}alt}(z)$ for each orbit, is to compute the fraction of cloudy bins obtained for various thresholds on $I_{part\text{-}alt\text{-}\delta P}(z) - I_{mol\text{-}alt}(z)$, ranging from 0.01 to 0.20 (arbitrary units). The fraction of cloudy bins decreases quickly for low values of $I_{part\text{-}alt\text{-}\delta P}(z) - I_{mol\text{-}alt}(z)$, which correspond to the clear sky and Poisson-distributed noise. It then decreases slowly, for larger particulate backscatters associated with clouds. The cloud detection threshold is found when abs(cloudy bin fraction(threshold) – threshold) is minimum. Note that the determination of the cloud detection threshold is evaluated while accumulating all altitudes between the surface and 18 km of altitude. Therefore, this threshold is strongly

weighted by the values of $I_{part-alt-\delta P}(z) - I_{mol-alt}(z)$ in the free troposphere, mostly free of Saharan dusts and aerosols, but containing a lot of clouds with larger $I_{part-alt-\delta P}(z) - I_{mol-alt}(z)$. Therefore, aerosol layers are classified as clear sky.

Figure A4 shows the map of low and mid-level clouds during the season where the most of Saharan dusts are observed. Overall, on the West coast of Africa at around 20° of latitude, the Aeolus cloud cover is approximately 10 % lower than
985 CALIPSO-GOCCP below 8 km of altitude. CALIPSO-GOCCP cloud detection threshold was already restrictive enough to flag Saharan dusts as clear sky. Therefore, Saharan dusts are even less likely to be flagged as clouds in our Aeolus dataset.

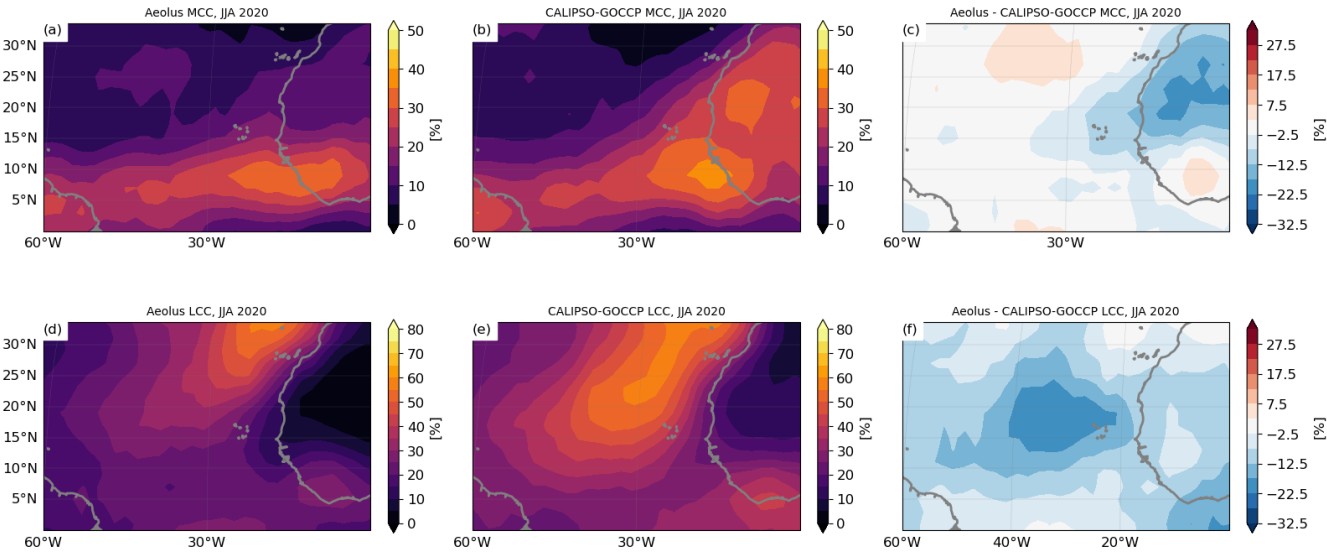

**Figure A4. Maps of mid-level (4-8 km) cloud cover (a) Aeolus dataset (b) CALIPSO-GOCCP and (c) the difference Aeolus –**
990 **CALIPSO-GOCCP for the year 2020. (d), (e) and (f) are the same but for low level clouds (0-4 km).**

 **Appendix B: Complements relative to the dynamic variables estimates**

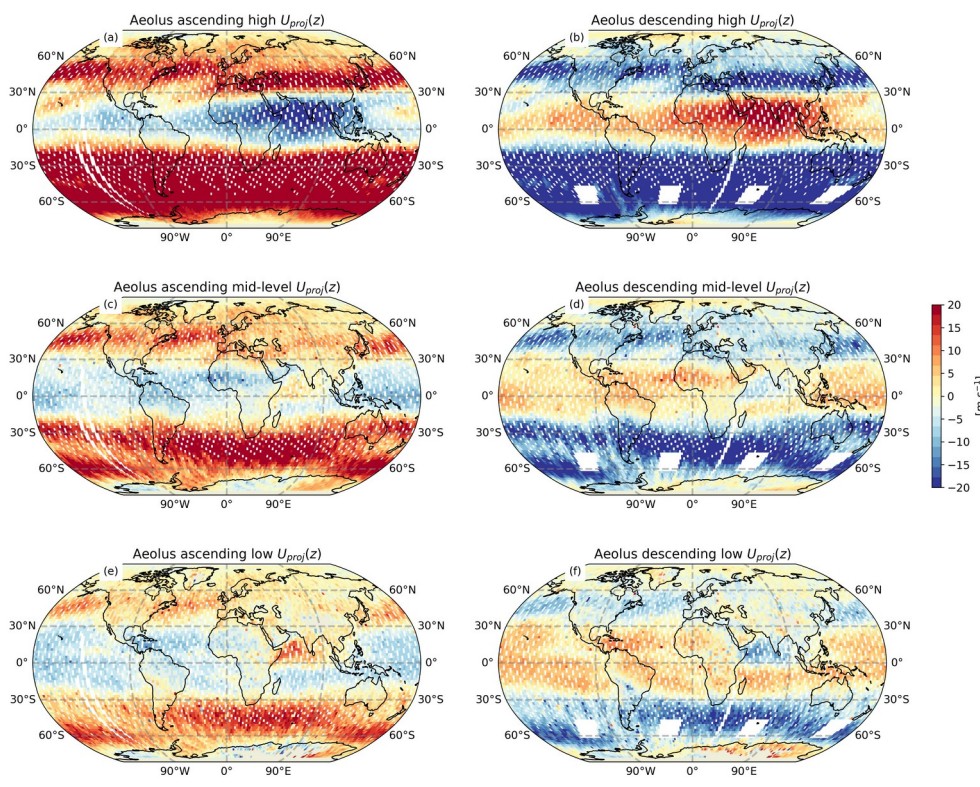

**Figure B1: Global maps of Aeolus all-sky winds (a) above 6.5 km, (c) between 3.2 and 6.5 km, (e) between the surface and 3.2 km for 1800 LT ascending orbits between Jun–Aug 2020. (b), (d) and (f) are the same but for 0600 LT descending orbits.**

Figure B1 illustrates that winds observed during ascending and descending orbits are nearly opposite and a change in the sign of the descending wind gives a good approximation of the zonal wind, especially within the latitude range 60°S-60°N. Indeed, differences exist between the wind observed at 0600 LT and 1800 LT and can be explained by a diurnal contrast of the wind and slight differences in the laser pointing direction (Fig. 1). It is possible to estimate zonal and meridional winds from Aeolus but these require making hypothesis about the wind direction or averaging successive ascending and descending orbits. Zonal wind retrievals are detailed in Krisch et al., 2022.

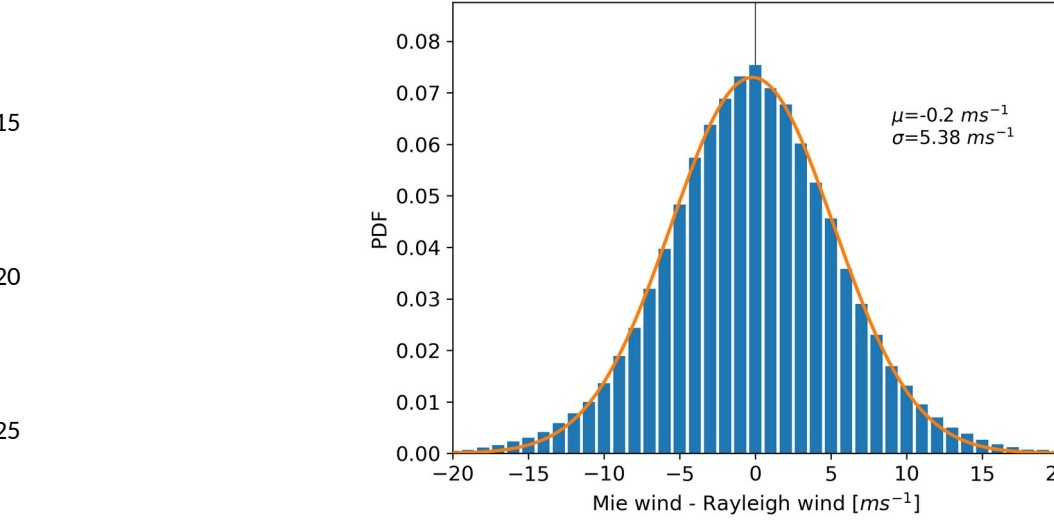

**Figure B2: Distribution of the differences between colocated, resampled, Mie winds retrieved in cloudy sky and Rayleigh winds that were substituted. Based on all bins during JJA 2020.**

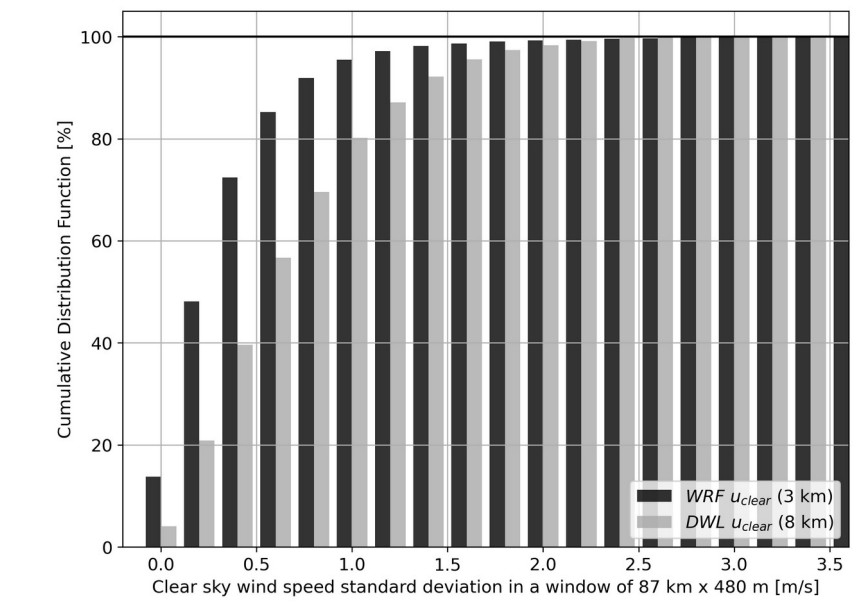

**Figure B3: Same as Fig. 5 but all altitudes between the surface and 19 km are included for WRF (about 4000 independent segments of 87 km x 480 m).**

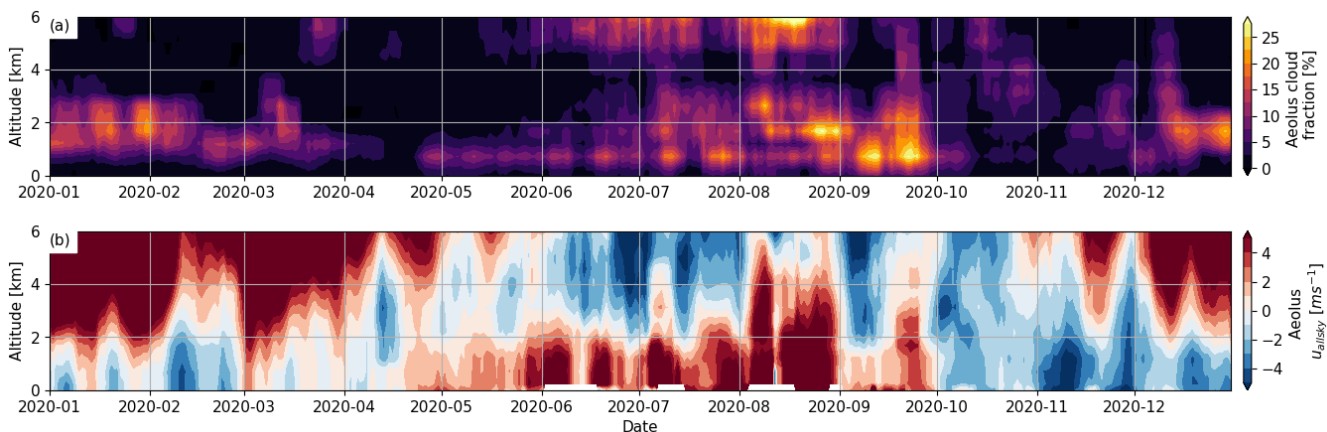

**Figure B4 :** Time series of (a) the average profiles of cloud fraction and (b) the average profiles of all-sky wind speed within (12°N-24°N, 55°E-75°E). A 7-day rolling mean is applied to see the typical direction of the wind and vertical cloud extent

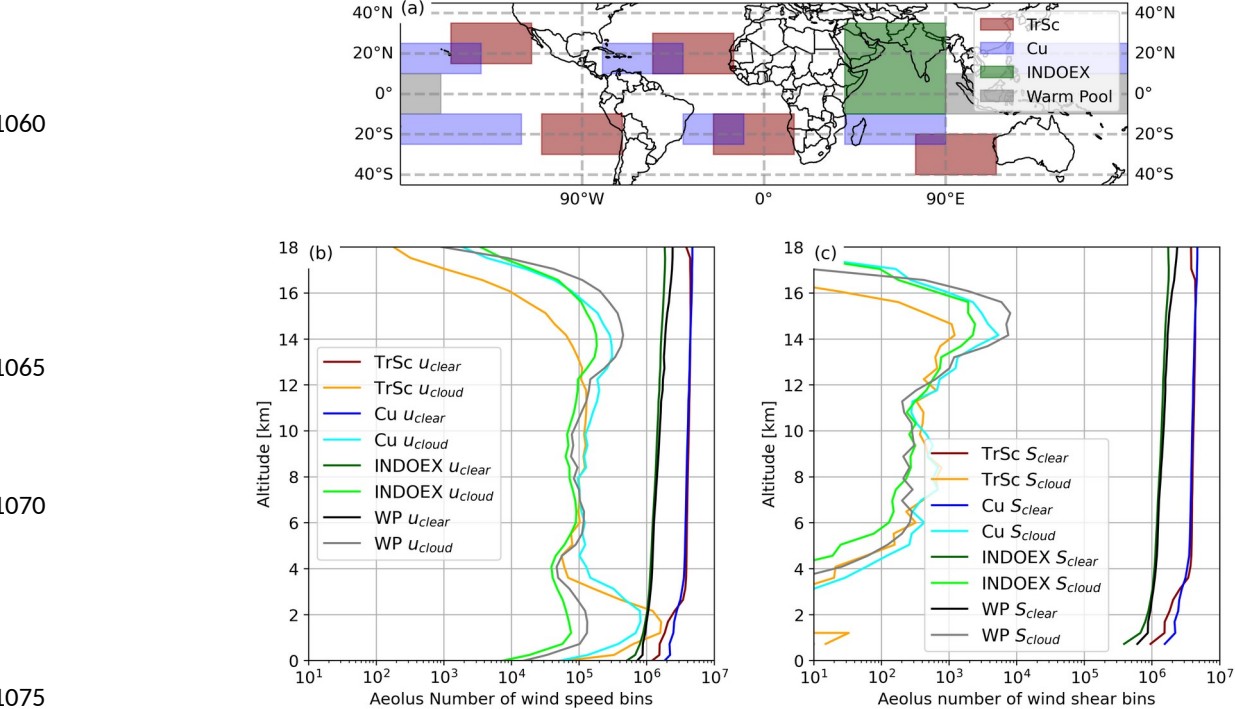

**Figure B5:** Map of the different regions for wind speed and wind shear differences. Stratocumulus transitioning (red) Cumulus (blue), INDOEX (green) and Pacific warm pool (black). Occurrences of (b) wind speed and (c) wind shear observations.

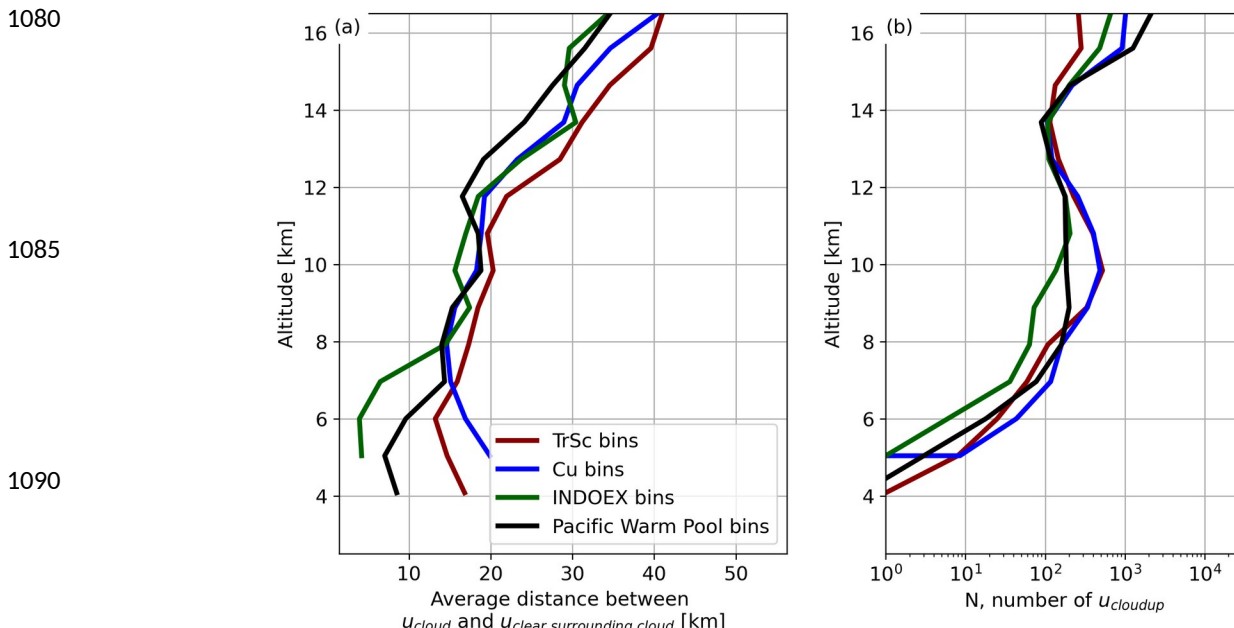

**Figure B6: Average distance between pairs of $u_{cloud}(z)$ and $u_{clear\_surrounding\_cloud}(z)$. All pairs separated by a distance of over 100 km were discarded. (b) Occurrences of pairs at each altitude level.**

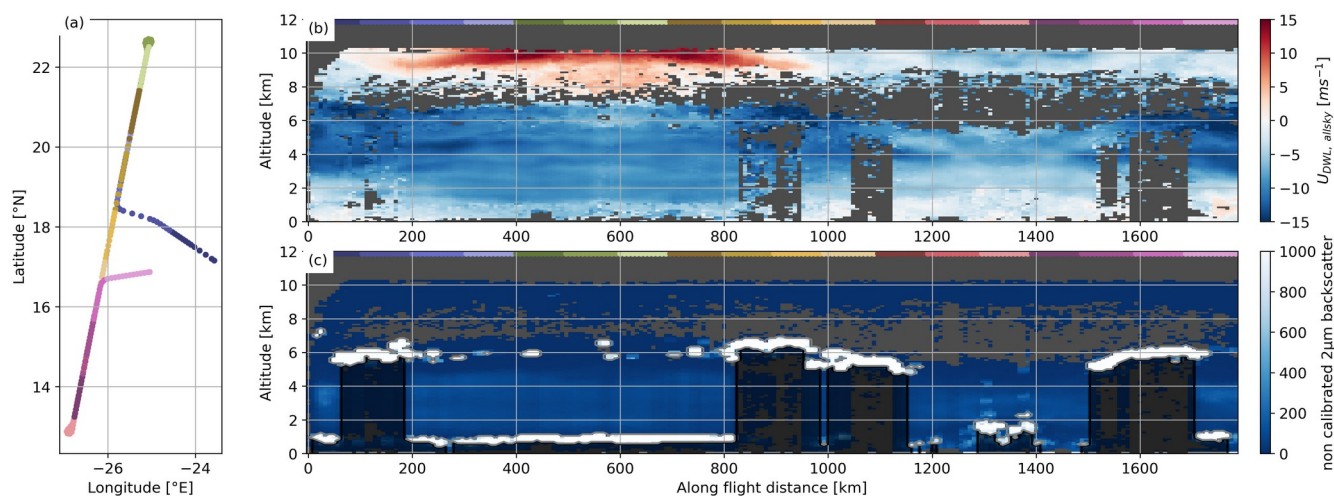

**Figure B7: (a) position of airborne LIDAR profiles during AVATAR-T Flight (2021-09-08, same as Fig. 10) over Cape Verde (b) horizontal wind projected along Aeolus line-of-sight ($u_{DWL,\ allsky}$) at a horizontal resolution of 8 km and vertical resolution of 100 m. and (c) uncalibrated 2µm backscatter with clouds contoured (grey) and the sub-cloud layers shaded (black). We estimated that uncalibrated 2µm backscatter exceeding 500 is associated with clouds.**

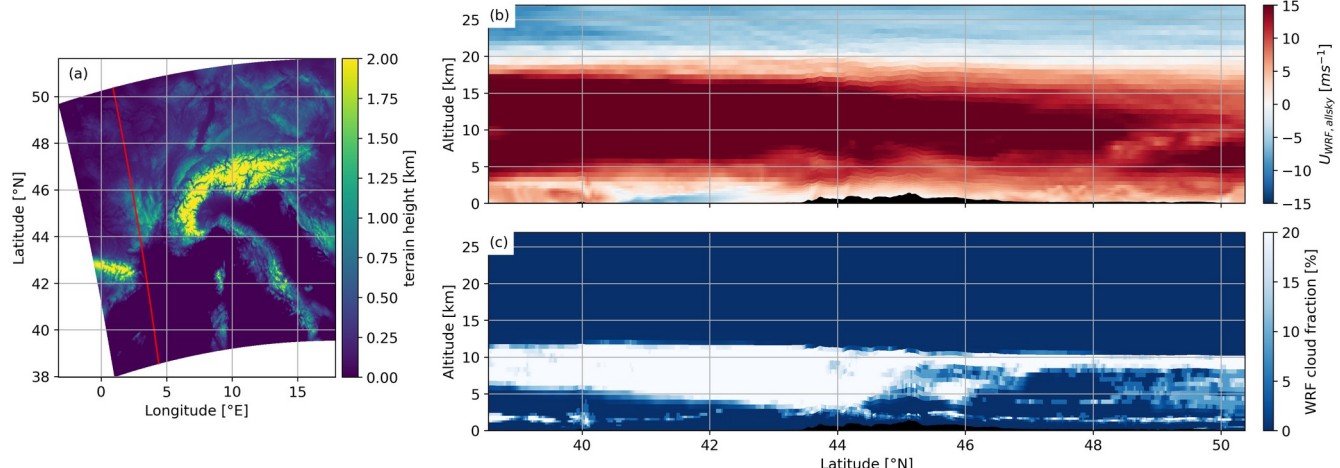

**Figure B8: (a)** terrain height of WRF simulation, the red curve corresponds to a theoretical orbit track of Aeolus. **(b)** Horizontal wind projected along Aeolus line-of-sight ($u_{WRF, allsky}$) at a horizontal resolution of 3 km and **(c)** the corresponding cloud fraction.

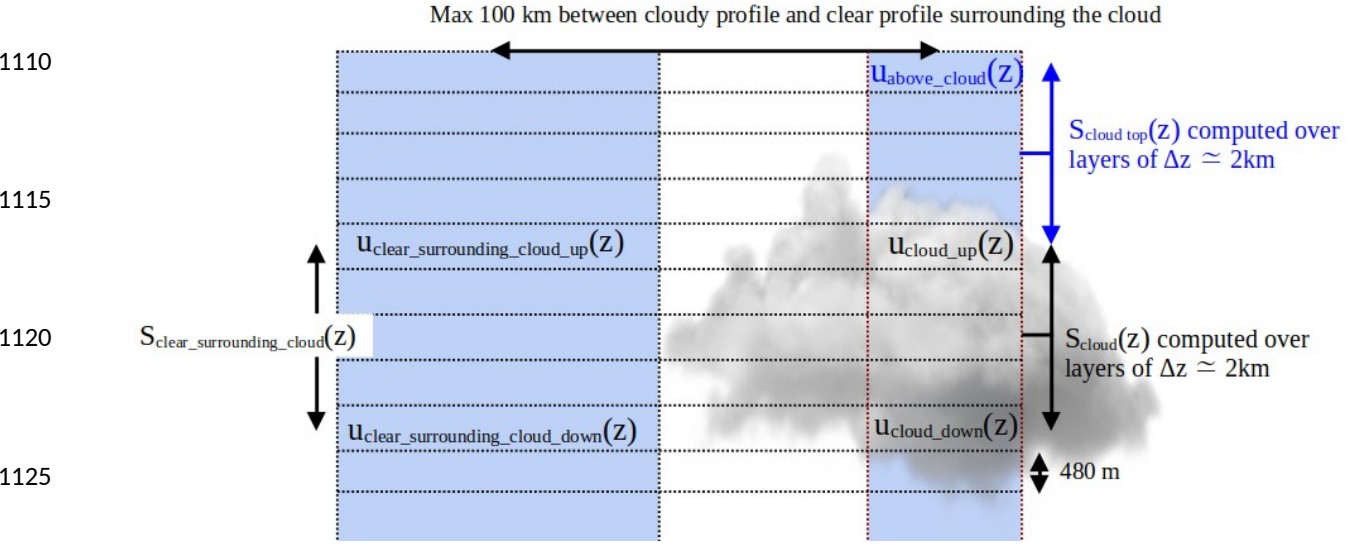

**Figure B9:** complements relative to the calculations of wind shears for Fig. 11. For each profile containing a cloud, we compute $S_{cloud\_top}(z)$ the wind shear between the clear sky above the cloud and the uppermost cloudy layer. We extract a sample of these profiles which has to respect two conditions : the cloud must be at least 2 km thick vertically, and there must be clear sky in the surrounding, within a distance of 100 km. We compute $S_{cloud}(z)$ the wind shear within the cloud, and $S_{clear\_surrounding\_cloud}(z)$ the wind shear in the surrounding of the cloud. Wind shears are calculated over 2 km thick layers.

The wind shear presented on Fig. B9 and along Sect. 4.3 are calculated as follows:

$$S_{clear\,surrounding\,cloud} = \frac{u_{clear\,surrounding\,cloud\,up} - u_{clear\,surrounding\,cloud\,down}}{\Delta z} \tag{5}$$

$$S_{cloud} = \frac{u_{cloud\,up} - u_{cloud\,down}}{\Delta z} \tag{6}$$

$$S_{cloud\,top} = \frac{u_{above\,cloud} - u_{cloud\,up}}{\Delta z} \tag{7}$$

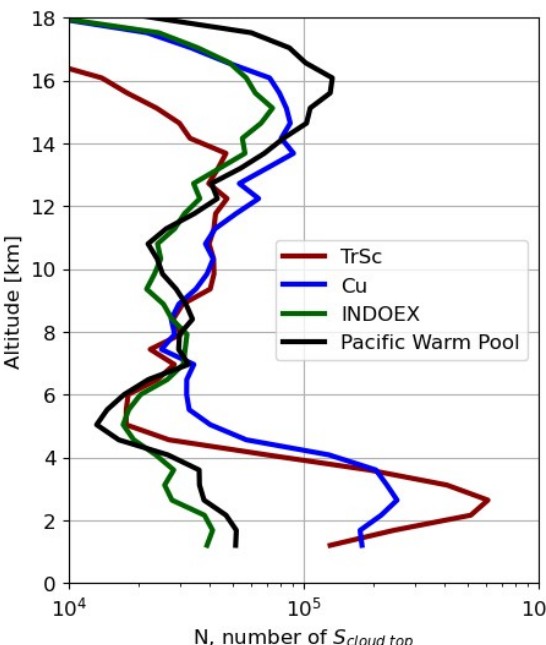

Figure B10: Occurrences of cloud top wind shears over each region

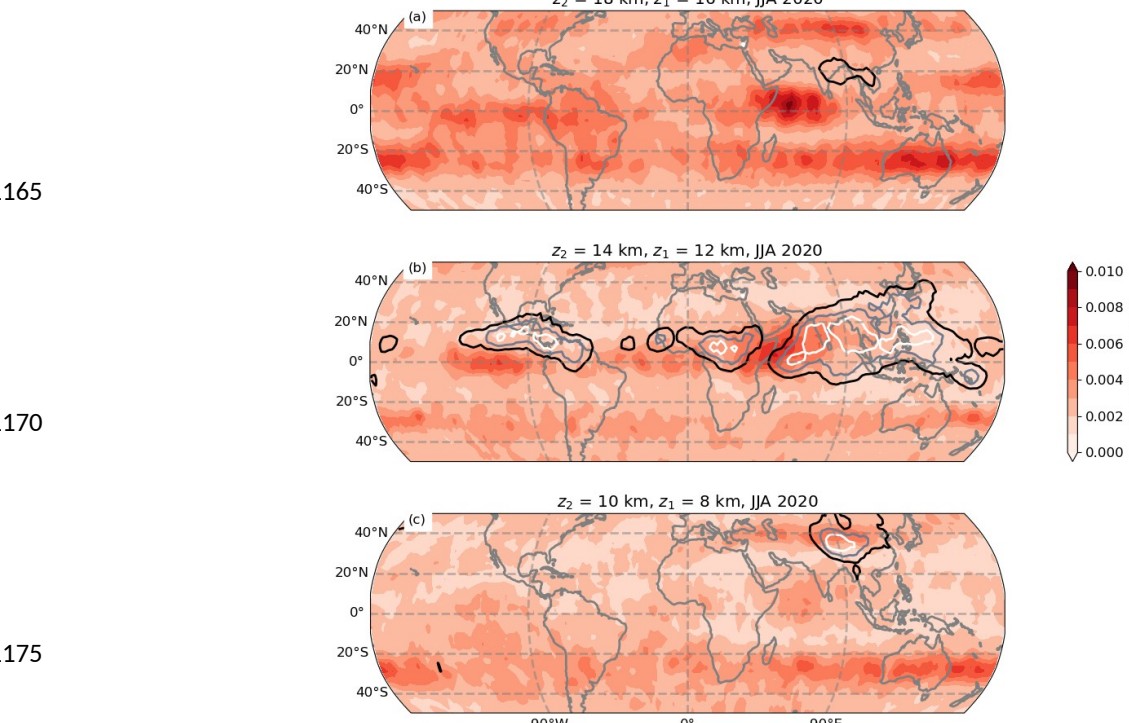

**Figure B11: Map of the median wind shear $|S_{allsky}|$ calculated a) between 8 km and 10 km, b) between 12 and 14 km and c) between 16 and 18 km from June to August 2020. Contours represent cloud covers of 5, 10, 15 and 20% for each altitude range.**

$$\left|S_{allsky}\right| = \frac{\left|u_{allsky}\left(z_2\right) - u_{allsky}\left(z_1\right)\right|}{\left(z_2 - z_1\right)}$$

(8)

*Data availability.* ALADIN/Aeolus orbit files and gridded data presented in this paper are available via AERIS (https://dx.doi.org/10.25326/746). They are built from Aeolus Level 1A and Level 2B observations that can be accessed via the ESA Aeolus Online Dissemination System (https://aeolus-ds.eo.esa.int/oads/access/). CALIPSO-GOCCP version 3.1.4 (Chepfer et al., 2010) and ERA5 reanalyses (Hersbach et al., 2020) were accessible via Mesocentre ESPRI/IPSL.

*Author contributions.* ZT, MB and HC drafted the article. The Aeolus dataset development was performed by ZT and AGF. The data analysis was performed by ZT, MB, and HC. The WRF simulation was performed by SB and the airborne lidar data were collected by BW. All authors were involved in the writing and investigation.

*Competing interests.* The authors declare that they have no conflict of interest.

*Acknowledgments.* The authors thank Mesocentre ESPRI/IPSL for the computation resources. We also thank ESA dissemination team for the access to Aeolus data and AERIS for the storage of the dataset. We thank Dr. Louise Nuijens for the interesting discussions regarding convective momentum transport that participated to the motivations of this study. The authors thank the people involved in the production of the WRF simulation, performed on the HPC resources of TGCC
under the allocations A0050106877 and A0070106877 made by GENCI. The authors also thank the DLR for the deployment of the Falcon and 2µm Doppler Wind Lidar during AVATAR-T campaign. We deeply thank the two anonymous reviewers for their great inputs, which helped to improve a lot the manuscript.

*Financial support.* This research was supported by European Space Agency and European Aeronautic Defence and Space
company (EADS) in the framework of ZT's PhD.

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
