# Peer review of "Demonstrating Aeolus capability to observe wind-cloud interactions"

_EGUsphere, 2025_

## Author Comment (AC1)

**Referee comments (questions) are in blue**

**Author comments (replies) are in black**

**Modifications made in the new manuscript by the authors are in red**

**Replies to Referee #1**

The manuscript by Titus et al. investigates wind-cloud interactions using data from the spaceborne Doppler wind lidar onboard the Aeolus satellite. To this end, Aeolus Level-1A data are processed to create a cloud mask at 3 km horizontal resolution, which is then used to resample the Aeolus wind data to a finer grid than that available in the original Level-2B wind product. The resampling approach is validated using airborne Doppler wind lidar measurements as well as a regional weather model simulation.

Using the resulting Aeolus cloud mask and wind data at 3-km resolution, the authors analyze the relationship between cloud cover, wind speed, and vertical wind shear within the troposphere during boreal summer 2020, at various spatial scales and across different global regions. Among other findings, they show that wind shear under cloudy-sky conditions is significantly smaller than in the surrounding clear sky. Moreover, the wind speed difference between cloudy and clear-sky conditions increases with the clear-sky wind shear, particularly in regions dominated by cumulus and stratocumulus clouds.

**General comment :**

This study is of significant interest to researchers in atmospheric physics, especially those studying global circulation and its interaction with cloud formation and development. The manuscript is well-structured, and the figures effectively support the analysis. The approach, starting from the Aeolus L1A product to derive a cloud mask at full 3-km resolution, is particularly innovative and promising.

However, the methodology outlined in Section 2.1 raises several questions and appears prone to potential systematic errors (see specific comments below). Similarly, the subsequent resampling of the Aeolus L2B wind data using the derived cloud mask needs further clarification to prevent misunderstandings about the approach.

While the use of airborne wind measurements and model simulations lends some confidence to the method, it is based on a few examples from a restricted geographical area. This raises the question of how representative these examples are for the much larger dataset analyzed later in the manuscript.

The results regarding wind-cloud interactions at both regional and sub-100 km scales are insightful and illustrate the capability of Aeolus to detect such phenomena, despite its relatively coarse resolution. As such, the study opens avenues for further research into topics such as jet stream shifts and their coupling with cloud radiative effects.

I recommend the manuscript for publication, contingent on the authors addressing the specific issues outlined below.

AC : We thank referee 1 for the useful comments and suggestions. Referee 1 highlighted potential issues with the processing of Aeolus L1A that we now addressed in the new version of the manuscript. We also clarified the resampling of L2B winds. We clarified why the airborne DWL observations, although geographically limited, are well representative of the tropical circulation in Sect. 3.2. Individual replies to specific and technical comments can be found below.

**Specific comments:**

1) Lines 72 ff.: The authors correctly state that Aeolus measured the projection of the horizontal wind on the laser line-of-sight (LOS). However, the symbol u is used throughout the manuscript (e.g., in Eq. (3), Figs. 6–11) to denote the horizontal LOS (HLOS) wind speed, which may be misleading, as u traditionally refers to the zonal wind component. I suggest adopting the notation vHLOS, which is commonly used in Aeolus-related literature.

AC : Following the referee recommendation, we specify in the Introduction of the new manuscript, lines 67-69 :
"The obtained measurement is not the actual horizontal wind, but the horizontal projection of the wind retrieved along Aeolus Line-of-Sight (LOS). In most of the Aeolus literature, this wind is noted as $v_{HLOS}$."

Regarding the notation "u" instead of vHLOS, we detailed in Sect. 3, lines 328-334
As this study is limited to the range 60°S – 60°N and as the laser pointing direction of Aeolus within this latitude range is quasi-eastward during ascending orbit and quasi-westward during descending orbit (Fig. 1, Krisch et al., 2022), the HLOS wind observed by Aeolus, that is often noted $v_{HLOS}$ in the literature, is simply noted "u" all along the paper for simplicity. We adopt the convention that u is negative (positive) when the wind is westward (eastward).

2) Line 120: To my knowledge, the number of accumulated backscatter profiles averaged onboard is 18, not 16.

AC : Correct. As the number of accumulated profiles is not necessary for the reader, we replaced the sentence of the previous version l. 120 : "We use the highest resolution Aeolus dataset (L1A), which corresponds to an on-board averaging of 16 subsequent accumulated backscatter profiles resulting in a horizontal resolution of 3 km along orbit track."

by new version line 114 : "In this approach, we use Aeolus Level 1A (L1A) raw data, with a horizontal resolution of 3 km."

3) Lines 140 ff.: Please provide a reference for the hot pixel map corresponding to the end of the mission. Since only a subset of these pixels was active during the main analysis period (primarily boreal summer 2020), discarding all of them may be unnecessarily conservative. This approach could introduce systematic errors in the retrieval of particulate and molecular backscatter profiles, and in turn affect the cloud mask. This is particularly relevant given that only very few pixels are used in the retrieval.

AC : The hot pixel map corresponding to the end of the mission (1 July 2023) is presented in the appendix A of the new manuscript (Fig. A2) together with the hot pixel maps of 14 June 2020 and 31 December 2020. We agree with referee 1 that using the hot pixel map from the end of the mission is unnecessarily conservative and therefore we switched to the map of 31 December 2020 in the new manuscript.

Figure RC1.1 shows the impact of considering the hot pixel map of the 31 December 2020 (Fig. RC1.1a) vs 1 July 2023 (Fig. RC1.1b) on the zonal average cloud fraction profile for JJA 2020. Overall, the changes of method and hot pixel map affected 3 rows of the detector, corresponding to altitudes of 1, 2 and 12 km and the zonal average cloud fraction profile has no discontinuity when using the hot pixel map of 31 December 2020 (Fig. A2b).

old manuscript Sect. 2.1, step 1, l. 140ff : "It has been known since the early days of the Aeolus mission that certain pixels of the detector are damaged by cosmic particles and that the number of these pixels almost linearly increases over the mission's lifetime (Weiler et al., 2021). Even though Weiler et al., (2021) proposed a compensation method for the signals affected by hot pixels, we preferred to discard all the pixels according to a hot pixel map corresponding to the end of Aeolus mission. Once the hot pixel is discarded, its value is replaced by an interpolated value of the two surrounding pixels. By doing so, we avoid potential false trends caused by a decreasing number of "normal" undamaged pixels with time.
"

replaced in the new manuscript by Sect. 2.1, step 1), lines 124-132: "It has been known since the early days of the Aeolus mission that certain pixels of the detector are damaged by cosmic particles and that the number of these pixels almost linearly increases over the mission's lifetime (Weiler et al., 2021). We discard the hot pixels following the hot pixel map of the 31 December 2020 (Fig. A2b), which corresponds to the end of the period that we considered for our study. Once the hot pixel is discarded, we apply a "sliding fit" approach (Goldberg et al., 2013; Feofilov and Stubenrauch, 2019) adapted to Aeolus (Feofilov, 2021), which considers the theoretical shape of the Mie backscatter spectrum. In this approach, a predefined spectral shape function is systematically shifted across each row of the Mie channel detector to find the optimal fit that minimizes the difference between the observed and theoretical spectral profiles, thereby simultaneously determining the Mie peak center frequency and reconstructing the complete spectral radiance values."

[Figure]

a)  14 June 2020      b) 31 December 2020      c)  1 July 2023

Figure A2. Hot pixel maps corresponding to (a) 14 June 2020 (compare with Weiler et al., 2021), (b) 31 December 2020 (end of our study) and 1 July 2023 (end of Aeolus mission). Hot pixels are identified by comparing each detector pixel value against its immediate neighbors within the same Mie detector row: pixels exceeding all neighboring values are flagged and counted across all daily orbits. The resulting frequency maps are normalized by their maximum count and thresholded at 0.2, with pixels above this empirical threshold marked as "hot".

[Figure]

Figure RC1.1 : Aeolus zonal average cloud fraction profile for JJA 2020 at 1800 LT using a) the hot pixel map of 31 December 2020 (end of the study),  b) the hot pixel map of 1 July 2023 (end of Aeolus) and c) the difference.

4) Sect. 2.1 and Appendix A: The derivation of particulate and molecular backscatter from the L1A signal appears oversimplified. Notably: a) The correction for solar background is not mentioned, although it also contributes to the overall signal. b) The use of signals from 3 and 6 pixels as proxies for particulate and molecular backscatter is a rough approximation. In contrast, Donovan et al. (2024b) describe a more rigorous method that involves determining crosstalk coefficients and accounts for the non-uniform intensity distribution across the Mie spectrometer. While a full implementation may be beyond the scope of this paper, the authors should explicitly acknowledge these simplifications and discuss their potential impact on the uncertainty in the derived cloud mask.

AC : a) In the new version of the manuscript, we have added  the correction for solar background and non-uniform intensity distribution across the Mie spectrometer following the method proposed by Donovan et al. (2024b), we did not include cross talk correction which is beyond the scope of this paper, but we correct for depolarization effect.

b) We chose signals from 3 pixels as proxies for particulate backscatter because "Mie signal peak" on the Mie detector occupies 3 pixels on each row.

Moreover, we assess the uncertainty on the derived cloud mask by comparing it to CALIPSO-GOCCP-COARSE in the new Fig. 2.

In the new manuscript, Sect. 2.1, step 2), lines 137-140, we added the following sentence : "For each profile and each altitude level, we subtracted the Detection Chain Offset (DCO, more details are given in Fig. A1), the solar background, and compensated for the non-uniform intensity distribution on the Mie spectrometer following Donovan et al., (2024b). Future work could include crosstalk correction."

Figure RC1.2 shows that when correcting for the solar background and non-uniform intensity distribution on the Mie spectrometer, the zonal average cloud fraction is almost unchanged.

[Figure]

Figure RC1.2: Aeolus zonal average cloud fraction profile for JJA 2020 at 1800 LT a) with a correction for the solar background and the non-uniform intensity distribution across the Mie spectrometer and b) without these corrections. c) shows the absolute difference between a) and b) shows the relative difference between a) and b) where cloud fraction is above 5%. The hot pixel map of 31 December 2020 is used.

5) Can the authors please comment on how the backscatter from aerosols, e.g., Saharan dust, is classified by their approach. I suspect that such regions are classified as "clear-sky". What are the implications on their studies of wind-cloud interactions?

AC : To study wind-cloud interactions, we want aerosols to be classified as clear and not as clouds. Therefore, in our study Saharan dust layers are classified as clear sky, but this was not explained in the previous version of the manuscript.

We added a sentence in the new manuscript Sect. 2.1, line 175 : "Note that this threshold will classify aerosol layers as clear sky."

And we added new pieces of information in Appendix A:

added paragraph in the new manuscript lines 328-333: "Note that the determination of the cloud detection threshold is evaluated while accumulating all altitudes between the surface and 18 km of altitude. Therefore this threshold is strongly weighted by the values of $I_{\text{part-alt-}\delta P}(z) - I_{\text{mol-alt}}(z)$ in the free troposphere, mostly free of Saharan dusts and aerosols, but containing a lot of clouds with larger $I_{\text{part-alt-}\delta P}(z) - I_{\text{mol-alt}}(z)$. Therefore, aerosol layers are classified as clear sky."

and Fig. A4 and lines 951-955: "Figure A4, shows the map of low and mid-level clouds during the season where the most Saharan dusts are observed. Overall, on the West coast of Africa at around 20° of latitude, Aeolus cloud cover is equal to or up to 10 % lower than CALIPSO-GOCCP below 8 km of altitude. CALIPSO-GOCCP cloud detection threshold was already restrictive enough to flag Saharan dust as clear sky. Therefore, Saharan dusts are even less likely to be flagged as clouds in our Aeolus dataset."

[Figure]

Fig. A4: Maps of mid-level (4-8 km) cloud cover (a) Aeolus dataset (b) CALIPSO-GOCCP and (c) the difference Aeolus - CALIPSO-GOCCP for the year 2020. (d), (e) and (f) are the same but for low level clouds (0-4 km).

6) Line 180: How can the total backscatter take values of exactly zero? I assume that, in case of strong signal attenuation, the values would fluctuate around zero due to noise. If a threshold was applied to discriminate such values, please specify.

AC : Right, we describe more precisely the full attenuation flag in the new version of the manuscript Sect. 2.1, lines 179-184 : "Below an opaque cloud, the laser is fully attenuated, making it impossible to retrieve valuable information, neither for the cloud detection, nor for the wind. For each profile containing cloudy layers, we evaluate $I_{mol-alt}(z)$ at each layer between the surface and 1 km below the lowest cloudy layer. If $I_{mol-alt}(z)$ at each layer is inferior to the noise level, all the layers between the surface and the lowest cloudy layers are flagged as fully attenuated. Otherwise, they are flagged as clear skies. For each orbit the noise level is defined as three times the standard deviation of $I_{mol-alt}(z)$ between 60°S and 40°S and between 16 and 18 km of altitude. "

old version of the manuscript Sect. 2.1, l. 181-182 has been removed : "For each profile, all the layers located below the lowest cloudy layer with a total backscatter I part-alt-δP (z) + I mol-alt (z) = 0 are flagged as fully attenuated."

7) l.282: Please clarify the term "closest L2B wind." Is proximity evaluated in both latitude and altitude, relative to the bin center?

This is important when duplicating L2B wind values, which are themselves derived from grouped L1B bins. I recommend to refer to the bin grouping algorithm described in the L2B Algorithm Theoretical Basis Document which should also be cited in the manuscript: https://earth.esa.int/eogateway/documents/20142/37627/Aeolus-L2B-Algorithm-ATBD.pdf

AC : This is now clarified for Rayleigh winds in the new manuscript, Sect. 3.1, line 294 : "We then display each wind at a fixed resolution of 3 km x 480 m. For each 3 km x 480 m bin, we look for the spatially closest L2B Rayleigh wind, evaluated in latitude along the orbit track and in altitude relative to the bin center (Rennie et al., 2020)."

as well as for Mie winds in the new manuscript, Sect. 3.1, line 298 : "For each 3 km x 480 m bin, we look for the spatially closest L2B Mie wind…"

8) l.296: Including statistics on how many of the resampled winds labeled as "clear" and "cloudy" correspond to L2B Rayleigh-clear and Mie-cloudy wind types would strengthen confidence in the cloud-masking approach.

AC : This was added to the new manuscript Sect. 2.1, step 3, lines 316-320 :

"During the period extending from June to August 2020, 83% of bins flagged as cloudy with our cloud detection contained both a Rayleigh and a Mie wind, while 10% of bins flagged as cloudy contained only a Mie wind and 7% only a Rayleigh wind (on the edge of clouds or at cloud tops). Similarly, 92% of bins flagged as clear contained only a Rayleigh wind, 7% contained both a Mie and a Rayleigh wind and 1% contained a Mie wind only."

9) l.325 ff.: Are the reported variations in wind speed shown in HLOS units? 5(c)-(e):

AC : Yes, This is now clarified in the response to specific comment 1)

10) 5(c)-5(e) : The effective vertical resolution of the resampled winds appears similar to that of the L2B product. Is this due to wind speed duplication across adjacent 3-km bins? If so, this should be discussed. Additionally, a finer vertical resolution of 480 m was expected, but is not evident, particularly at higher altitudes.

AC : Correct, it was due to the wind speed duplication across adjacent 3-km bins. We replaced the duplication by an interpolation procedure. The effective horizontal (vertical) resolution is now 3 km (480 m) everywhere. Figures 5c and 5e have been modified accordingly.
The new interpolation procedure is now mentioned in Sect. 3.1, lines 296-298 : "...before performing a 2D linear interpolation via a sliding average. The sliding window has the resolution of original L2B Rayleigh wind observations of Aeolus (87 km horizontally and 500 m to 1 km vertically)."

and lines 299-301 : "...before performing a 2D linear interpolation via a sliding average. The sliding window has the resolution of original L2B Mie wind observations of Aeolus (3 to 15 km horizontally and 500 m to 1 km vertically)."

11) l.552 ff.: The statement that "even at the coarse vertical and horizontal resolutions of Aeolus, it is possible to capture significantly different wind speeds within clouds and within their surroundings in shallow convection" may be misleading given that Fig. 9 presents airborne lidar data. This data has much lower random errors than Aeolus which enables it to resolve the depicted features. Please clarify this distinction.

AC : We corrected the paragraph in the new manuscript, Sect. 4.3.1, lines 620-625 : "The case study presented in Figure 9 shows that when averaging the airborne DWL wind data to the coarser horizontal and vertical resolutions of Aeolus, it remains possible to capture significantly different wind speeds within clouds and within their surroundings in shallow convection. This finding encourages us to observe the impact of convective motions on the horizontal wind speed with the Aeolus dataset. However, because of Aeolus wind observations having a larger random error than the airborne DWL wind observations, an averaging of multiple Aeolus wind profiles is necessary to observe significant wind speed differences between the cloud and its surrounding clear sky."

**Technical comments:**

Line 270: Define the term "observation_type" and specify which values represent "cloudy" and "clear." Ensure a space between numerical values and their units throughout the manuscript (e.g., "480 m" instead of "480m").

AC : Added in the new manuscript Sect. 3.1, l. ~262-263: "*observation_type* = 1, cloudy" and "*observation_type* = 0, clear"

also added l. ~264-265 : "The *observation_type* flag filters out Mie wind observations when particulate backscatter is weak (no aerosols nor clouds) and filters out Rayleigh wind observations in the presence of strong particulate backscatter (clouds or aerosols)."

Fig. 7, 8: Extend the color scales to show the full range of wind shear (Fig. 7) and wind speed (Fig. 8), respectively.

AC : The color scale was extended to 40 ms-1 for Fig. 7 and 35 ms-1 for Fig. 8

Caption of Fig. 8: Insert a semicolon after "February 2020" to improve readability.

AC : Semicolon added in Fig. 8 caption

Line 528: Define the acronym "LES" at first use.

AC : Large Eddy Simulation added in the new manuscript in Sect. 4.3, l. ~594

Line 550: Add missing punctuation at the end of the sentence.

AC : Punctuation added in the new manuscript in Sect. 4.3.1, l. ~619

Caption of Fig. 9: Specify that the data shown were acquired with the airborne Doppler wind lidar.

AC : Information added in the Fig. 9 caption in the new manuscript in Sect. 4.3.1, l. ~644 : "Curtain of wind acquired by the airborne DWL and projected along the laser pointing direction of Aeolus during the AVATAR-T campaign."

Section 4.3.2: At the beginning of the section, restate the time period covered by the dataset to aid reader comprehension.

AC : It is now added : "...large scale circulations and we focus on the entire year 2020."

Fig. 10: Use more distinct colors for "TrSc u_clear surrounding cloud up" and "Cu u_clear surrounding cloud up," which are difficult to distinguish.

AC : Colors have been changed : dark blue and cyan for Cu region, dark red and light red for TrSc, which are closer to colors chosen in Cesana et al., (2019). Colors were changed accordingly in the map (Fig. B2) and profiles (Fig. B3).

Caption of Fig. B1: The last sentence appears incomplete. Please revise.

Last sentence of Fig. B1 caption has been removed.

Line 792: Correct to: "can be explained" B6: Ensure consistency between the labels in the figure and the text: the terms "U_clear_neighbor_cloud_up" and "U_clear_neighbor_cloud_down" differ from the "surrounding" terminology used elsewhere.

AC : A new version of this figure, now Fig. B9, was implemented in the manuscript with the proper labels "uclear_surrounding_cloud_up(z)", "uclear_surrounding_cloud_down(z)" and also "Sclear_surrounding_cloud(z)".

**Replies to Referee #2**

**General comment:**

The authors introduce an all-sky interpolated wind dataset from Aeolus, and showcase some very promising applications. My main criticism here is that, for all the technical prowess of putting the dataset together, the authors show a limited amount of results, and apply a relatively simple analysis on it. There is also a lack of depth in the discussions with earlier literature. At times I felt that, with the current amount of actual novel results on wind-cloud interactions (relative to the dataset's complexity), this manuscript was maybe fitting more in AMT.

I still want to highlight the high ceiling of this study -- if the authors increase discussion depth and show more detailed analysis in some specific sections, this can become a very strong paper. Especially if specific comments 1+2 are addressed, this manuscript and follow-on studies will benefit a lot in my opinion. Below I list my suggestions to strengthen the current manuscript.

I should point out I am not an expert in cloud mask derivation and backscatter signal handling, but referee #1 fortunately has good insight on that part.

AC : We thank referee 2 for his useful and constructive comments. The analysis presented in this paper has been reinforced by adding the new Fig. 7 proposed by the referee. We also added a Figure in Section 4.3.2 (Fig. 10 in the new manuscript), showing typical wind profiles in regions dominated by different cloud regimes, and in this section we tried to show the novelty of our study by comparison with earlier literature. We also addressed specific comments 1) and 2) as well as all the others specific comments. Please see the detailed response to the comments here below.

**Specific comments:**

1) Section 2.1, l.160-165 : This is simply linear interpolation, correct? Call it as such. The largest possible vertical mismatch between datasets (going from the original vertical grid to the interpolated grid) is half the vertical grid spacing that you define.
Since the 480m won't necessarily coincide with the RBS of Aeolus, how about interpolating both Aeolus and CALIPSO datasets to a better resolution e.g. 200m or better? This way you minimize the portion of gradients that are lost in the interpolation, and avoid the largest mismatches in the vertical location of the data. While this aspect is not crucial for this particular manuscript (you show statistics of shear on 2km vertical scale in the end), you should make the reader aware that some of the gradients from Aeolus dataset are partially lost with these specifications.

AC : Regarding the linear interpolation, the referee is right, we performed a vertical linear interpolation of Imol(z) and Ipart(z) on the 480 m vertical grid prior to the cloud detection (Sect. 2.1). Similarly, we also performed a 2D vertical and horizontal linear interpolation of the Mie and Rayleigh wind speed values on the 3 km horizontal x 480 m vertical grid. This is

more clearly specified in Sect. 2.1, line 153 and in Sect. 3.1, lines 295 and 299 of the new manuscript.

Regarding Aeolus vertical resolution, the Aeolus RBS is 500 m in the boundary layer and 1 km in the free troposphere based on ATBD of L1B data (used to produce the L2B wind). A reference to this ATBD is now added in the new manuscript line 151 https://earth.esa.int/eogateway/documents/20142/37627/Aeolus_ATBD_Level_1B_Products.pdf/83e4f869-1632-bb60-f133-a71b821c32e9. Every mention of "250 m vertical resolution" in the old manuscript (lines 158 and 260) was replaced by "500 m vertical resolution" in the new manuscript, lines 150 and 255.

Moreover, we agree with the referee that interpolating at a vertical resolution of 480 m implies possibly losing portions of gradients from the original Aeolus dataset because of vertical mismatches in the altitudes of both dataset. Nevertheless, we keep a vertical resolution of 480m, to facilitate the use of Aeolus data by the climate model community (CALIPSO-GOCCP, COSP Lidar Simulator). A first example of such study at the interface between Aeolus data and a climate model focusing on clouds alone can be found in Roussel et al., (ACP discussion, submitted). Another study using both winds and clouds is currently under preparation.

We now mention in the new manuscript, Sect. 2.1, lines 109-114: "Hereafter, we build Aeolus cloud statistics based on a cloud mask built at 480 m vertical resolution and best possible horizontal resolution to compare Aeolus data with CALIPSO-GOCCP (Chepfer et al., 2010) and to facilitate future use of Aeolus data by the climate model community through the COSP Lidar Simulator (Bodas-Salcedo et al., 2011). To build this cloud mask from Aeolus particulate and molecular backscatter profiles, we follow an approach similar to the one proposed by Donovan et al., 2024b, with the following additions: a cross-polar correction from CALIPSO-GOCCP and a dedicated processing of hot pixels."

and at lines 158-160 : "Note that the choice of a 480 m vertical resolution implies possibly losing portions of gradients from the original Aeolus dataset due to altitude mismatches in the original and re-sampled datasets."

2) Section 2.2.2, l.252-255:
Nice that you explain and detail this immediately, I was very intrigued by this feature in Fig. 2 from the beginning. For a better comparison, would this motivate coarsening CALIPSO horizontal resolution to Aeolus'? I would strongly encourage doing so, in combination with my comment about vertical resolution of the datasets (1). --> In the end you want to avoid
a) the artifact of coarser horizontal resolution on cloud fraction values (which to some degree might affect clouds at other layers too, impossible to rule out or quantify in the current dataset)
b) losing any of the stronger wind shear values from Aeolus

If you performed your horizontal-vertical interpolation this way, results would become a lot more robust, and the comparison settings would be ideal in my opinion.

Follow-on work is planned by the authors with this dataset, and if the authors want to do more detailed research (which will eventually require shorter vertical scales than 2km),

thoroughly addressing specific comments 1) and 2) would really help to get robust results out of their dataset.

AC : Following the referee suggestion, "The horizontal along orbit track resolution is 333 m for CALIPSO-GOCCP and 3 km for Aeolus. For a consistent comparison between the two instruments, we build the CALIPSO-GOCCP-COARSE dataset, whose spatial resolution is set to the same as Aeolus (3 km) prior to the cloud detection." is specified in Sect. 2.1, lines 2.19-221, and we updated Fig. 2.

New Fig. 2 shows that overall CALIPSO-GOCCP-COARSE sees the same cloud patterns as Aeolus. The differences are non-significant everywhere (dotted bins) except in the boundary layer below the ITCZ (10°S - 25°N, surface - 2 km), the laser being often fully attenuated in the free troposphere by high clouds.

[Figure]

New Figure 2 : Zonal average of cloud fraction profiles for (a) Aeolus at 1800 LT and (b) CALIPSO-GOCCP-COARSE at 0130 LT corrected for the diurnal cycle. (c) is the absolute difference between Aeolus and CALIPSO-GOCCP. Non-significant differences (T-test with p-values $< 0.05$) are dotted. The lowest bin encompasses the surface or is under the ground and is discarded in this study (opaque gray bar). Cloud fractions $< 1\%$ are masked in gray.

3) Section 3.2, l.316-318 why not interpolate to 480m as in the previous comparison?

I would strongly suggest to make the same vertical interpolation as in section 2

If you decide for 480m, you can average 5 consecutive bins with appropriate weighting to account for less representativity at the upper-lower edges.

AC : Correct, we implemented this suggestion here.

4) Beginning of section 4.1:

Not sure this part (Fig.6 and related discussion) provides really novel results, it could be moved to the appendix/supplement.

See below some suggestions to strengthen this section (Fig.7)

AC : We followed the referee's suggestion and included Fig. 7 (see reply to comment below). We still kept Fig. 6 as it  is a common representation  among the community studying jet-cloud interactions.

5) Figure 7:

This is not really representative of shear and shows basically what the circulation is at 10km.

Preferred format:

- take e.g. the 8-10km layer
- calculate ABSOLUTE shear between the levels within that layer from your resampled data --> you'll have a distribution for JJA

--> show e.g. the median of that

A figure 7 with panels like that for several height ranges e.g. a) 8-10km b) 12-14km and c)16-18km

--> would be a lot more informative of shear conditions and a more novel contribution to literature.

Note that can be easily obtained from your dataset as it is.

AC : This is a great suggestion. We have added a new Fig. 7 in Sect. 4.1 following the referee's suggestion.

[Figure]

New Figure 7: Map of the median wind shear $S_{allsky}$ calculated a) between 8 km and 10 km, b) between 12 and 14 km and c) between 16 and 18 km. Maps are 2° x 2°, from June to August 2020. Contours represent cloud covers of 5, 10, 15 and 20% for each altitude range.

- How come regions where the ground is above 2km are not masked out? What level is being used in those cases?

AC : This was induced by the surface elevation not being properly recorded for a few profiles but is now solved even though the figures which were affected are not in the paper anymore (Fig. 7 in the old manuscript).

6) Method to average shear in section 4.3.2

with this method, if you have large westerly and large easterly shear values within your sample, they will average out.

It'd be more interesting to show the profiles of S, with the averaging done over their absolute values.

I.e. $|S|_(z)$

In Fig. 10, you only show shear within the cloud, and the equivalent clear surroundings.

AC : It is true that when averaging instantaneous values of S(z), positive and negative shear will average out. However, standard error on S(z) is reduced when computing the average of S(z) (because random error on individual values of S(z) can be positive or negative). When averaging instantaneous values of |S(z)|, wind shear values having opposite signs won't cancel out, but neither will the random error. Therefore, to reduce the standard error, we compute the average (or median) values of S(z) before moving to the absolute value (as in Fig. 7).

Fig. 7: Furthermore, as the results presented are averaged during JJA 2020, most large scale circulations do not change direction and keeping the sign of S(z) tells us in which direction the median shear is, which is very nice when looking at the new map suggested by the referee.

For comparison with Fig. 7,  Fig. AC2.1 below, shows the median of |S(z)| :

[Figure]

Figure AC2.1 : Same as Fig. 7 but with the median of $|S_{allsky}|$

Could you show a third panel with the same but above the cloud top? I.e. difference between u_cloud_up and +2km.

--> Would be very valuable to compare shear values within and right above the cloud

Again preferably as $|S|_{(z)}$

AC : The referee is right to point out that adding the information about S at the cloud top and above would be very interesting. We plan to do it in a following study dedicated to specific cloud types and associated scientific questions.

7) l.612-620 Very basic statistical concepts are attempted to be explained here, and they are coming out wrong. What you call your random error, decreasing by factor N^(1/2), is simply the standard deviation of your mean value. "The random error is thus smaller than the typical horizontal wind speed difference... " The only valuable use of your std of the mean of cloudy layer and (std of the mean of) surrounding clear sky; is to tell whether the difference between them (the two means) is significant.

AC : We agree that we misused the expression "random error", we actually meant "standard error". Furthermore, it was confusing as this is not the "random error" nor the "standard error" that we used for the significance testing. We therefore removed this paragraph in the manuscript as it was not useful. Moreover, we corrected the manuscript as follows:

The following sentence line 395 in the old manuscript: "a temporal or spatial averaging is necessary to reduce the random error of Aeolus wind measurements" was removed

and removed the following sentence line 589 in the old manuscript : "In order to study wind speed differences of only a few ms-1 , it is necessary to reduce the random error and therefore average a large number of wind profiles."

Which is the significance testing that you apply in Fig. 10? Two-sided t-test, Welch test?

AC : The information about the significance testing is added in the new manuscript lines 723-724 : "We performed a two-sided T-test at every altitude level. We consider that the mean wind speed (or wind shear) profiles are significantly different when p-value < 0.05"

"documented to be several ms -1 generally." --> firstly, references are needed for this statement, and secondly, the documented random errors of Aeolus Rayleigh clear or Mie cloudy measurements are not calculated the way you do and thus are not comparable.

I suggest removing these lines.

AC : Correct, we removed these lines from this section.

8) text of section 4.3.2 in general

No discussion about tropical cloud or shear literature whatsoever. Needs to be expanded and highlight what this analysis adds upon previous works.

AC : We integrated more information about tropical clouds in Sect. 4.3.2.

We also added references and sentences on wind shears and on tropical clouds in Section 4.3.2, to highlight what this analysis adds upon previous works.

lines 654-656 about Stratocumulus clouds : "They are prevalent in the eastern subtropical oceans (Wood, 2012) and are capped by a strong inversion, usually created by the large scale subsidence associated with the descending branch of the Hadley-Walker circulation. The inversion is characterised by a sharp transition in most meteorological variables (Wang et al., 2008; **Hourdin et al., 2019**)"

lines 658-659 about Cumulus clouds : "Cumulus clouds are usually found in the western subtropical oceans and are associated with a deeper boundary layer compared to the Stratocumulus region (**Scott et al., 2010**)"

lines 666-685 : "While **Houchi et al. (2010)** performed a climatology of atmospheric horizontal wind and wind shear, they did not examine the typical wind profiles for different cloud regimes… **Tian et al., 2021**, **Savazzi et al., 2022**…" along with Fig. 10

line 687 : "**Hibbert et al., 2023**"

lines 724-726 about lidar observations of Stratocumulus and low liquid clouds : "Because of the smaller vertical extent of low clouds, and particularly boundary layer clouds (Wood et al., 2012; **Cesana et al., 2019**) which typically do not exceed 1 km, and because of the stronger attenuation of low liquid clouds (**Guzman et al., 2017**), there are no occurrences of Aeolus wind shears calculated over 2 km vertical layers below 5 km of altitude (Fig. B6)"

9) Would it be possible to compare the wind on the cloud's uppermost layer from your all-sky dataset (that you show in this manuscript), compared to the collocated Rayleigh-clear value -- which you substituted by cloudy wind.

It would be super interesting to see whether there's agreement or some systematic differences between the two.

Especially as most studies rely on using Rayleigh-clear winds, this would really show the added value of using your dataset for a better estimation of wind shear around (and atop of) clouds.

AC : We added a paragraph at the end of Sect. 3.1 to quantify within clouds, what are the average and standard deviation of the differences between Mie winds and the substituted Rayleigh winds. In the new manuscript section 3.1 l. ~347-353 : "For cloudy bins where both Mie winds and Rayleigh winds coexist, the difference between Mie and Rayleigh wind is -0.20 ms$^{-1}$ in average over all data collected from June to August 2020 (Fig. B2). However, the standard deviation of the differences is quite large at 5.38 ms$^{-1}$. This is essentially a consequence of the random error on wind observations and differences in horizontal resolution of the Mie winds and the Rayleigh winds. The substitution of Rayleigh winds by Mie winds in cloudy sky will improve the study of wind-cloud interactions because the random error of Mie winds (3 ms$^{-1}$) is lower than for Rayleigh winds (5 ms$^{-1}$) and the horizontal resolution of Mie winds is at least 5 times finer than for Rayleigh winds."

**Minor/technical comments:**

Title

This is more of a demonstration of the capabilities of your curated dataset, results on wind-cloud interaction itself don't really go in-depth.

I suggest adjusting the title accordingly, e.g.: "Demonstrating Aeolus capability to observe wind-cloud interactions with a merged all-sky dataset"

AC : We replaced the title by "Demonstrating Aeolus capability to observe wind-cloud interactions".

l.20-21: please specify this is shear inside the cloud.

AC : We replaced "shear in cloudy sky" by "shear inside clouds…", line 19 in the new manuscript.

l.63-68: please provide a summarized version for the intro, such detail belongs more in data/methods section. Make the intro less technical, focus more on the key things you've done that make this novel approach valuable. Same for l. 71-76

AC : We removed the following technical sentences from the introduction: "Compared to the approach introduced by Donovan et al., 2024b, we apply slight differences such as a compensation of the missing cross-polar signal of Aeolus from a climatology of the GCM-Oriented Cloud-Aerosol Lidar and Infrared Pathfinder Satellite Observation Cloud Product (CALIPSO-GOCCP) observations, as well as the systematic discarding of hot pixels." (l. 65-69 in the old manuscript)

"which have been continuously validated during the mission with airborne lidars (Lux et al., 2020 ; Witschas et al., 2020 ; Witschas et al., 2022), ground based lidars, radars and radiosondes (Ratynski et al., 2023 ; Iwai et al., 2021 ; Belova et al., 2021 ; Baars et al., 2020)." (l. 75-78 in the old manuscript)"The Rayleigh and Mie channels have horizontal and vertical resolutions that vary to optimize the signal to noise ratio and vertical coverage. The latest validation campaigns of Aeolus showed that the systematic error (bias) for wind measurements remained within the mission requirements of 0.7 ms$^{-1}$ for both Mie and Rayleigh channels, while the random error was about 3 ms$^{-1}$ for the Mie channel and 5 to 7 ms$^{-1}$ for the Rayleigh channel." (l. 80-84 in the old manuscript)

"(reduced by a factor N$^{1/2}$ with N independent profiles)" (l. 86 in the old manuscript)

and refined the beginning of Sect. 3, lines 246-258 :

"The wind profiles from Aeolus Level 2B (L2B) scientific wind product have been continuously validated during the mission with airborne lidars (Lux et al., 2020; Witschas et al., 2020; Witschas et al., 2022), ground based lidars, radars and radiosondes (Ratynski et al., 2023; Iwai et al., 2021; Belova et al., 2021; Baars et al., 2020). So far, Aeolus wind data (L2B) provided to the community are orbit files that contain 2 types of wind profiles (the Mie wind and the Rayleigh wind) estimated from the molecular and particulate backscattered signals respectively. The latest validation campaigns of Aeolus showed that the systematic

error (bias) for wind measurements remained within the mission requirements of 0.7 ms$^{-1}$ for both Mie and Rayleigh channels, while the random error was about 3 ms$^{-1}$ for the Mie channel and 5 to 7 ms$^{-1}$ for the Rayleigh channel. This study benefits from the latest reprocessing of L2B Baseline 16.

The Mie and Rayleigh wind profiles have a varying vertical resolution (500 m to 1 km) but also a varying horizontal resolution (ranging from 3 km to 15 km in the Mie channel and fixed at 87 km in the Rayleigh channel). Having a dataset with Aeolus wind profiles resampled at the same fixed resolution as Aeolus cloud profiles is crucial to ease the use of these data for wind cloud interaction studies."

Figure 1: Can be moved to an appendix or supplement, not really relevant/necessary here, geometry of HLOS is very well known.

The referee is right in the sense that this figure doesn't bring anything to people familiar with Aeolus LOS (or DWL in general) but perhaps this is useful for people with no expertise, who could assume that Aeolus retrieves both components of the horizontal wind. We left it as Fig. 1.

l.113: the airborne DWL, please state here from what campaign.

AC : This was mentioned in the old manuscript l. 310 and l. 534, but this is now also mentioned in the Introduction, lines 101-102: "...to a higher horizontal sub-grid resolution of 8 km using high spatial resolution airborne Doppler Wind Lidar (DWL) AVATAR-T (Aeolus Validation Through Airborne Lidars in the Tropics) campaign, and…"

l.124-131: you need to name here the data requirements for your study: why you process L1A yourself, adjust to CALIPSO vertical resolution for comparison, etc..

AC : In the new version of the manuscript, we added the requirements for our study at the beginning of Sect 2.1, lines 109-114 : "Hereafter, we build Aeolus cloud statistics based on a cloud mask built at 480 m vertical resolution and best possible horizontal resolution to compare Aeolus data with CALIPSO-GOCCP (Chepfer et al., 2010) and to facilitate future use of Aeolus data by the climate model community through the COSP Lidar Simulator (Bodas-Salcedo et al., 2011). To build this cloud mask from Aeolus particulate and molecular backscatter profiles, we follow an approach similar to the one proposed by Donovan et al., 2024b, with the following additions: a cross-polar correction from CALIPSO-GOCCP and a dedicated processing of hot pixels."

l.134: It reads very awkward to me, that these 6 steps end up with the letter 'g' as the final product... Please number them 1-6 accordingly.

AC : Agree, we now number them 1-7 as well as for the re-sampling of the wind in Sect. 3.1 which was also indexed with letters in the old manuscript.

l.190-194: please state here and/or at the beginning of this subsection, whether the method is similar to previous ones for cloud detection, and benefits of making your additions.

AC : Correct, we now detail this in the beginning of Sect. 2.1, lines 109-114 (see the reply to minor comment about lines 124-131).

l.212-214: this is the kind of detail that needs to be brought up much earlier, to reassure the reader that the method/comparison is not coming out of nowhere ;) and has been validated before

- Also I think the AMT paper from 2022 by Feofilov should be cited instead of the 2024 book chapter.

AC : Agree, we now reference Feofilov et al., 2022

Figure 2: panel c) please mask out insignificant values.

- adding a panel d) with same as panel (c) but standard deviation of the differences, would be very helpful in my opinion.

AC : We changed Fig. 2 and masked insignificant values in panel c).

It would show which regions are the most uncertain. In (d) no significance testing would be needed.

l.318-320: please start a new paragraph somewhere around this point, the current one is too long and information-dense.

AC : We start a new paragraph before "We then extract from the segment…", line 352 in the new manuscript.

l.338: "not shown" --> please add figure to the supplement to support this statement.

AC : Fig. B3 was added in the Appendix B.

Figure 4: please increase the size of the axis labels and the legend

AC : Correct, font was increased for Fig. 4

beginning of section 3.3: Feels a bit like coming out of the blue, please motivate at the beginning of subsection 3.3 and at the beginning of section 3 that this case study serves as a great example of your resampling output.

We add the following introductory sentence in the new manuscript Sect. 3.3, lines 401-403: "Figure 5 illustrates how Aelous resampled cloud mask and winds allow us to observe from space different features ranging from cyclones to cumulus clouds".

and removed : "For a general audience, the most famous phenomenon associating clouds and winds is certainly the cyclone. This mesoscale system that develops over warm oceans combines both an opaque cloud cover that can be observed from space (Fig. 5a) and very fast winds around its centre." in Sect 3.3, lines 378-381 in the old manuscript.

Also, please name the tropical cyclone in the subsection title and text.

AC : Right, this is the tropical cyclone Paulette

The name Paulette is now mentioned in the subsection 3.3 title (line 400), in the text (line 404) and the caption of Fig. 5.

l.382: 'cyclic season' sounds very awkward, do you mean hurricane season?

AC : This was a typo, we replaced it with "hurricane season" in the new manuscript line 404

l.383-386: state the name of the TC and the date in this paragraph

AC : This is now mentioned line 404 : "Figure 5a shows an example of intersection between Aeolus and the tropical cyclone Paulette over the Atlantic Ocean during the hurricane season, on 12 September 2020."

Figure 5:

- caption: specify convention of positive-negative winds. (even if one can guess from the figure itself, and mentioned elsewhere in the text, should be specified in the caption)

AC : This is now specified in the caption of Fig. 5

- In panels d+e, I suggest adding a contour line (e.g. grey for contrast with dark blue) surrounding the cloud mask for a better reference

AC : The contour line has been added surrounding the cloud mask in Fig. 5

Figure 6: you may cut the figure at 65° latitude then

AC : This is now done.

- un-saturate the color scale, you can go till 40m/s
- I suggest to make color separations every 2.5m/s, as it is now, one cannot tell by eye from the figure whether it is 20 or 25m/s

AC : Correct, both suggestions were included

l.480: very awkward wording --> "As a result, IT is also .... WHERE Aeolus retrieves... "

AC : This text has been removed as Fig. 7 has changed

l.484: " that are typical of a strongly stratified free troposphere."

--> I'm not sure this is general knowledge in shear-related literature, do you have a reference backing up those numbers?

AC : This sentence has been removed as Fig. 7 has changed

l.505-506: is this a novel finding? I'd like to see some further discussion with more references in this section

AC : The expression "The most striking feature…" line 505 in the old manuscript was misleading. This is not a novel finding, the sentence was changed and we added new references (see the comment below).

l.508: more up-to-date literature on Indian ocean cloud cover would be nice to discuss here as well.

AC : We added the following references about the Indian Ocean:

lines 510-511 about deep convection over the Bay of Bengal: "This region is also subject to deep convection, particularly over the Bay of Bengal (**Zuidema, 2003** ; **Hemanth Kumar et al., 2015**)"

lines 543-544 about the longitudinal variability of the core of the Tropical Easterly Jet : "over the Arabian Sea from June to August 2020 (consistent with **Liu et al., 2024**)"

lines 557-561 about the seasonal cycle of low tropical marine clouds: "Part of this seasonal cycle of cloud top height is explained by a cooler Sea Surface Temperature (SST) from January to February, favorable for higher cloud tops (**Höjgård-Olsen et al., 2022**)), and a warmer SST afterwards. Moreover, during the reversal of the winds, the evaporation flux at the surface the Indian Ocean is reduced, resulting in a shallower and dryer boundary layer, less favorable for the formation of low clouds (**Mieslinger et al., 2019**, **Nuijens and Stevens, 2012**)."

lines 564-565 about the cover of deep convective clouds : "However, deep convective cloud cover accounts for only 9% of the area of the Indian Ocean in July (**Massie et al. 2002**)"

lines 569-571 about the reduced lifetime of cirrus clouds in the presence of large wind shear : "However, below and above the core of the TEJ, we observe wind shears larger than $10^{-2}$ s$^{-1}$, which were found to alter the cirrus structure and reduce its lifetime (**Jensen et al., 2025**)"

and lines 571-573 about the diurnal cycle of cirrus clouds : "Note that the time of ascending orbits of Aeolus (1800 LT) corresponds to a maximum of continental deep convection (Noel et al., 2018) and maximum of deep convection over the Bay of Bengal (Zuidema, 2003), while descending orbits (0600 LT) occur before the dissipation of cirrus clouds (**Ali et al., 2022**)".

l.541: "We performed a sliding average vertically of 500 m"

--> For a second time on the average profile, or is this just a repeated phrase? (you mention this on lines 538-539)

AC :

We clarified the processing in Sect. 4.3.1 lines 600-606: "We then coarsen the uncalibrated backscatter at the native horizontal and vertical resolutions of Aeolus before interpolating the signal at 3 km horizontal x 480 m vertical resolution, to create a cloud mask (Fig. 9b), consistent with our dataset. In the same way, we coarsen the profiles of $u_{DWL, allsky}$ at the native resolution of Aeolus, before interpolation on the 3km x 480 m grid. We further average the $u_{DWL, allsky}$ profiles encompassing the centre of the cloud between 1280 and 1320 km along flight (Fig. 9c, red curve). Similarly, we average the clear sky wind profiles on the left edge of the cloud, between 1210 and 1275 km along flight, to simulate a portion of clear sky wind observed by Aeolus (Fig. 9c, black curve)."

and added panel b) to Fig. 9 which shows what the airborne wind curtain looks like when processed like our Aeolus dataset.

l.549-550: stronger wind shear around the cloud-top inversion layer is a common occurrence, right?

AC : Right, this phenomenon is known and was observed locally with an airborne campaign (eg. Wang et al., 2008). and models (Hourdin et al., 2019)

Figure 9: there is no (a) and (b) in the figure itself

AC : (a) and (b) are now added.

l.591: please show the map here, not in the supplement.

AC : Agree, map in Fig. B2 was moved in Fig. 10a in the new manuscript

l.655: Is this really unexpected? You mention later this agrees with modeled results from the year 2000.

AC : Agree, we replace "One of the striking conclusions…" by "We observe for the first time from space, that…" in the new manuscript Sect. 4.3.2, line 770.

l.658-659: Are you sure these values are not for R²? Fig. 11 shows anticorrelation…

AC : We corrected this error. We replaced "correlation" by "anti-correlation". R is indeed negative.

Figure 11: this is just a correlation plot, you're not really quantifying convective momentum transport here, just validating K-theory relations. Please rephrase the subsection title to make it more accurate.

AC : Right, the old title of subsection 4.3.3 "First observation of Convective Momentum Transport" is now replaced by "First validation of K-theory for the wind in the free troposphere with Aeolus" in the new manuscript Sect. 4.3.2, line 769.

Figure B1 caption: incomplete, ends abruptly

Now corrected, the incomplete sentence was removed.

Figure B2: this figure fits more in the main manuscript in the methods section - or in the corresponding results section

Now corrected, Fig. B2 is now in Fig. 10a.

l.970: reference doi link incorrect, should be amt-15..

Right, this is now corrected.

New references added to the manuscript :

[revised manuscript text omitted]

---

## Author Response (AR2)

**Referee comments (questions) are in blue**

**Author comments (replies) are in black**

**Modifications made in the new manuscript by the authors are in red**

I refer to line numbers in the track-change version of the manuscript

**Replies to Referee #2**

I thank the authors for addressing my comments, I am satisfied with the vast majority of revisions and my impression is that the paper is in a very good shape now. The authors have produced a robust and very exciting dataset.

I recommend publication -- pending a couple specific comments on small parts of the manuscript that can be easily expanded a bit, and would strengthen the study further:

Note that I refer to line numbers in the tracked-changes version of the document.

AC : We thank referee #1 for their feedback and referee #2 for their additional reviews and suggestions. Figure 4 was added to compare perfectly co-located Mie and Rayleigh winds within the cloud mask. We also clarified Fig. 6 and added a panel c) in Fig. 12 to introduce the cloud top wind shear. In accordance with the suggestions of the editor, Figs. 11 and 13 (previously Figs. 10 and 12) along with Figs. A2 and A3 were reproduced using a colour scheme that is more suitable for readers with colour vision deficiencies. Please see the detailed response to the comments below.

**Specific comments:**

1)

I still miss a little bit more information on what exactly is being substituted by u_cloud in the allsky product.

1.1)

New text in l.418-420: "This small systematic difference is reassuring as the winds are perfectly co-located in the cloud, however, the standard deviation of the differences is quite large at 5.38 m/s"

RC on l.415-424 and Fig. B2:

--> To be sure about the source of std of the differences that you mention in this paragraph, a 2-dimensional pdf would be more informative:

I.e. probability density estimate (or simply a scatterplot) relative to:

Mie winds retrieved in cloudy sky (e.g. x-axis) vs Rayleigh winds that were substituted (y-axis). -- or alternatively vs the difference from Rayleigh winds.

AC: New Fig. 4 shows Rayleigh winds vs Mie winds when they both coexist within the cloud mask

[Figure]

**Figure 4: (a) 2D-PDF of pairs of colocated Mie winds and the Rayleigh winds when they both coexist within the cloud mask. The black dotted line represents the best linear regression. The 1:1 line is represented as a solid black line. For each point along this 1:1 line, a Gaussian was fitted to all data points lying along a perpendicular transect. Where the data spread and statistics allow a satisfactory fit, the maximum of the Gaussian is plotted as a red filled circle each 0.5 ms$^{-1}$. (b) Maximum of the Gaussian of the differences between Rayleigh and Mie winds within the cloud mask as a function of the Mie winds within the cloud mask. A sample of 50 orbit files of the year 2020 are analysed with a total of $10^6$ bins of 3 km x 480 m where both Rayleigh and Mie winds coexist within the cloud mask.**

Does spread (std) happen around the 1:1 line (zero-line if you use differences), or is the slope different from 1 (zero if you use differences), i.e. does one dataset underestimate the magnitude of positive/negative wind regimes? Note in the latter case, the differences still can average out .

AC: The following paragraph was added at the end of Sect. 3.1, lines 323-332:

"Figure 4a shows the 2D-PDF of pairs of colocated Mie winds and Rayleigh winds which coexist within the cloud mask. The distribution is located around the 1:1 line for the entire range of wind speed, and particularly between -50 and 50 ms$^{-1}$ (98.8% of the values). For Mie wind speed between -40 and 10 ms$^{-1}$, we systematically observe Mie winds up to 1 ms$^{-1}$ larger than the co-located Rayleigh winds (Fig. 4b). For wind speeds between 10 and 50 ms$^{-1}$, the systematic differences switch signs and Rayleigh winds are up to 1 ms$^{-1}$ larger than the co-located Mie winds (Fig. 4b). For most of the wind speed values encountered in the troposphere, pairs of co-located Mie and Rayleigh winds within the cloud mask agree well, with systematic differences below 1 ms$^{-1}$ (similar to the maximum bias of Rayleigh winds, Aeolus DISC, 2024). The large spread is essentially caused by the random error of Mie and Rayleigh winds. Therefore, given the finer spatial resolution, lower random and systematic errors of Mie winds, it is preferable to substitute Rayleigh winds by the Mie winds within the cloud mask, especially for the study of wind-cloud interactions."

and removed the following lines which used to describe the 1D-PDF

"For cloudy bins where both Mie winds and Rayleigh winds coexist, the average difference between Mie and Rayleigh wind is -0.20 ms$^{-1}$ from June to August 2020 (Fig. B2). This small systematic difference is reassuring as the winds are perfectly co-located in the cloud, however, the standard deviation of the differences is quite large at 5.38 ms$^{-1}$. This is essentially a consequence of the random error on wind observations (approximately 5 ms$^{-1}$ for the Rayleigh winds and 3 ms$^{-1}$ for the Mie winds) and the (at least) 5 times finer native horizontal resolution of the Mie winds compared to the Rayleigh winds. The substitution of Rayleigh winds by Mie winds in cloudy sky will therefore improve the study of wind-cloud interactions."

We added two sentences in the conclusion lines 878-880 :

"We showed that perfectly colocated Rayleigh and Mie wind values agree well within the cloud mask with differences below 1 ms$^{-1}$. As Mie winds have a better spatial resolution, lower systematic and random errors than Rayleigh winds, we substituted Rayleigh wind values by Mie wind values within the cloud mask."

Even in the extreme case where you have a distribution of Mie winds, and another distribution of near-zeros, you can get a PDF of the differences very similar to the one in Fig.B2. Note in Fig. 5 your values are within +-20m/s.

Important notes on this:

a) no additional data processing needed for addressing this, just plotting the data in a different way and adding a fit.

b) if such a figure is produced (showing a 2D pdf), it's worth adding it to the main manuscript: whether both winds agree well (little systematic difference despite high std), or whether the slope deviates from the expected agreement, both outcomes are an important result to show.

1.2)

Fig.5 (now Fig. 6):

in d) within the cloud mask there are actually still some u_clear values, correct?

AC : There are no u_clear in the cloud mask but there are indeed Rayleigh wind values in the cloud mask. By definition (lines 368-370), u_clear does not contain any Rayleigh wind values within the cloud mask as they are all substituted by Mie wind values when available and by no wind value otherwise.

I still miss a little bit more information on what exactly is being substituted by u_cloud in the allsky product."

Since the mask is shown with the black contours, I would ask to show the u_clear values that may exist within the mask in d).

As a reader, I'd like to see what exactly is being substituted by u_cloud in the allsky product.

--> This would really visualize and complement what is contained in the current Fig. B2: even if one gets a sense of little systematic differences, you should highlight that the cloudy part adds significantly to the dataset.

--> You mention earlier that "83% of... bins flagged as cloudy... contained both a Rayleigh and a Mie wind", and it's in this figure that you can describe how reliable Rayleigh is (compared to Mie), the deeper you go into the cloud (from the top).

AC : To clarify what exactly is being substituted by u_cloud in the allsky product, we built a new Fig. 6 (previous Fig. 5), which now contains all Rayleigh wind values (Fig. 6c) and Rayleigh winds only outside of the cloud mask, named u_clear (Fig. 6d). Accordingly, we also display all Mie wind values (Fig. 6e) and Mie wind values only within the cloud mask, named u_cloud (Fig. 6f). Figure 6g is u_allsky, the merging of u_clear and u_cloud.

[Figure]

**Figure 6: (a) Descending orbit segment crossing the tropical cyclone Paulette over the Atlantic ocean (2020-09-12T09–2020-09-12T11) plotted in red over a MODIS/Terra reflectance image. The red arrows represent the laser pointing direction. (b) Aeolus cloud mask. Aeolus (c) all Rayleigh winds and (d) Rayleigh winds only outside of the cloud mask, noted u_clear along the paper. Aeolus (e) all Mie winds, (f) Mie winds only within the cloud mask, noted u_cloud along the paper. (g) All-sky winds, noted u_allsky, result from the merging of u_clear and u_cloud. The winds are negative when blowing westward and positive when blowing eastward. For panels (b-g), the resolution of the re-sampled data is 3 km horizontally and 480 m vertically and the black contour is the cloud mask.**

We revised the final paragraph of Sect. 3.3, lines 464-489:

"Figure 6 illustrates how Aeolus resampled cloud mask and winds allow us to observe from space different features ranging from cyclones to cumulus clouds. During its lifetime, Aeolus observed multiple cyclones, sometimes crossing them near their centre (Marinescu et al., 2022). Figure 6a shows an example of intersection between Aeolus and the tropical cyclone Paulette over the Atlantic Ocean during the hurricane season, on 12 September 2020. The wind and cloud curtains are displayed between 20°N and 40°N (Fig. 6b-g). Note that Aeolus covers this distance in about 4 minutes, so the curtains represent a snapshot of the scene. The cyclone is identified by the continuous high cloud cover between 26°N and 32°N at about 12 km of altitude (Fig. 6b). The laser typically only penetrates 1 to 2 km below the uppermost cloudy layer of the cyclone. This particular case study is also interesting as it encounters a diversity of clouds. We observe a cirrus cloud, northward of the cyclone, extending from 33°N to 34°N and between 12 and 15 km of altitude. Along half of its length, this cirrus does not fully attenuate the laser as some clear sky layers can be retrieved below its base. We also observe shallow cumulus clouds (Fig. 6a, 6b) between 20°N and 26°N, with their tops below 3 km of altitude and sometimes only occupying a single profile, surrounded by clear sky profiles. This stresses out the importance of performing cloud detection at full horizontal resolution of 3 km.  Aeolus retrieves Rayleigh winds above and around the cyclone, up to 18 km of altitude (Fig. 6c). As the horizontal resolution of Rayleigh winds is fixed to 87 km, and molecular signal is still retrieved within clouds, some Rayleigh winds can be retrieved within clouds. For example, there are Rayleigh winds within the upper cloudy layers of the cyclone and in the entire boundary layer, even within shallow cumulus clouds (Fig. 6c). However, we only keep Rayleigh wind values outside of the cloud mask when building $u_{clear}$ (Fig. 6d). The cross section of $u_{clear}$ (Fig. 6d) reveals the wind shear found where counter-clockwise winds around the cyclone base meet the clockwise winds at the top of the cyclone. This happens at about 8 km of altitude at 25°N and at 35°N. The further we look from the cyclone, the higher in altitude the reversal of the wind occurs. Figure 6e shows the Mie winds retrieved by Aeolus. Most Mie winds are retrieved within the cloud mask. As the native resolution of Mie winds can be as coarse as 15 km, it is possible that Mie winds extend horizontally beyond the cloud mask as shown around shallow cumulus clouds, between 20°N and 26°N, below 3 km of altitude (Fig. 6e). $u_{cloud}$ (Fig. 6f) contains only Mie winds values within the cloud mask (as detailed in Sect 3.1). The merging of $u_{clear}$ and $u_{cloud}$ constitutes the all-sky wind, $u_{allsky}$ (Fig. 6f)."

2)

Response to specific comment 6

"The referee is right to point out that adding the information about S at the cloud top and above would be very interesting. We plan to do it in a following study dedicated to specific cloud types and associated scientific questions".

RC: I'd still suggest adding it -- including this would require very little work, as the parameter is already calculated, and it wouldn't really take much from your follow-on study. On the contrary I think it would strongly motivate the mentioned follow-on study. While reading this section of the paper this is the first thing that pops in my mind as a tiny missing element that would strengthen and complement the section a lot.

-- Having said the above, it's also not a critical part of the study so I'm fine if the Editor does not deem it necessary.

Cloud top wind shear profiles are added in the new panel c) of Fig. 12 (previously Fig. 11) and occurrences of cloud top wind shear measurements at each altitude are added in Fig. B10. We also updated Fig. B9 accordingly to describe where and how the cloud top wind shear $S_{cloud\_top}$ is computed.

[Figure]

**Figure 12: (a) average wind speed profiles retrieved within the uppermost cloudy layer $u_{cloud\ up}(z)$ and average of the closest clear sky wind speed $u_{clear\ surrounding\ cloud\ up}(z)$ observed over each region. Only values where $u_{cloud\ up}(z)$ and $u_{clear\ surrounding\ cloud\ up}(z)$ are significantly different (two sided T-test with p-value < 0.05) are plotted. (b) Average wind shear profiles within the cloud $S_{cloud}(z)$, and in the surrounding clear sky $S_{clear\ surrounding\ cloud}(z)$ are computed at each altitude z using the the wind speed observed at $z_1$ located 1 km below z and at the wind speed observed at $z_2$ located 1 km above z. Note that for (a) and (b), only a sample of the data is used as each cloud should be at least 2 km thick vertically, and the horizontal distance between $S_{cloud}(z)$ and $S_{clear\ surrounding\ cloud}(z)$ must be < 100 km and only values where $S_{cloud\ up}(z)$ and $S_{clear\ surrounding\ cloud\ up}(z)$ are significantly different (two sided T-test with p-value < 0.05) are plotted. (c) Average cloud top wind shear $S_{cloud\ top}(z)$ profiles retrieved in each region during the year 2020 using all data collected by Aeolus over each region contrarily to (a) and (b) that use only a sample of the data.**

[Figure]

**Figure B10: Occurrences of cloud top wind shears over each region**

[Figure]

**Figure B9: complements relative to the calculations of wind shears for Fig. 11. For each profile containing a cloud, we compute $S_{cloud\_top}(z)$ the wind shear between the clear sky above the cloud and the uppermost cloudy layer. We extract a sample of these profiles which has to respect two conditions : the cloud must be at least 2 km thick vertically, and there must be clear sky in the surrounding, within a distance of 100 km. We compute $S_{cloud}(z)$ the wind shear within the cloud, and $S_{clear\_surrounding\_cloud}(z)$ the wind shear in the surrounding of the cloud. Wind shears are calculated over 2 km thick layers.**

We added the following paragraph in the revised manuscript in Sect. 4.3.3, lines 798-809:

"By retrieving the wind both within clouds and above cloud tops, this Aeolus dataset gives access to the wind shear at the top of the clouds (Fig. 12c). Note that to study the wind shear at the top of clouds (Fig. 12c), we analyse all the data collected over each region contrarily to Fig. 12a and 12b. Within each region, we record the wind speed observed in the uppermost cloudy layer (noted $u_{cloud\_up}$) and the clear sky wind speed 2 km above (noted $u_{above\_cloud}$). We then compute $S_{cloud\_top}$, the wind shear between these two layers (see Fig. B9). Figure 12c shows the average profile of $S_{cloud\_top}(z)$ for the different regions. Within the lower troposphere, the largest number of cloud top wind shear observations is found between 2 and 3 km of altitude over the TrSc and Cu regions (Fig. B10), which is consistent with Wood (2012) and Cesana et al., (2019). At these altitudes, the average wind shear at cloud top $S_{cloud\_top}(z)$ ($2 \times 10^{-3}$ to$3 \times 10^{-3}$ s$^{-1}$, Fig. 12c), is larger than in all sky conditions $S_{allsky}(z)$ (about $1.5 \times 10^{-3}$ s$^{-1}$, Fig. 11c). This result is consistent with previous work stating that a temperature inversion above cloud tops isolates the cloudy layer from the clear sky above. A zone of larger wind shear can thus develop around the temperature inversion (Wang et al., 2008; Hourdin et al., 2019), which can in turn affect the morphology of these clouds through entrainment and drying of the boundary layer (Schulz & Mellado, 2018; Zamora Zapata et al., 2021)."

We added a sentence in the conclusion, lines 902-903:

"We also found that the observed cloud top wind shear above Stratocumulus and Cumulus clouds ($2 \times 10^{-3}$ to $3 \times 10^{-3}$ s$^{-1}$) was larger than the observed all-sky wind shear at the same altitude ($1.5 \times 10^{-3}$ s$^{-1}$)."

**Minor/technical comments:**

1)

Figure AC2.1:

This would be a very welcome addition to the supplement. Please specify in the main manuscript -- when discussing Fig.7 -- that for completeness you show absolute shear values in the appendix.

Fig. B11 was added in the Appendix as well as the formula to compute the absolute wind shear. This new figure is mentioned in the manuscript in Sect. 4.1, line 547: "Note that a similar map, but with absolute wind shears is shown in Fig. B11."

[Figure]

**Figure B11: Map of the median absolute wind shear $|S_{allsky}|$ calculated a) between 8 km and 10 km, b) between 12 and 14 km and c) between 16 and 18 km from June to August 2020. Contours represent cloud covers of 5, 10, 15 and 20% for each altitude range.**

2)

l.343-345: reference needed, need to be specific about which of the above references are the latest you refer to.

Reference to the latest Aeolus DISC report was added in Sect. 3, lines 253-255 :

"The latest validation report of Aeolus showed systematic error (bias) of below 0.5 ms-1 for Mie winds and below 1 ms-1 for Rayleigh winds, while the random error is about 3 to 4 ms-1 for Mie winds and 3 to 6 ms-1 for Rayleigh winds (Aeolus DISC, 2024)."

and replace the old version that was in the track change manuscript : ""

3) Perhaps I missed it, but now the INDOEX region is no longer included in Fig. 11 (now Fig. 12, Fig.10 in previous manuscript version) and I can't seem to find any note/reference/justification for that change.

Correct. The INDOEX region is no longer visible in Fig. 12 as there are no altitudes at which both $u_{cloud\_up}$ and $u_{clear\_surrounding\_cloud\_up}$, $S_{cloud}$ and $S_{clear\_surrounding\_cloud}$ are significantly different. We now mention in the text why INDOEX is not visible in Fig. 12 in Sect. 4.3.2, lines 779-781 : "Note that over the INDOEX region, there are no altitudes where $u_{cloud\_up}(z)$ and $u_{clear\_surrounding\_cloud\_up}(z)$ are significantly different and where $S_{cloud}(z)$ and $S_{clear\_surrounding\_cloud}(z)$ are significantly different, hence this region does not appear in Fig. 12a and Fig. 12b."

4)

l.665-680: please add a little bit of discussion regarding shear, use the Jensen reference here already (you name it in the next subsection)

We added two pieces of discussion regarding Fig. 8. In Sect. 4.1 lines 555-557 :

"Jensen et al., (2025) demonstrated that wind shears of $10\times10^{-3}$ s$^{-1}$ were favourable for a faster sublimation of cirrus clouds particles, reducing the lifetimes of these clouds."

you may add also e.g. Schaefler et al. (2020) https://doi.org/10.1175/MWR-D-19-0229.1

e.g. are your magnitudes of shear near the jets close to their Fig. 9?

and in Sect. 4.1 lines 571-577 :

"At the North bound of the map, at about 50°N, the tropopause layer is located between 9 and 10 km of altitude at the end of boreal summer (Schäfler et al., 2020). We observe a wind shear of about $1\times10^{-3}$ s$^{-1}$ around the globe at this latitude (Fig. 8c). Indeed, the wind profile is tilted eastward below the tropopause and westward above, which explains the weak positive wind shear around the tropopause. Schäfler et al., (2020) reported a weak, but negative wind shear of $-1\times10^{-3}$ s$^{-1}$ between 8 and 10 km for the month of October at about 60°N. The change of sign might be explained by a lower altitude tropopause at 60°N, 10 degrees northward of our observations, and thus by a larger contribution of the westward tilted profile above the tropopause."

Note that to derive the wind shear from Schäfler et al., 2020, we used the average over all flights (their Fig. 1) rather than the case study (their Fig. 9).

5)

l.906-907: "This suggests an important role...": sentence too vague, and wind shear role on convective organization is well known. Can be removed, or expanded with proper referencing.

We removed the following sentence in Sect. 4.3.2, line 734: ""

---

## Author Response (AR3)

**Referee comments (questions) are in blue**

**Author comments (replies) are in black**

**Response to the editor**

Dear Dr. Titus,

Thank you for your thorough revision of your manuscript. The reviewers are satisfied with the current version, safe for one small issue about figure 4a. If you could check that you did the correct linear regression, I am happy to accept your manuscript for publication. Thank you for choosing ACP!

AC : We thank referee #1 for their feedback, referee #2 for their feedback and final remarks, and the editor for their supervision of the entire process. Figure 4 was updated with the corrected linear regression. No changes were made to the text of the manuscript. Please see the detailed response below.

**Specific comment from anonymous reviewer #2 :**

**Could the authors double-check the best linear regression in Fig. 4a ?**

The vast majority of the data lies within +-20m/s, where Rayleigh winds end up being +-1m/s stronger than Mie.

I'd expect that the overall slope is dominated by the region with most data, which should be slightly larger than 1. The actual slope is 0.9 instead. It might be that the fit is done with the datasets switched in the equation: X actually being Rayleigh winds instead of Mie in the code?

Correct, the fit was performed with the datasets switched in the equation. In the updated Fig. 4 (see below) the fit is performed with X being Mie winds and Y being Rayleigh winds. The slope increases from 0.90 in the previous version to 0.99.

[Figure]

**Figure 4: (a) 2D-PDF of pairs of colocated Mie winds and the Rayleigh winds when they both coexist within the cloud mask. The black dotted line represents the best linear regression. The 1:1 line is represented as a solid black line. For each point along this 1:1 line, a Gaussian was fitted to all data points lying along a perpendicular transect. Where the data spread and statistics allow a satisfactory fit, the maximum of the Gaussian is plotted as a red filled circle each 0.5 ms⁻¹. (b) Maximum of the Gaussian of the differences between Rayleigh and Mie winds within the cloud mask as a function of the Mie winds within the cloud mask. A sample of 50 orbit files of the year 2020 are analysed with a total of $10^6$ bins of 3 km x 480 m where both Rayleigh and Mie winds coexist within the cloud mask.**